# Old carbon routed from land to the atmosphere by global river systems

Joshua F. Dean[1✉], Gemma Coxon[1], Yanchen Zheng[1], Jack Bishop[1], Mark H. Garnett[2], David Bastviken[3], Valier Galy[4], Robert G. M. Spencer[5], Suzanne E. Tank[6], Edward T. Tipper[7], Jorien E. Vonk[8], Marcus B. Wallin[9], Liwei Zhang[10], Chris D. Evans[11] & Robert G. Hilton[12✉]

Rivers and streams are an important pathway in the global carbon cycle, releasing carbon dioxide ($CO_2$) and methane ($CH_4$) from their water surfaces to the atmosphere[1,2]. Until now, $CO_2$ and $CH_4$ emitted from rivers were thought to be predominantly derived from recent (sub-decadal) biomass production and, thus, part of ecosystem respiration[3–6]. Here we combine new and published measurements to create a global database of the radiocarbon content of river dissolved inorganic carbon (DIC), $CO_2$ and $CH_4$. Isotopic mass balance of our database suggests that 59 ± 17% of global river $CO_2$ emissions are derived from old carbon (millennial or older), the release of which is linked to river catchment lithology and biome. This previously unrecognized release of old, pre-industrial-aged carbon to the atmosphere from long-term soil, sediment and geologic carbon stores through lateral hydrological routing equates to 1.2 ± 0.3 Pg C year$^{-1}$, similar in magnitude to terrestrial net ecosystem exchange. A consequence of this flux is a greater than expected net loss of carbon from aged organic matter stores on land. This requires a reassessment of the fate of anthropogenic carbon in terrestrial systems and in global carbon cycle budgets and models.

River networks form a crucial link between the terrestrial, atmospheric and marine carbon cycles, storing, transforming and exporting inorganic and organic carbon[3,4]. Globally, rivers and streams emit an estimated 2.0 (1.6–2.2) Pg C year$^{-1}$ to the atmosphere as $CO_2$, along with 28 (16.7–39.7) Tg of $CH_4$ per year (refs. 1,2,4,5). These carbon emissions are equivalent to 59% of net terrestrial carbon uptake (net ecosystem exchange)[7] or about 1.8% of terrestrial gross primary production (GPP)[6]. Export of carbon by rivers is often the second largest component of ecosystem carbon loss after soil respiration[8]. The age of the carbon fuelling river emissions to the atmosphere—whether supplied by rapid, sub-decadal or much older sources—is a notable knowledge gap in pre-industrial, contemporary and future carbon cycles[5].

Rivers are at the interface of carbon cycling across timescales. A large part of river $CO_2$ emissions is generated by a combination of terrestrial respiration of organic carbon recently fixed by photosynthesis and within-river production and respiration[9–11]. Thus, river $CO_2$ emissions are generally considered a component of the contemporary carbon cycle fuelled by annual to decadal carbon turnover[3,5,6]. However, rivers also transport older carbon, such as organic matter in particulate[12,13] and dissolved forms[14–16], whereas aged riverine DIC[17,18], $CO_2$ (refs. 19,20) and $CH_4$ (ref. 21) have all been directly observed.

River DIC, $CO_2$ and $CH_4$ ages vary based on the source of carbon delivered to rivers. The oldest carbon stores in river catchments are rock-derived ('petrogenic' or geologic) carbon in carbonate minerals and rock organic matter. Chemical weathering and erosion can mobilize these carbon sources and route them into rivers[22–26]. By contrast, heterotrophic respiration of soil organic matter can produce $CO_2$ and $CH_4$ ranging in age from several years to millennia[27,28]. These gases can be dissolved in water and moved from soils and sediments into stream and river waters. Older carbon, which can be sourced from deeper in soil profiles[29], represents a reintroduction of previously stored soil carbon to the contemporary carbon cycle and, where associated with anthropogenic perturbations such as land-use change, may represent a source of anthropogenic greenhouse gas emissions[15,18,30,31]. We define these three potential river carbon sources as 'decadal' (fixed into the biosphere through photosynthesis since 1955), 'millennial' (biospheric carbon that is hundreds to several thousands of years old) and 'petrogenic' (older than about 55,000 years)[3,32]. To understand the role of global river carbon emissions in the climate system, it is essential to determine the relative contributions of decadal inputs versus these 'old' (millennial-aged and petrogenic) carbon sources.

Here we constrain the age and source of river carbon emissions at the global scale using the radiocarbon composition (reported as fraction modern, $F^{14}C$ (ref. 33)) of river DIC, $CO_2$ and $CH_4$ (Fig. 1 and Supplementary Information section 1). The $F^{14}C$ value of DIC provides a surrogate for the isotope composition of river $CO_2$ emissions owing to the fast equilibration times between DIC and $CO_2$ relative to water flow path lengths (Supplementary Information section 2). We provide an extra subset of paired DIC and $CO_2$ $F^{14}C$ measurements to show that these values are generally within 0.02 of one another (for $F^{14}C$ of 1.0

[1]School of Geographical Sciences, University of Bristol, Bristol, UK. [2]NEIF Radiocarbon Laboratory, Scottish Universities Environmental Research Centre, East Kilbride, UK. [3]Department of Thematic Studies – Environmental Change, Linköping University, Linköping, Sweden. [4]Marine Chemistry & Geochemistry, Woods Hole Oceanographic Institution, Woods Hole, MA, USA. [5]Department of Earth, Ocean and Atmospheric Science, Florida State University, Tallahassee, FL, USA. [6]Department of Biological Sciences, University of Alberta, Edmonton, Alberta, Canada. [7]Department of Earth Sciences, University of Cambridge, Cambridge, UK. [8]Department of Earth Sciences, Vrije Universiteit Amsterdam, Amsterdam, The Netherlands. [9]Department of Aquatic Sciences and Assessment, Swedish University of Agricultural Sciences, Uppsala, Sweden. [10]State Key Laboratory of Estuarine and Coastal Research, East China Normal University, Shanghai, China. [11]UK Centre for Ecology & Hydrology, Bangor, UK. [12]Department of Earth Sciences, University of Oxford, Oxford, UK. ✉e-mail: josh.dean@bristol.ac.uk; robert.hilton@earth.ox.ac.uk

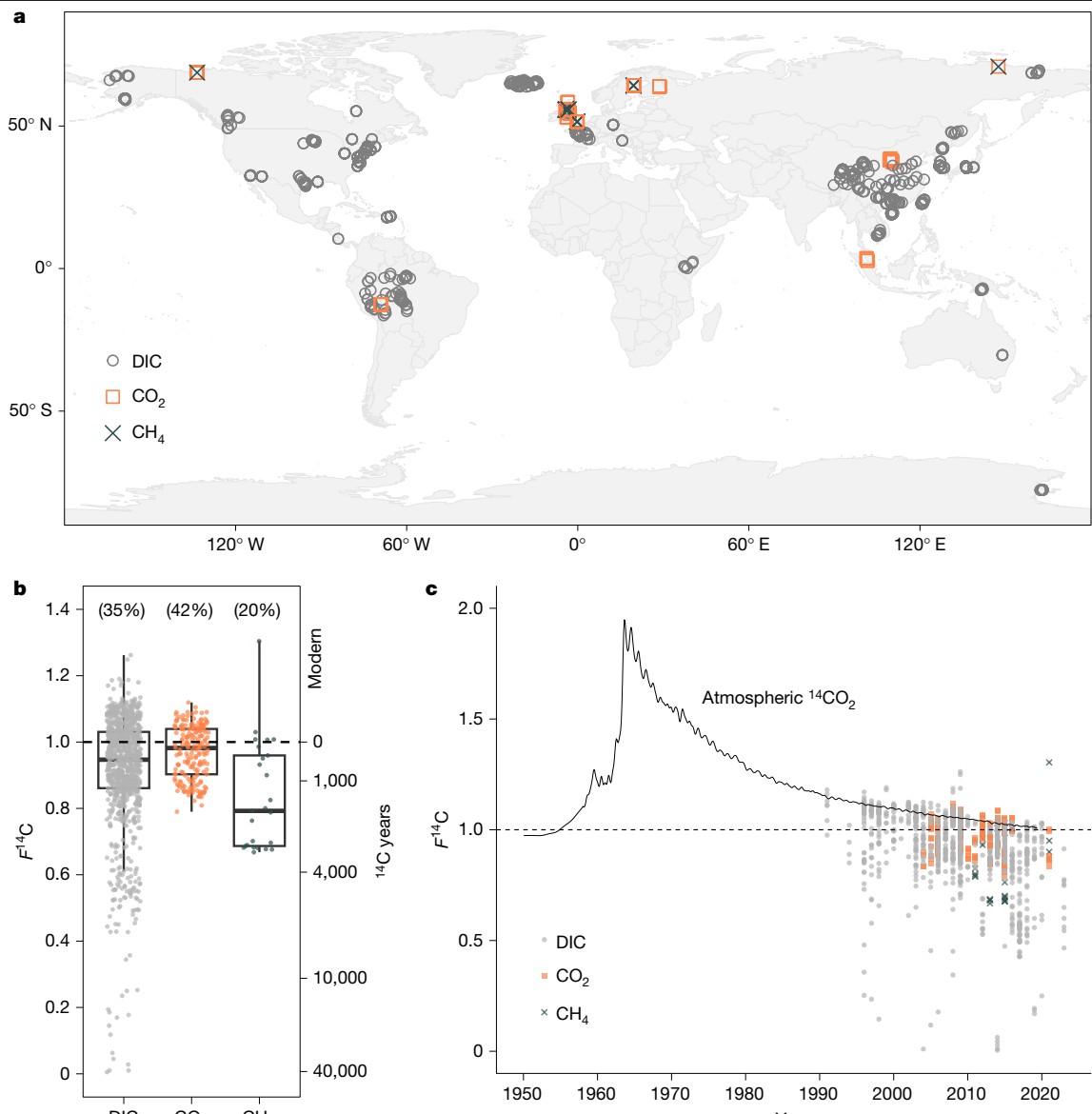

**Fig. 1 | Global $^{14}$C patterns in river DIC, $CO_2$ and $CH_4$. a**, Map of sampling locations. **b**, All $F^{14}$C values assembled in the database separated by compound. The dashed horizontal line shows $F^{14}$C = 1.0 (atmospheric $CO_2$ in 1955 CE, panel **c**), mean age in uncalibrated $^{14}$C years is indicated on the right axis, values in parentheses indicate percentage of observations for which $F^{14}$C > 1.0 (younger than 1955 CE, for which no $^{14}$C age can be calculated and are considered 'modern');

lines in the middle of the boxes represent the median, box limits represent the upper and lower quartiles and whiskers extend to 1.5 times the interquartile range. **c**, All $F^{14}$C values assembled in the database plotted by year of sample collection and shown in the context of atmospheric $^{14}CO_2$ (1950 to 2019 CE; black line; data from ref. 34). The dashed horizontal line shows $F^{14}$C = 1.0; $F^{14}$C values are shown separated by compound.

versus 0.98, this equates to 162 $^{14}$C years), which, although higher than analytical uncertainty, is 5–20 times smaller than the variability we find across the entire database assembled here (Extended Data Table 1 and Supplementary Fig. 1). As such, the large DIC component of our database allows us to robustly assess the radiocarbon content of river $CO_2$ emissions. The assembled database contains 1,141 published observations and 54 new measurements (1,195 total from 67 distinct studies; Supplementary Table 1) and includes observations across most of the main land masses, biomes and lithologies, including North and South America, Iceland, Europe, Scandinavia, East Africa, China, Southeast Asia, Australia and Antarctica (Fig. 1a, Supplementary Figs. 2 and 3 and Supplementary Information section 1). Overall, the distribution of sample locations captures global proportions of the main lithologies and biomes (Supplementary Information section 1 and Supplementary Fig. 4).

## Age of river $CO_2$ and $CH_4$ emissions

The mean $F^{14}$C of all river DIC and $CO_2$ measurements were 0.914 ± 0.184 (±1$\sigma$) and 0.961 ± 0.074, respectively, equivalent to radiocarbon ages of 722 ± 1,264 and 320 ± 483 $^{14}$C years. The mean $F^{14}$C for $CH_4$ was 0.879 ± 0.167 (1,036 ± 1,364 $^{14}$C years), but because we found a more limited number of available $^{14}CH_4$ observations, we focus our analysis on DIC and $CO_2$ measurements (Extended Data Table 1). There is notable variability in atmospheric $^{14}CO_2$ content over the past 70 years, driven by nuclear weapons testing (increasing $F^{14}$C values to above 1.0 since 1955 CE), dilution by fossil fuel emissions (lowering $F^{14}$C values) and variability in rates of natural $^{14}$C production in the troposphere[34] (Fig. 1c). River $CO_2$ and $CH_4$ with $F^{14}$C values > 1.0 are thus expected if the degradation of organic matter formed through photosynthesis since 1955 is generating these greenhouse gases.

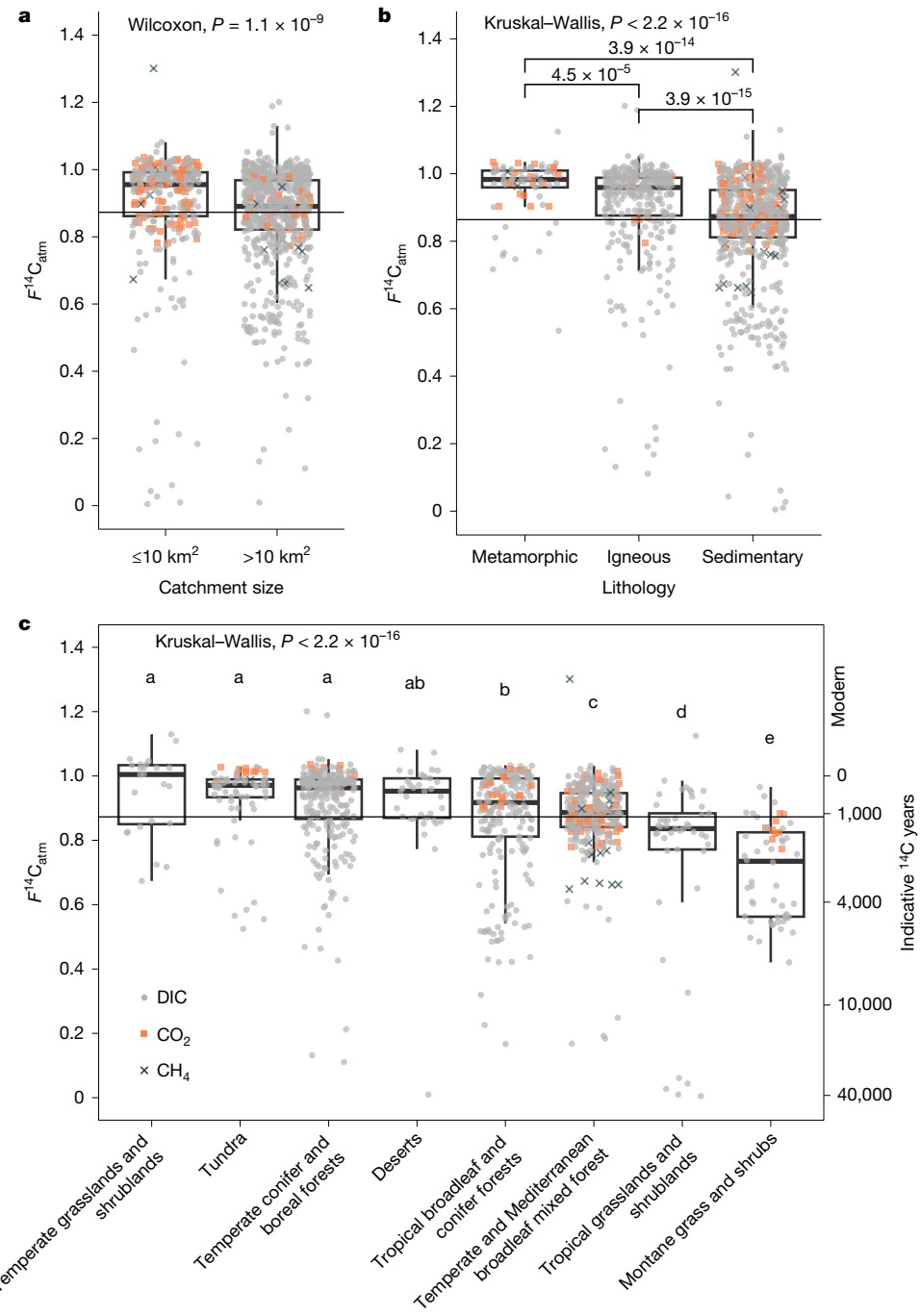

**Fig. 2 | Influence of catchment size, river-reach biome and lithology on $^{14}$C in river DIC, $CO_2$ and $CH_4$. a**, Normalized $F^{14}C_{atm}$ values for DIC, $CO_2$ and $CH_4$ separated by catchment size, either ≤10 km$^2$ or >10 km$^2$ (Methods and Supplementary Fig. 6); statistical difference is indicated by the *P* value (shown at the top) derived from an unpaired two-sample Wilcoxon test. **b**, $F^{14}C_{atm}$ values separated by lithology of the river reach (within a 1-km$^2$ radius of the sampling location) as defined in HydroATLAS and binned for comparison (Methods and Supplementary Fig. 3); statistically significant differences are indicated by *P* values when comparing across all three lithologies using a Kruskal–Wallis test (shown at the top) and unpaired two-sample Wilcoxon tests (*P* values and horizontal bars). **c**, $F^{14}C_{atm}$ values separated by the biome of the river reach (Methods and Supplementary Fig. 2); lowercase letters indicate statistically significant differences (*P* < 0.05) using a Kruskal–Wallis test (*P* value shown at the top) and Conover–Iman post hoc. The horizontal black line in each panel represents the mean-normalized $F^{14}C_{atm}$ for all samples; box and whisker dimensions follow Fig. 1b.

Most $F^{14}C$ observations were less than 1.0 (62–74% for DIC, $CO_2$ and $CH_4$; Fig. 1b and Supplementary Fig. 5), indicating that carbon older than 1955 is contributing to river emissions. The remaining 26–38% of $F^{14}C$ observations were greater than 1.0, which can be explained by inputs from decadal carbon sources. To further explore the youngest carbon sources to river systems independent of atmospheric $^{14}CO_2$ variability, we calculated $F^{14}C_{atm}$, which represents the fraction modern for each observation after normalizing to atmospheric $^{14}CO_2$ in the year of sample collection[35]. The database reveals 430 (36%) $F^{14}C_{atm}$ values that are greater than 1.0 (Extended Data Fig. 1). In many terrestrial ecosystems, the residence time of carbon is very short, on the order of less than 20 years (refs. 36,37). The highest river $F^{14}C_{atm}$ values suggest an efficient route for ecosystem respiration into some streams and rivers. Otherwise, the $F^{14}C_{atm}$ values suggest

a declining trend from 1991 to 2023 ($R^2 = 0.04$, $P \ll 0.001$; Extended Data Fig. 1).

Low (old) $F^{14}C_{atm}$ values were prevalent in catchments of all sizes (Fig. 2a and Extended Data Fig. 2). We would expect larger catchments to be less affected by specific processes that mobilize old carbon (for example, localized erosion or groundwater inputs) and therefore be more likely to find older carbon in smaller catchments. By contrast, we found that $F^{14}C_{atm}$ values were lower (older) as catchments got larger (Extended Data Fig. 2), suggesting that contributions of old carbon to river $CO_2$ from deeper hydrologic flow paths or exposed old carbon stores are occurring across large scales.

An important spatial predictor of $F^{14}C_{atm}$ was lithology (Fig. 2b), with catchments underlain by sedimentary lithologies, including carbonates, having a lower mean $F^{14}C_{atm}$ (0.848 ± 0.159, median = 0.873 for DIC, $CO_2$ and $CH_4$) compared with igneous (0.903 ± 0.156, median = 0.959; $P = 4.5 \times 10^{-5}$) and metamorphic (0.957 ± 0.092, median = 0.983; $P = 3.9 \times 10^{-15}$) lithologies. Weathering of carbonate minerals and rock organic carbon in sedimentary lithologies contributes petrogenic carbon ($F^{14}C \approx 0$) to rivers from weathering processes[38,39] (Supplementary Information section 2); these petrogenic contributions from rock weathering and oxidation would be unlikely in igneous or metamorphic lithologies. Therefore, other $^{14}C$-depleted carbon contributions are required to explain the prevalence of $F^{14}C_{atm}$ values of less than 1.0 across all lithologies.

The variability of river $F^{14}C_{atm}$ values across biomes was statistically significant (Fig. 2c), but there were no clear trends based on differences between biomes. The lowest $F^{14}C_{atm}$ values (oldest) were collected from the montane grassland and tropical grassland and shrubland biomes. In high-elevation zones, climatic and geomorphic conditions can promote erosion (for example, steep channel slopes, landslides), which could increase petrogenic inputs from the underlying sedimentary lithologies and thus lower $F^{14}C_{atm}$ values. In other more productive temperate and tropical biomes, decadal-aged to millennial-aged carbon inputs could be supplied from recently fixed organic carbon and older soil carbon pools[27] such as peat[19,40], particularly where soils have been affected by drainage[31] or agriculture[15].

To further explore the potential drivers of river DIC $F^{14}C_{atm}$, we applied a random forest model to determine which parameters could explain the $F^{14}C_{atm}$ DIC dynamics in the database[41]. For large catchments (>10 km² in area), the most important parameters (in descending order of importance) were: mean annual precipitation, mean elevation, mean annual air temperature, karst percentage cover and forest percentage cover of the catchment (Extended Data Fig. 3). For small catchments (≤10 km²), these were slightly different: mean elevation, soil organic carbon content, soil sand content and mean annual air temperature of the river reach (within a 1-km² radius; Extended Data Fig. 4). Mean elevation (large and small catchments) and karst area (large catchments only) had a negative relationship with $F^{14}C_{atm}$, indicating that catchments with higher elevations and carbonate lithologies released more $^{14}C$-depleted (older) DIC. Mean annual precipitation (large catchments) and temperature (large and small catchments) were generally positively related to $F^{14}C_{atm}$, although there was an upper limit to this influence in large catchments: above 2,000 mm rainfall and above 20 °C, $F^{14}C_{atm}$ tended to decrease (Extended Data Fig. 3b,d). This suggests that catchments receiving higher precipitation and with warmer temperatures tended to release less $^{14}C$-depleted (younger) DIC, although the limit to this mechanism indicates that more arid or especially warm and wet regions may store more carbon and/or release older carbon. In small catchments, the high (>15) increase in the mean square error values from the random forest model for soil organic carbon and sand content demonstrate the potential importance of small-scale controls on the age of DIC released by rivers, influencing organic carbon mobilization and hydrologic flow paths. These results, and the influence of lithology and biome, highlight that river DIC and $CO_2$ are driven by more than just recently fixed carbon, with important contributions from millennial and petrogenic carbon sources.

**Table 1 | Modelled source contributions to global river $CO_2$ emissions**

| | Vertical $CO_2$ emissions | | | |
| | Monte Carlo simulation | | Bayesian isotope mixing model | |
| | Proportion | $CO_2$ emissions (PgC year$^{-1}$) | Proportion | $CO_2$ emissions (PgC year$^{-1}$) |
| --- | --- | --- | --- | --- |
| Decadal | 0.41±0.16 | 0.9±0.3 | 0.50±0.12 | 1.0±0.2 |
| Millennial | 0.52±0.16 | 1.1±0.3 | 0.34±0.17 | 0.7±0.3 |
| Petrogenic | 0.07±0.01 | 0.1±0.0 | 0.15±0.06 | 0.3±0.1 |
| 'Old' carbon | 0.59±0.17 | 1.2±0.3 | 0.49±0.23 | 1.0±0.4 |

First-order estimates of the contributions of decadal, millennial and petrogenic carbon sources to $CO_2$ emissions (proportions and flux, PgC year$^{-1}$; mean±1σ) by global river systems from the two-endmember isotope mixing model and Monte Carlo simulation (assuming known petrogenic inputs as a prior; equation (7); Extended Data Fig. 5) and the three-endmember Bayesian isotope mixing model (excluding any previous fluxes, mean±1σ from the model scenarios; Supplementary Table 2 and Extended Data Fig. 6). 'Old' carbon represents combined millennial and petrogenic contributions.

## Old $CO_2$ emitted from global rivers

To better constrain the origin of river $CO_2$ emissions, we modelled the potential contributions from decadal and millennial carbon sources after accounting for published estimates of river petrogenic inputs from carbonate mineral and rock organic matter weathering. Owing to limited data availability, it is not possible to correlate our $F^{14}C$ data with catchment-specific weathering information (for example, solute export[22]), so we take a global view using our mean $F^{14}C$ DIC values and assess the petrogenic inputs using global estimates of carbonate and rock organic carbon weathering rates. Using the global river $CO_2$ emission flux of 2.0 ± 0.2 Pg C year$^{-1}$ (refs. 5,6), a lateral export of DIC to the oceans of 0.5 Pg C year$^{-1}$ (ref. 42), we then account for the estimated range of petrogenic inputs by rock weathering of 0.150–0.218 Pg C year$^{-1}$ (refs. 22,39). The non-petrogenic flux of river $CO_2$ and DIC is thus 2.28–2.35 Pg C year$^{-1}$ and has an $F^{14}C$ value of 0.978–1.007 (Methods; equation (7)).

The remaining total DIC flux, having accounted for petrogenic carbon inputs, must be some mixture of: (1) DIC supplied from soil or atmospheric $CO_2$ during carbonate and silicate weathering; (2) $CO_2$ derived from ecosystem respiration supplied by hydrological flow paths through shallow and deeper soils; (3) $CO_2$ derived from within-river heterotrophic and autotrophic respiration; and (4) invasion of atmospheric $CO_2$ if rivers are undersaturated with respect to the atmosphere. We note that (1) and (2) may be supplied concurrently in some systems, whereas (2) and (3) include direct soil respiration plus soil decomposition products that can be respired within rivers, such as dissolved and particulate organic carbon[43,44]. Carbon inputs (1) to (4) can be decadal in age ($F^{14}C$ value similar to the atmosphere; Fig. 1c), whereas (1) to (3) can be millennial in age if derived from deeper soil respiration flushed to rivers laterally by hydrological flow paths.

Using the non-petrogenic component of the river DIC pool (vertical $CO_2$ emission plus lateral DIC export) of 2.28–2.35 Pg C year$^{-1}$ and its $F^{14}C$ value of 0.978–1.007, we use a two-endmember isotope mixing model and Monte Carlo simulation to estimate the remaining proportional contributions of carbon inputs from decadal and millennial carbon sources (Methods and Extended Data Fig. 5). The mean modelled proportional contribution to global river $CO_2$ from decadal carbon sources was 0.41 ± 0.16 (±1σ), equivalent to a vertical emission flux of 0.9 ± 0.3 Pg C year$^{-1}$; for millennial sources, the mean proportional contribution was 0.52 ± 0.16, or 1.1 ± 0.3 Pg C year$^{-1}$ (Table 1). When petrogenic and millennial soil carbon inputs are combined, we estimate that these old carbon sources (millennial or greater in age) could be contributing as much as 1.2 ± 0.3 Pg C year$^{-1}$ to the atmosphere

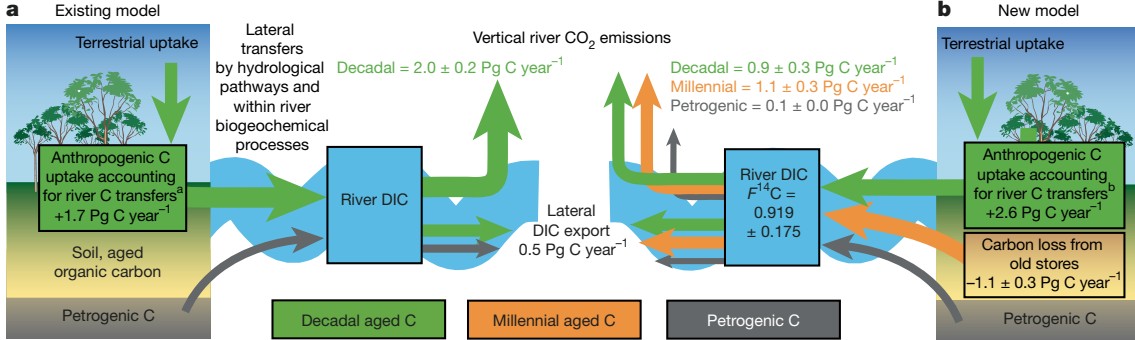

**Fig. 3 | The importance of river CO₂ emission age for the global carbon cycle.**
**a**, Existing model in which river $CO_2$ is only derived from young, rapid-cycling carbon (decadal-aged = green); lateral DIC export to the coast is considered a mixture of decadal and petrogenic inputs (grey). By accounting for these river carbon losses, it is estimated that 1.7 Pg C year⁻¹ of anthropogenic carbon emitted to the atmosphere may be accumulating in the rapid-cycling terrestrial carbon pools[3]; [a]note that this estimate is based on a lower estimate of vertical river $CO_2$ emissions of 1.51 Pg C year⁻¹. **b**, Revised conceptual model based on the assembled $F^{14}C$ values of river DIC, $CO_2$ and $CH_4$ presented here; millennial carbon inputs are needed from organic matter degradation in soils or river sediments (orange) as well as petrogenic carbon from rock weathering to explain the observed $F^{14}C$ values in our database. This revised conceptual model indicates a loss of carbon from an old (millennial) store on land through vertical river $CO_2$ emissions; a first-order estimate of the impact on the partitioning of carbon in the biosphere and soils is provided ([b]).

from global river systems. To independently assess the outputs of this petrogenic-constrained, two-endmember isotope mixing model, we also ran a Bayesian isotope mixing model (Supplementary Information section 3), which quantified potential contributions from three carbon sources (decadal, millennial and petrogenic) without any priors other than $F^{14}C$ ranges for each endmember (Extended Data Fig. 6 and Supplementary Table 2). The two-endmember Monte Carlo simulation and three-endmember Bayesian analysis outputs were in agreement with one another (Table 1). Altogether, these findings suggest that across both biome and lithological variability, a third to two-thirds of river $CO_2$ emissions are derived from old carbon sources (Table 1 and Supplementary Table 2).

## A new conceptual model of river CO₂

Widespread contribution of old carbon to river $CO_2$ emissions challenges existing models (Fig. 3a). River $CO_2$ emissions are commonly assumed to be dominated by the lateral routing of terrestrial GPP, alongside within-river production[3–6]. Some estimates of global river $CO_2$ emissions state that petrogenic carbon sources are minor, based on a limited number of ¹⁴C-DIC observations[6]. Other inland water $CO_2$ emission syntheses have noted that millennial and petrogenic inputs are a substantial knowledge gap[32], and where these old sources are included in analyses, they are not considered a direct contributor to river $CO_2$ release[3,8]. Also, the influence of deeper soil (millennial) and groundwater (millennial/petrogenic) inputs of $CO_2$ to river carbon emissions is assumed to be less important as river size increases[5,6,10], affecting only relatively short river reaches[5,45]. This relative decrease in terrestrial and groundwater inputs is thought to be offset by increased within-river $CO_2$ production and, potentially, riparian wetland inputs[5,6,10,46]. These conceptual models, in which within-river production offsets groundwater inputs as river size increases and/or terrestrial GPP dominates, cannot account for the sizeable contribution of old carbon to river $CO_2$ emissions evident in our analysis across biome, lithology and catchment size. As a result, current numerical models of river carbon transport and emission also fail to account for inputs from old carbon sources.

On the basis of our findings, we propose a new conceptual model of river $CO_2$ emissions that accounts for a mixture of decadal-aged and millennial-aged inputs from the biosphere and their impacts on and response to carbon cycle perturbations (Fig. 3b). There are further contributions from petrogenic sources (carbonate weathering, rock oxidation) that may or may not be vulnerable to catchment perturbations in the same way as the biosphere (Extended Data Fig. 7).

The second largest proportion (41 ± 16%) of the $CO_2$ emitted by rivers is attributed to rapid, decadal carbon cycling through ecosystems (Fig. 3b). Most of this decadal-aged proportion of river $CO_2$ is probably produced at the near surface through root respiration and/or surface litter decomposition. Some of this $CO_2$ may be used during chemical weathering of carbonate and silicate minerals to generate DIC and some carried as dissolved $CO_2$ laterally to rivers and streams. Within-river aquatic metabolism is likely to supplement these rapid-cycling river $CO_2$ emissions[10] alongside degradation of young and reactive river dissolved and particulate organic carbon[44,47]. This fraction of river $CO_2$ emissions is a loss pathway from ecosystem respiration, whose transit time for carbon is typically on the order of years to decades[37].

The largest proportion (52 ± 16%) of river $CO_2$ emissions is sourced from millennial-aged carbon on the basis of the global-scale assessment presented here (Fig. 3b). Hydrological flow paths can mobilize dissolved $CO_2$ and DIC produced by soil respiration from deeper in the soil profile. This depth may coincide with the production of $CO_2$ through root respiration, linking decadal-aged to millennial-aged $CO_2$ sources. However, inputs of older $CO_2$ from deeper in soil profiles, recently eroded or degraded soil surfaces, hyporheic zones and degradation of older river dissolved and particulate organic carbon could all contribute[13,48].

The remaining 7 ± 1% of river $CO_2$ emissions is derived from petrogenic carbon (Fig. 3b). Hydrological flow paths can also readily reach deeper into the bedrock underlying soils, supplying rivers[49], soils[26] and plants[50], and connectivity can also occur where bedrock is exposed or soil coverage is minimal. The petrogenic carbon contained within carbonate rocks and rock organic matter can thus be mobilized by chemical weathering and erosion and delivered to river systems (Fig. 3).

The last two old (millennial and petrogenic) components of river $CO_2$ may not necessarily have contributed to local ecosystem respiration. Instead, they represent a leak of older terrestrial carbon that escapes to the atmosphere through river surfaces (Extended Data Fig. 7).

## Implications for global carbon budget

River $CO_2$ emissions represent a mirror of ecosystem processes liberating DIC and $CO_2$ from organic and mineral carbon stores (Extended Data Fig. 7). Having shown that more than 50% of river $CO_2$ emissions are derived from these old organic or mineral sources, we assess their impact on relevant terrestrial carbon stores and budgets.

Decadal-aged river $CO_2$ emissions are about 1% of global terrestrial GPP (109 Pg C year⁻¹ (ref. 6)). As such, our findings suggest that

$0.9 \pm 0.3$ Pg C year$^{-1}$ may be leaving this rapid carbon loop from river surfaces. Although this is a relatively small annual flux compared with global GPP, over decadal to centennial timescales, these losses are large and mean that river $CO_2$ emissions need to be accounted for in terrestrial carbon budgets.

The millennial-aged river $CO_2$ emissions identified here are 2–3% of global soil heterotrophic respiration rates (39 to 51 Pg C year$^{-1}$ (refs. 8,51)). River $CO_2$ emissions therefore act as a loss term from older soil organic carbon reservoirs. When compared with the total stock of soil organic carbon (roughly $840 \pm 280$ Pg C (ref. 27)), river $CO_2$ loss from this store ($1.1 \pm 0.3$ Pg C year$^{-1}$; Table 1) would suggest soil carbon residence times of about 400–1,400 years at steady state. This is within the range of soil age values from bulk $^{14}$C activity[27] and could suggest that hydrological flow paths are an important carbon loss pathway from deeper soil storage[49].

Our insight into the age of river $CO_2$ emissions can be used to reassess an existing mass balance of land to ocean carbon transfers and their impact on the carbon cycle[3] (Fig. 3). With no constraint on river carbon age, this previous analysis[3] calculates that terrestrial ecosystems take up about 2.3 Pg C year$^{-1}$ of anthropogenic carbon (26% of fossil fuel emissions), but only store about 1.7 Pg C year$^{-1}$, with the remaining approximately 0.6 Pg C year$^{-1}$ released back to the atmosphere by rivers, transported to coastal oceans or stored in sediments[3]. Here we suggest that only $41 \pm 16\%$ of river $CO_2$ emissions ($0.9 \pm 0.3$ Pg C year$^{-1}$) could contain recent anthropogenic-derived carbon (decadal-aged or younger). This means that the riverine loss from this decadal carbon store is approximately half of the total flux used in this previous mass balance[3]. This budget adjustment suggests that the decadal-aged biosphere is storing more anthropogenic carbon than previously suggested[3], whereas the remaining river $CO_2$ leak is from soil and geologic carbon stores that predate widespread anthropogenic fossil fuel $CO_2$ emissions (Fig. 3b). This fundamentally changes our inference of where anthropogenic carbon resides within the main Earth system carbon reservoirs.

Whether or not anthropogenic perturbation has increased the leak of old carbon to the atmosphere through rivers that we observe here remains a notable knowledge gap. The dataset shows a trend of increasing age of $F^{14}C_{atm}$ in river DIC, $CO_2$ and $CH_4$ during the observation period (Extended Data Fig. 1). This could indicate increasing emissions of old carbon through time owing to destabilization of global soil carbon stocks[14,15,31,48,52] and changes to weathering, erosion and rock oxidation rates[22,38,53] as a result of climate and anthropogenic perturbations. Anthropogenic climate change may increase $CO_2$ supply to rivers as soils warm and/or get wetter and microbial respiration increases[54], whereas the delivery of DIC and $CO_2$ from rock weathering may also increase as landscapes warm[22,53]. However, we do not know whether the trend in Extended Data Fig. 1 is because of increased perturbation (Extended Data Fig. 7), the declining atmospheric $^{14}CO_2$ signal moving through the biosphere (Fig. 1c), sampling bias (Supplementary Information section 4) or a combination of these. Regardless, our analysis indicates that river $CO_2$ emissions are responsive to inputs from old carbon sources and could increase under direct anthropogenic disturbance regimes such as landscape drainage, clearance, burning and agricultural soil cultivation, as well as because of anthropogenic climate change.

Knowledge of how the source of river $CO_2$ emissions has changed through time is at present data-limited—we lack time series samples of river $^{14}$C-DIC and $^{14}$C-$CO_2$—and we have no way as yet to reconstruct the source of river carbon emissions in the past. These observations are crucial for improving our ability to partition and explain the drivers of this substantial global carbon flux[55]. Nevertheless, we provide evidence for a previously unrecognized, planetary-scale release of old, pre-industrial-aged carbon from land to the atmosphere through rivers. River emissions are thus vulnerable to perturbations of short-term carbon cycling (GPP), millennial soil carbon stocks and geologic carbon cycling, which can route carbon from catchments to the atmosphere through river surfaces (Extended Data Fig. 7).

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

## Methods

### Study approach and methods summary

Our aim was to analyse global-scale patterns of the age and source of river carbon emissions. Radiocarbon studies of river DIC, $CO_2$ and $CH_4$ have been conducted from ecological to geochemical perspectives and varying degrees in between. As a result, widely differing sets of variables are available from each study, making river to river comparisons difficult. Here we instead focus on a global approach to compare the $^{14}$C content of river $CO_2$ emissions and the probable carbon source components within this flux to the atmosphere.

We first assembled a database of measurements of the radiocarbon content of river DIC, $CO_2$ and $CH_4$ from the literature and then added a subset of unpublished data. Radiocarbon content is presented here as $F^{14}C$. In conventional $^{14}$C dating, $F^{14}C = 1.0$ represents 1950 CE; however, in relation to carbon cycling and the atmospheric $^{14}CO_2$ record (Fig. 1c), $F^{14}C < 1.0$ indicates carbon older than 1955 CE and $F^{14}C > 1.0$ is carbon younger than 1955 CE (Fig. 1b). The database was dominated by DIC measurements, so we demonstrated that DIC and $CO_2$ are probably in isotopic equilibrium (Supplementary Information section 1), allowing us to explore the $F^{14}C$ content of global river $CO_2$ emissions from the database.

Owing to inconsistencies in how catchment characteristics were reported in the literature from which we assembled the radiocarbon measurements (if they were reported at all), we used a global database of hydrological-environmental characteristics (HydroATLAS[56]) to extract key catchment information for each sampling location in a consistent manner (for example, catchment size, lithology, biome).

The resolution limits of HydroATLAS mean that, for catchments <10 km$^2$ in size, it was difficult to ensure that the correct catchment characteristics were extracted. Further, not all data in HydroATLAS are available at the catchment scale. For this reason, we extracted data from HydroATLAS at both the reach (1-km radius from the sampling location) and the catchment scale and use the reach characteristics for small catchments ≤10 km$^2$ and the catchment characteristics for large catchments >10 km$^2$.

We used the HydroATLAS information to explore the potential underlying drivers of the $F^{14}C$ content of river DIC, $CO_2$ and $CH_4$, first through mapping key characteristics such as catchment size, lithology and biome (Fig. 2) and then using a random forest model to explore a wide range of catchment characteristics (such as climate and soil properties). Owing to the catchment size issue identified above, we ran the model separately for catchments ≤10 km$^2$ and >10 km$^2$ to ensure that we did not incorporate anomalous catchment characteristics into the model.

To determine the potential contributions of different carbon sources, defined here as decadal, millennial and petrogenic carbon sources, we used a two-endmember isotope mixing model with Monte Carlo simulations, constrained by known inputs of petrogenic carbon to river DIC from weathering. To independently assess this approach, we also independently used an unconstrained three-endmember Bayesian isotope mixing model. We then used these estimates of the proportional contribution of different carbon sources to global river DIC to estimate the magnitude of river $CO_2$ emissions to the atmosphere derived from old carbon (millennial or older).

### New data

We include data from three unpublished works in our analysis. These include dissolved $CO_2$, $CH_4$ and DIC data from the UK, Taiwan, Cambodia and China.

Eight dissolved $CO_2$ and three dissolved $CH_4$ samples were collected for $^{14}$C analysis from a range of urban rivers and canals in London in September 2021. Paired $CO_2$ and $CH_4$ samples were collected from the River Brent, Regent's Canal and the River Thames; $CO_2$ samples were collected from Bow Creek and more sites on Regent's Canal and the River Thames (Supplementary Table 1). $CO_2$ samples were collected

using the super headspace method[57], with samples collected by equilibrating 3 l of water with 1 l of $CO_2$-free headspace for three minutes and the headspace injected into a molecular sieve cartridge for transport to the National Environmental Isotope Facility (NEIF) Radiocarbon Laboratory in East Kilbride, UK. $CH_4$ samples were collected with the coiled membrane method[21], in which water was slowly pumped through a hydrophobic, gas-permeable membrane into a headspace containing ambient air. The vessel was left to collect $CH_4$ overnight and recovered after 12–18 h; the headspace was collected into foil gas bags and transported by land to the NEIF Radiocarbon Laboratory. $CH_4$ samples were corrected for the ambient air in the headspace following refs. 21,52.

River water DIC samples from Taiwanese rivers and the Mekong River in Cambodia were collected using the methods outlined in refs. 18,58. In Taiwanese rivers, 1-l sampling bottles were submerged into the middle of the channel using a weighted Teflon sampler. On the Mekong River, near-surface samples were collected using a horizontally mounted Niskin-type sampler. River water was then filtered directly into preweighed 1-l foil bags (FlexFoil PLUS) through polyethersulfone filters (0.22 μm) using syringe-mounted filtration, with care to avoid any atmospheric air mixing. The foil bag was filled with approximately 200–500 ml of filtered river water (depending on expected DIC concentration) and then gently squeezed before closing to ensure that no air was trapped. The filled bag was reweighed and stored at 4 °C during fieldwork, before shipping to the UK, in which the sample was frozen within about a week of collection[58].

The UK, Taiwan and Cambodia samples were processed at the NEIF Radiocarbon Laboratory. $CO_2$ samples were retrieved from the molecular sieve cartridges by heating to 425 °C. $CH_4$ samples were passed through soda lime and molecular sieve filters to remove residual $CO_2$ and then combusted to $CO_2$ using platinum bead catalyst at 950 °C. For the DIC samples, orthophosphoric acid was added to the defrosted, filtered water sample in the foil bag and the degassed $CO_2$ collected, isolated and purified using cryogenic traps[58]. The $CO_2$ for $^{14}$C analysis was then cryogenically recovered and graphitized using Fe–Zn reduction and analysed for $^{14}$C content by Accelerator Mass Spectrometry (AMS) at the Scottish Universities Environmental Research Centre in East Kilbride. For quality assurance, standard materials of known $^{14}$C content were processed alongside the samples.

We collected $^{14}$C samples for DIC from 19 river sites on or draining the Qinghai–Tibet Plateau. Samples were collected in 2017, 2018 and 2023, with five sites visited in both 2018 and 2023 (MD, NQ, TNH, XD and ZMD; Supplementary Table 1). River water samples were filtered to 0.45 μm using polyethersulfone filters and collected in acid-washed (10% HCl v/v, 24 h) 1-l HDPE Nalgene bottles rinsed three times with filtered river water before collection. Samples were kept refrigerated between collections and analysis. DIC was processed to $CO_2$ for $^{14}$C analysis within 2–3 weeks of collection.

Samples collected in 2017 and 2018 were processed and analysed at the Peking University AMS facility (PKU_AMS) in Beijing, China, following ref. 59. Water samples were acidified with phosphoric acid and shaken and heated to 75 °C for 2 h to convert all DIC to $CO_2$. The $CO_2$ was then purified cryogenically on a vacuum line and graphitized using zinc reduction. Samples collected in 2023 were processed and analysed at the Beta Lab AMS facility in Miami, Florida, USA. Samples were acidified using phosphoric acid and stripped from the water by bubbling pure $N_2$ or Ar gas through the sample. The resulting $CO_2$ was collected cryogenically and graphitized using hydrogen reduction of the $CO_2$ sample over a cobalt catalyst. In both the PKU and Beta labs, reference standards, internal QA samples and backgrounds were processed alongside the samples.

### $F^{14}C$ data assembly from the literature

We initially compiled our database using the 209 DIC values available in ref. 17. We then searched for further studies of river DIC, $CO_2$ and $CH_4$ $^{14}$C values from the peer-reviewed literature. We searched and

compiled studies published before 2023 using Web of Science and Google Scholar[60] (Supplementary Fig. 7). The following string terms were used in the search: (dissolved inorganic carbon OR DIC OR carbon dioxide OR CO2 OR methane OR CH4) AND (14C OR radiocarbon) AND (stream OR river); (dissolved inorganic carbon OR DIC OR carbon dioxide OR CO2 OR methane OR CH4) AND (14C OR radiocarbon). We undertook the search several times to ensure completeness. Measurements from groundwater seeps or similar extreme endmembers were excluded, extracting only data from flowing, open water streams and rivers. We augmented these search results with our own knowledge of the literature for which studies were missed in the above searches. Ultimately, we were able to obtain 1,195 observations of fluvial DIC, $CO_2$ and $CH_4$ [14]C from 67 studies, including our own data collection outlined above.

From each study, we collected the following information when available (Supplementary Table 1):
- Site identifier (ID)
- Date and year of sample collection
- Brief site description
- Catchment name
- Compound (DIC, $CO_2$ or $CH_4$)
- DIC concentration (converted to $\mu$mol l$^{-1}$)
- $\delta^{13}$C (‰ VPDB) and associated uncertainty
- $\delta^{2}$H-$CH_4$ (‰) and associated uncertainty
- Radiocarbon publication code
- Radiocarbon content in $F^{14}$C (fraction modern) and $\Delta^{14}$C (‰) and associated uncertainty
- Radiocarbon age ([14]C years) and associated uncertainty
- Sample water pH
- Sample water temperature (°C)
- Latitude and longitude, country and hemisphere of sampling location
- Water type (river, stream and so on)
- Brief method outlines for sample collection and processing
- Exact watershed size (km$^2$)

We then provided flags for the coordinates and data uncertainties collected from the literature.

For the coordinates flags:
- Exact sampling location from the original study
- General but not exact location provided in the original study (for example, centre of catchment)
- Estimated on the basis of the map in the original study in conjunction with Google Earth Pro

For the uncertainty flags:
- Uncertainties provided in the original study
- Average uncertainties from the facility in which samples were analysed

When data were reported in $\Delta^{14}$C, we also calculated $F^{14}$C and vice versa:

$$\Delta^{14}C = 1{,}000 \times (F^{14}C \times \exp^{-\lambda(y-1950)} - 1) \tag{1}$$

$$F^{14}C = \left(\left(\frac{\Delta^{14}C}{1{,}000}\right) + 1\right) \times \exp^{\lambda(y-1950)} \tag{2}$$

in which $\lambda = 1/8{,}267$ year$^{-1}$ and $y$ is the year of sample collection.

Some locations in the database were sampled more than once. This is because of a combination of experimental approaches, for example, repeat sampling, exploration of temporal variations and method development. When a sample location was repeat sampled more than four times in a calendar year (that is, more than 0.5% of all observations), we took the average of the $F^{14}$C observations at that location for that year and recalculated a new radiocarbon age and uncertainty[61]. This removal left $n = 1{,}020$ observations (Extended Data Table 1).

## Normalization of $F^{14}$C values to atmospheric $^{14}CO_2$

We normalized the $F^{14}$C values in the database for each measurement to the $F^{14}$C-$CO_2$ content in the atmosphere in the year of sample collection, defined as $F^{14}C_{atm}$:

$$F^{14}C_{atm} = \frac{Fm_{sample}}{Fm_{atmosphere}} \tag{3}$$

in which $F^{14}C_{atm}$ is the normalized $F^{14}$C value of the sample ($Fm_{sample}$ in fraction modern) divided by the $F^{14}$C value of the atmosphere in the year of sampling ($Fm_{atmosphere}$ in fraction modern)[35].

The atmospheric $F^{14}$C-$CO_2$ values used in equation (3) were compiled from 1950 to 2023. Atmospheric $^{14}CO_2$ is from ref. 34 for 1950 to 2019. For 2020 to 2023, annual $^{14}CO_2$ was estimated by extrapolating the declining annual trend of $^{14}CO_2$ observed between 2014 and 2019. This period was chosen because the curve seemed to be flattening during this period relative to the steeper decline seen earlier in the data (Fig. 1c). Although we note that the relative contributions of contemporary biomass and soil respiration to river carbon emissions are probably not globally consistent and cannot be captured by normalizing to a single year atmospheric $^{14}CO_2$ value, this method allows a consistent normalization of the entire database irrespective of individual river catchment characteristics.

## Paired DIC–$CO_2$ $^{14}$C measurements

To explore the relationship between $F^{14}$C in DIC and $CO_2$ emissions, we compiled 15 paired $F^{14}$C measurements of DIC and $CO_2$ (Supplementary Fig. 1, Supplementary Table 3 and Supplementary Information section 2). These paired samples cover 11 distinct sites and a river pH range of 4.2–7.7, indicative of the range of pH found in natural waters. Six of these paired observations come from ref. 19, collected from two peatland headwater streams, one in north England (Moor House) and one in southern Scotland (Auchencorth Moss). Eight unpublished paired measurements were also obtained from headwater streams in the north of Scotland, four in the Flow Country and four on the Isle of Lewis. Another unpublished paired measurement was obtained from Peru, from the Manu River. Sample $^{14}$C collection and processing for the new Scotland and Peru measurements were the same as for the London, Taiwan and Cambodia samples outlined above.

## Data extraction from HydroATLAS

For each data point in the radiocarbon database, we collected information on the catchment characteristics of the sampled river. Unfortunately, this information was reported in a highly inconsistent manner and, in many cases, not at all, in the published literature. Therefore, for consistency in our analysis, we extracted catchment and hydrological characteristics from HydroATLAS[56]. HydroATLAS provides catchment and reach characteristics for rivers across the globe at 15-arcsecond resolution and includes parameters on hydrology, physical catchment settings, climate, land cover and use, soils and geology and anthropogenic influences. We extracted selected parameters at both the reach and catchment scale where possible and added these to our database (Supplementary Tables 1 and 4).

To ensure that we were extracting catchment characteristics for the correct river in HydroATLAS, we collected the details of catchment area for each sampling point from the original study; when this was not available, we estimated catchment size based on indicative values in the original study, published catchment sizes found in other studies of the same rivers or through order of magnitude estimates from visual assessment on Google Earth Pro (for example, 1 km$^2$, 10 km$^2$, 100 km$^2$, 1,000 km$^2$ and so on; Supplementary Table 1). Exact catchment sizes and combined exact/estimated catchment sizes are provided separately in the database. We compared the exact or estimated catchment size with the extracted catchment size from HydroATLAS (Supplementary Fig. 6). For most catchments greater than 10 km$^2$ in size, the values matched

well. For catchments less than 10 km², the relationship broke down because of the resolution of HydroATLAS. For further analysis, we only used reach characteristics (extracted for the nearest river reach within 1 km² of the sampling point) for sampling points with exact or estimated catchment size ≤10 km². For catchments greater than 10 km², we used the catchment characteristics. Note that, for some parameters, only catchment or reach characteristics were available (Supplementary Table 4).

From the catchment size information, we produced two sets of classifications. (1) A binary 'small' (≤10 km²) and 'large' (>10 km²) classification—this classification was chosen owing to the lower river basin size limit of the HydroATLAS (10 km²) and was based on the exact/estimated catchment size information extracted from the original studies. This binary size class was used primarily for QA/QC checks in the database and also in defining whether to use reach or catchment parameters from HydroATLAS in the random forest model (Extended Data Figs. 3 and 4). (2) A multiclass exponential classification of 0–10 km², 100 km² (10 to 100), 1,000 km² (100 to 1,000) 10,000 km² (1,000 to 10,000) 100,000 km² (10,000 to 100,000), 1,000,000 km² (>100,000)—this classification was based on binary class (1) above for the 0–10-km² class and catchment size extracted from HydroATLAS for the other classes and was used in the analysis presented here. Both size classifications were created manually and are provided in Supplementary Table 1.

From the biomes provided in HydroATLAS, we simplified these into eight classes (Supplementary Fig. 2):
1. Temperate grasslands and shrublands, which was the same as HydroATLAS biome '8. Temperate Grasslands, Savannas & Shrublands'.
2. Tropical grasslands and shrublands, which included HydroATLAS biomes '7. Tropical & Subtropical Grasslands, Savannas & Shrublands' and '9. Flooded Grasslands & Savannas' (which occur mostly in tropical regions[62]).
3. Temperate conifer and boreal forests, which included HydroATLAS biomes '5. Temperate Conifer Forests' and '6. Boreal Forests/Taiga'.
4. Tropical broadleaf and conifer forests, which included HydroATLAS biomes '1. Tropical & Subtropical Moist Broadleaf Forests', '2. Tropical & Subtropical Dry Broadleaf Forests' and '3. Tropical & Subtropical Coniferous Forests'.
5. Temperate and Mediterranean broadleaf mixed forest, which included HydroATLAS biomes '4. Temperate Broadleaf & Mixed Forests' (although no samples in the database come from this biome) and '12. Mediterranean Forests, Woodlands & Scrub'.
6. Tundra, which was the same as HydroATLAS biome '11. Tundra'.
7. Montane grass and shrubs, which was the same as HydroATLAS biome '10. Montane Grasslands & Shrublands'.
8. Deserts, which included HydroATLAS biome '13. Deserts & Xeric Shrublands' and also a further classification '15. Polar Desert' added here to include the samples in the database from the Antarctic.

From the lithology classifications provided in HydroATLAS, we simplified these into three classes (Supplementary Fig. 3):
1. Metamorphic, which was the same as HydroATLAS class '8. Metamorphic Rocks (MT)'.
2. Igneous, which included the HydroATLAS classes '2. Basic Volcanic Rocks (VB)', '4. Basic Plutonic Rocks (PB)', '7. Acid Volcanic Rocks (VA)', '9. Acid Plutonic Rocks (PA)', '10. Intermediate Volcanic Rocks (VI)', '12. Pyroclastics (PY)' and '13. Intermediate Plutonic Rocks (PI)'.
3. Sedimentary, which included the HydroATLAS classes '1. Unconsolidated Sediments (SU)', '3. Siliciclastic Sedimentary Rocks (SS)', '5. Mixed Sedimentary Rocks (SM)' and '6. Carbonate Sedimentary Rocks (SC)'.

One data point returned 'No Data (ND)' from the HydroATLAS lithology classes and was excluded from the lithology analysis. Data from Antarctica were also excluded from the analysis owing to a lack of lithology data (returning 'Ice and Glaciers (IG)' from the HydroATLAS lithology classes).

## Statistical analyses

Statistical analyses were carried out in R version 4.1.1 (ref. 63). We used nonparametric Kruskal–Wallis tests with the kruskal.test function in R, supplemented by post hoc analyses consisting of Conover–Iman tests using the conover.test function and unpaired two-sample Wilcoxon tests using the wilcox.test function. We undertook linear regression analyses using the lm function. The details of where each analysis is applied are provided in the Figures in the main text, Extended Data and Supplementary Information.

## Random forest model

We explored potential drivers of the age of river carbon emissions using a random forest model. Random forests are a machine learning model that integrate numerous regression trees to make predictions. Owing to its capacity to capture nonlinear relationships, and mitigate the risk of data overfitting, this approach has proved to be successful in numerous environmental studies for unravelling the interplay among variables[64–66]. In this study, we use random forest models to investigate the relationships between key catchment characteristics extracted from HydroATLAS and $F^{14}C_{atm}$ values in the database. We aimed to identify which variables have the strongest control on $F^{14}C_{atm}$ of river carbon emissions.

To select the input variables for the model, we first removed variables that correlated significantly with other potential input variables based on a Spearman correlation greater than 0.6 to avoid the results being influenced by correlated input variables. The remaining variables are shown in Supplementary Table 5 and includes the year of sample collection ('year').

We split the model runs by catchment size (Extended Data Figs. 3 and 4) using whole catchment characteristics for rivers with catchments greater than 10 km² and reach characteristics for rivers with catchments ≤10 km². Owing to limits on the number of data points, we only applied the model to DIC data, in which observations were $n > 100$ when separated by size.

We conducted the random forest analysis using the randomForest 4.6-14 package in R (ref. 41). Random forest models were built for the $F^{14}C_{atm}$-DIC (having removed repeat sampled locations in a calendar year) using all 19 variables from all of the 673 large catchments. We assessed the performance of the random forest model prediction by calculating the coefficient of determination ($R_d^2$) and determining the importance of each variable through the increase in the mean square error. A tenfold cross-validation was used to enhance the robustness of the results. The dataset was randomly divided into ten equal-sized samples, with 90% of the data used for training the random forest model, whereas the remaining 10% was used to assess model performance. This process was iterated ten times until each 10% sample was used and the final model performance was computed as the mean of the ten evaluation results. Following the same approach, random forest models were also built for $F^{14}C_{atm}$-DIC using all 18 variables across all of the 211 small catchments.

We assessed the association between predictor variables and $F^{14}C_{atm}$ with partial dependence plots using the pdp R package[67]. The partial dependence plots show how $F^{14}C_{atm}$ changes when a given input variable (Supplementary Table 5) varies but all other variables are held constant in the random forest model. We performed the partial dependence analysis ten times (mirroring the ten iterations of random forest models from using tenfold cross-validation) and plotted the mean values from these ten runs, with the variability across the runs indicated by the shaded area (Extended Data Figs. 3 and 4).

## Endmember isotope mixing model and Monte Carlo simulation

We used an endmember isotope mixing model and Monte Carlo approach to constrain the role of decadal versus centennial and older carbon inputs to river DIC and its contribution to river $CO_2$ emissions.

To do this, we sought to account for petrogenic inputs from carbonate mineral and rock organic matter weathering and calculate an $F^{14}C$ value for the non-petrogenic residual. This non-petrogenic residual is a combination of: (1) the DIC supplied from soil or atmospheric $CO_2$ during carbonate weathering; (2) DIC supplied by silicate mineral weathering from soil or atmospheric $CO_2$; (3) $CO_2$ derived from ecosystem respiration and delivered by water flowing through catchments; (4) $CO_2$ derived from within-river respiration of dissolved and particulate organic carbon by aquatic flora and fauna; and (5) the potential invasion of atmospheric $CO_2$ if rivers are undersaturated with respect to atmospheric concentrations.

In an ideal world, it would be possible to account for petrogenic inputs to DIC and $CO_2$ for each watershed in the database (and potentially for each sampling point). To do this, we would need to use dissolved cation ($Na^+$, $Ca^{2+}$, $Mg^{2+}$, $K^+$) and anion ($Cl^-$, $SO_4^{2-}$, Re) data to assess the weathering acids and contributions from carbonate and rock organic matter weathering[18,38,39]. Unfortunately, most of the studies reporting river DIC and $CO_2$ $F^{14}C$ measurements do not report dissolved ion data, or if they do, do not report the necessary range of cation and anion measurements to complete a weathering-source inversion. As such, we take a global view using our mean $F^{14}C$ DIC values and assess the petrogenic inputs using global estimates of carbonate and rock organic carbon weathering rates.

We can express total river DIC–$CO_2$ export as a mass balance of the known lateral and vertical fluxes (concentrations per unit area per unit time):

$$\text{Total river DIC flux} = \text{lateral DIC export to ocean} \\ + \text{vertical } CO_2 \text{ emission flux} + \text{carbonate precipitation} \quad (4)$$

in which DIC is the sum of dissolved $CO_2$, $HCO_3^-$ and $CO_3^{2-}$ (Supplementary Information section 2), we express all fluxes at the global scale in Pg C year$^{-1}$ and we assume that carbonate precipitation is negligible at the global scale[68]. Lateral DIC export from rivers to the global oceans is estimated to be $0.52 \pm 0.17$ Pg C year$^{-1}$ (ref. 42), and global vertical $CO_2$ emissions from rivers are estimated to be $2.0 \pm 0.2$ Pg C year$^{-1}$ (ref. 6), producing a total river DIC flux of $2.5 \pm 0.4$ Pg C year$^{-1}$.

We can also express global river $F^{14}C$ of DIC and $CO_2$ ($F^{14}C_{river}$) as the mass balance of the three main carbon sources defined in this study, for which the proportional contributions from all three carbon sources ($a + b + c$) sum to 1:

$$F^{14}C_{river} = a \times F^{14}C_{decadal} + b \times F^{14}C_{millennial} + c \times F^{14}C_{petro} \quad (5)$$

We can then combine these two mass balances to provide a first-order estimate of the contributions of these sources to the global river DIC flux:

$$\text{Total river DIC flux} \times F^{14}C_{river} \\ = (\text{lateral DIC export to ocean} + \text{vertical } CO_2 \text{ emissions flux}) \\ \times (a \times F^{14}C_{decadal} + b \times F^{14}C_{millennial} + c \times F^{14}C_{petro}) \quad (6)$$

To further constrain equation (6), we account for published estimates of petrogenic DIC inputs to the global river DIC flux derived from weathering of carbonate and rock organic matter. Global carbonate mineral weathering rates are relatively well constrained at an input of 0.15 Pg C year$^{-1}$ to DIC from petrogenic carbon in the $CaCO_3$ mineral[22]. If driven by carbonic acid weathering, this carbon flux is likely to be delivered by hydrological flow paths from weathering zones to streams and rivers. However, if sulfuric acid weathering is operating in landscapes, some of this petrogenic carbon may be released to the atmosphere as $CO_2$ and not enter the DIC pool[53]. This fate of carbon is not well constrained globally. Also, rock organic carbon oxidation has been estimated to contribute 0.068 Pg C year$^{-1}$ in the weathering zone[39]. Again, it is not known what proportion of this carbon enters

the DIC pool[53,69] and may contribute to the global river DIC flux. We thus considered the full range between two scenarios of petrogenic carbon inputs. First, a 0.15 Pg C year$^{-1}$ scenario, which may represent a lower bound. Second, we consider 0.218 Pg C year$^{-1}$, which is likely to be an upper bound, summing both carbonate and rock organic matter weathering (we incorporate this as $0.18 \pm 0.034$ for consistency with other fluxes and uncertainties). Incorporating this 'weathering input' flux constraint into equation (6) gives:

$$\text{Total river DIC flux} \times F^{14}C_{river} \\ = ((\text{Lateral DIC export to ocean} + \text{vertical } CO_2 \text{ emissions flux} \\ - \text{weathering inputs}) \times (a \times F^{14}C_{decadal} + b \times F^{14}C_{millennial})) \\ + (c \times F^{14}C_{petro} \times \text{weathering inputs}) \quad (7)$$

Using the mean $F^{14}C$ value for DIC, $CO_2$ and $CH_4$ across all rivers in our database of $F^{14}C_{river} = 0.919$ (Extended Data Table 1) and subtracting petrogenic C inputs (0.150–0.218 Pg C year$^{-1}$) from the sum of lateral DIC export to the ocean and vertical river $CO_2$ emissions ($2.5 \pm 0.4$ Pg C year$^{-1}$), we can simplify equation (7) to:

$$F^{14}C_{river} \times 2.5 = F^{14}C_{decadal+millennial} \times (2.28 \text{ to } 2.35) \\ + F^{14}C_{petro} \times (0.15 \text{ to } 0.218) \quad (8)$$

We can then calculate the non-petrogenic $F^{14}C$ value ($F^{14}C_{decadal+millennial}$), because the petrogenic source is assumed to contain no radiocarbon (that is, $F^{14}C = 0.0$). This provided an estimate of the $F^{14}C_{decadal+millennial} = 0.978$ to 1.007.

We then assumed that this residual non-petrogenic carbon was a mixture of a decadal-aged carbon source (using mean $\pm 1\sigma$ $F^{14}C$ content of atmospheric $CO_2$ between 1950 and 2023, $F^{14}C = 1.226 \pm 0.216$ (ref. 34)) and a millennial-aged carbon source (using the carbon-weighted mean ($\pm 1\sigma$) age of global mineral soil carbon in the upper 0–30 cm, $F^{14}C = 0.841 \pm 0.033$ (ref. 27)). The conceptual model of this carbon source partitioning, decadal and millennial (and petrogenic; see Bayesian isotope mixing model methods in Supplementary Information section 3), follows refs. 14,32. The decadal source endmember captures annual to decadal carbon cycling through biomass and soils, including the decomposition of dissolved organic carbon, which tends to have an $F^{14}C$ value indicative of annual-decadal terrestrial residence times[17]. The millennial source endmember captures carbon in soil stores of 0–30 cm depth (and deeper in some regions[27]), which includes the potential decomposition of older dissolved[14,17,48] and particulate[12] organic matter. To estimate the most probable composition and its uncertainty, we use a Monte Carlo simulation to generate 10,000 model runs, varying the petrogenic flux (0.150–0.218 Pg C year$^{-1}$) and the $F^{14}C$ values of the decadal (1.011–1.442) and millennial (0.808–0.874) inputs to equation (8). We report the mean proportional contributions of the decadal and millennial contributions $\pm 1\sigma$ of the 10,000 model runs (Extended Data Fig. 5). We then convert these to proportions of the vertical river $CO_2$ flux by first quantifying the proportional contribution of petrogenic carbon: $0.180 \pm 0.034$ Pg C year$^{-1}$ of 2.5 Pg C year$^{-1}$ (total river DIC flux) = $0.07 \pm 0.03$ (Table 1) and then subtracting the petrogenic proportion from the total to give 0.93 and multiplying this by the mean decadal and millennial contributions to give $0.41 \pm 0.16$ and $0.52 \pm 0.16$, respectively (Table 1). We then multiplied estimated vertical $CO_2$ emissions from global rivers ($2.0 \pm 0.2$ Pg C year$^{-1}$) from ref. 6 by these proportional carbon source contributions (Table 1). We note that there may be some equilibration of the DIC/$CO_2$ pool with the atmosphere during river transport and emission, which adds young carbon to the $CO_2$ pool[70], meaning that our estimates of old carbon contributions may be conservative.

We were not able to collect consistent, site-specific concentration or emission flux data alongside the $F^{14}C$ data extracted from the literature. This means that we were not able to scale the $F^{14}C$ values in

the database with local and regional emission fluxes (Supplementary Information section 4).

## Data availability

All data used in this analysis are available in the Supplementary Information and in the Zenodo repository: https://doi.org/10.5281/zenodo.14989633 (ref. 71).

## Code availability

The R script detailing the analyses presented in this manuscript can be found in the GitHub repository: https://github.com/jfdean1/GlobRiv14C-GHG.

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

**Acknowledgements** This work was primarily supported by a UK Natural Environment Research Council (NERC) grant NE/V009001/1 awarded to J.F.D., R.G.H. and C.D.E. J.F.D. received further support from a UK Research and Innovation Future Leaders Fellowship MR/V025082/1. R.G.H. was supported by an ERC Consolidator Grant (RIV-ESCAPE, 101002563). C.D.E. was also supported by the NERC BIOPOLE project NE/W004933/1. G.C. and Y.Z. were supported by a UK Research and Innovation Future Leaders Fellowship MR/V022857/1. Analyses on dissolved inorganic carbon from the Mekong River were financed by a NERC Environmental Isotope Facility Radiocarbon grant to R.G.H., E.T.T. and M.H.G. (1951.1015 and 1999.0416) and E.T.T. and R.G.H. were supported by NERC grant NE/P011659/1. D.B. had support from FORMAS (2018-01794), the Swedish Research Council (2022-03841) and ERC (METLAKE, 725546). V.G. was supported by the US National Science Foundation (OCE-1851309). R.G.M.S. was supported by the US National Science Foundation (OPP-1914081 and OCE-2333961). M.B.W. was supported by the Swedish Research Council (2021-04058). Extended Data Fig. 7 was designed by M. Kouvari from Science Graphic Design.

**Author contributions** J.F.D., R.G.H. and C.D.E. conceived the study. J.F.D. designed the database and collated data with J.B. and G.C. New samples and data were obtained by J.F.D., J.B., R.G.H., E.T.T., M.H.G. and L.Z. Y.Z. and G.C. led the geospatial analysis, including the random forest model, with J.F.D. D.B., V.G., R.G.M.S., S.E.T., J.E.V. and M.B.W. contributed to the collaborative network 'CONFLUENCE' led by J.F.D., R.G.H. and C.D.E., which supported the creation and analysis of the database underlying this work. J.F.D. and R.G.H. interpreted the data and wrote the manuscript, with intellectual input from G.C., Y.Z., J.B., M.H.G., D.B., V.G., R.G.M.S., S.E.T., E.T.T., J.E.V., M.B.W., L.Z. and C.D.E.

**Competing interests** The authors declare no competing interests.

**Additional information**
**Correspondence and requests for materials** should be addressed to Joshua F. Dean or Robert G. Hilton.

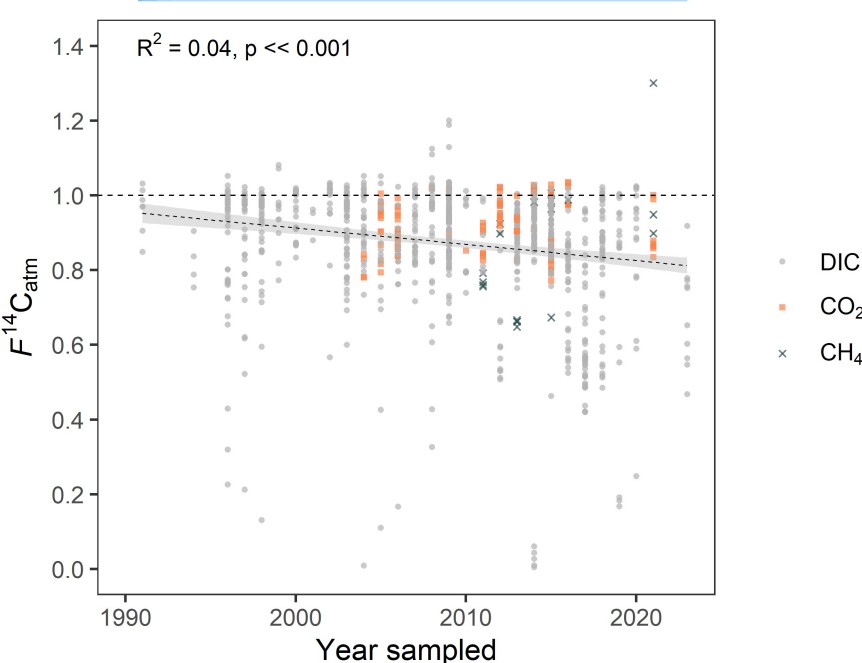

**Extended Data Fig. 1 | Temporal $F^{14}C_{atm}$ trends in river DIC, $CO_2$ and $CH_4$.** Fraction Modern values normalized to atmospheric $^{14}CO_2$ in the year of sampling[35] ($F^{14}C_{atm}$; Methods) over the available period of observations (1991–2023). The trend line, $R^2$ and $P$-values are from linear regression across all data; the dashed horizontal line indicates $F^{14}C_{atm} = 1.0$, for which $F^{14}C$ content is in equilibrium with atmospheric $^{14}CO_2$ in the year of sample collection. The density plot for the $F^{14}C_{atm}$ values is shown above the main panel.

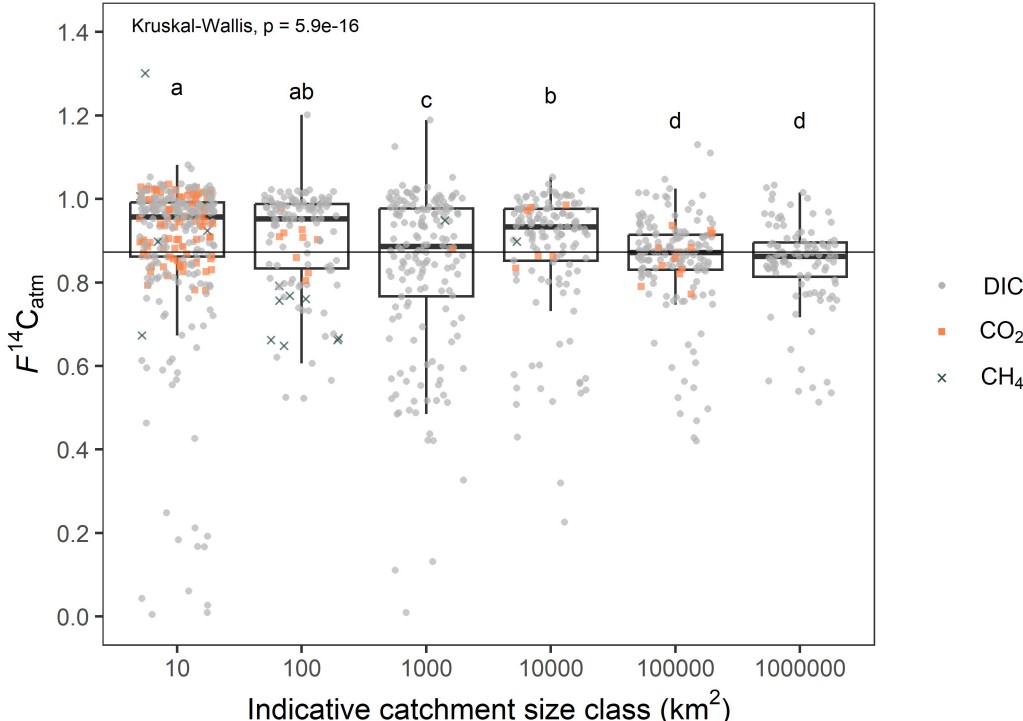

**Extended Data Fig. 2 | Influence of catchment size on $^{14}$C in river DIC, $CO_2$ and $CH_4$.** Normalized $F^{14}C_{atm}$ values for DIC, $CO_2$ and $CH_4$ separated by indicative size class on an exponential scale (that is, catchment size <10 km$^2$, approximately 100 km$^2$, approximately 1,000 km$^2$ and so on; Methods). The horizontal black line represents the mean-normalized $F^{14}C_{atm}$ for all samples, box and whisker dimensions follow Fig. 1b, lowercase letters indicate statistically significant differences ($P < 0.05$) using a Kruskal–Wallis test ($P$-value shown at the top) and Conover–Iman post hoc.

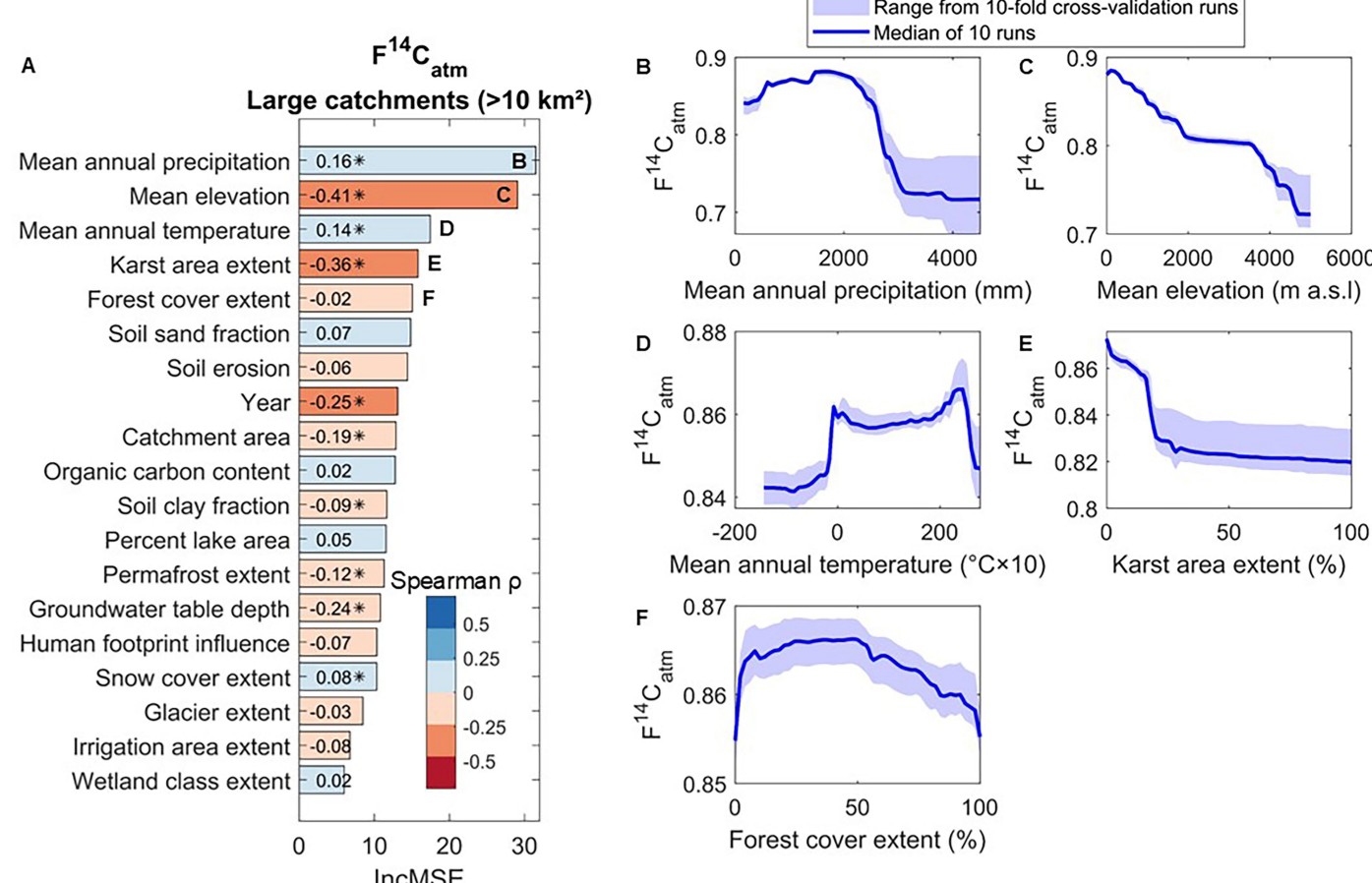

**Extended Data Fig. 3 | Potential controls on DIC $F^{14}C_{atm}$ in large catchments (>10 km²). a**, Ranking of variables by their potential importance in describing the database $F^{14}C_{atm}$ values using a random forest model; * denotes statistically significant correlations with $F^{14}C_{atm}$ ($P < 0.05$) from a Spearman's rank test calculated independently for each variable. **b–f**, Partial dependence plots showing how $F^{14}C_{atm}$ responds to the variations of a specific catchment characteristic while all other characteristics were held constant in the random forest model; plots are only shown for catchment characteristics identified by the random forest model as potentially significant controls on DIC $F^{14}C_{atm}$: mean annual precipitation (**b**), mean elevation (**c**), mean annual air temperature (in °C multiplied by 10) (**d**), the extent of karst area (**e**) and the extent of forested area within the catchment upstream of the sampling location (**f**). See Supplementary Tables 4 and 5 for a full description of variables.

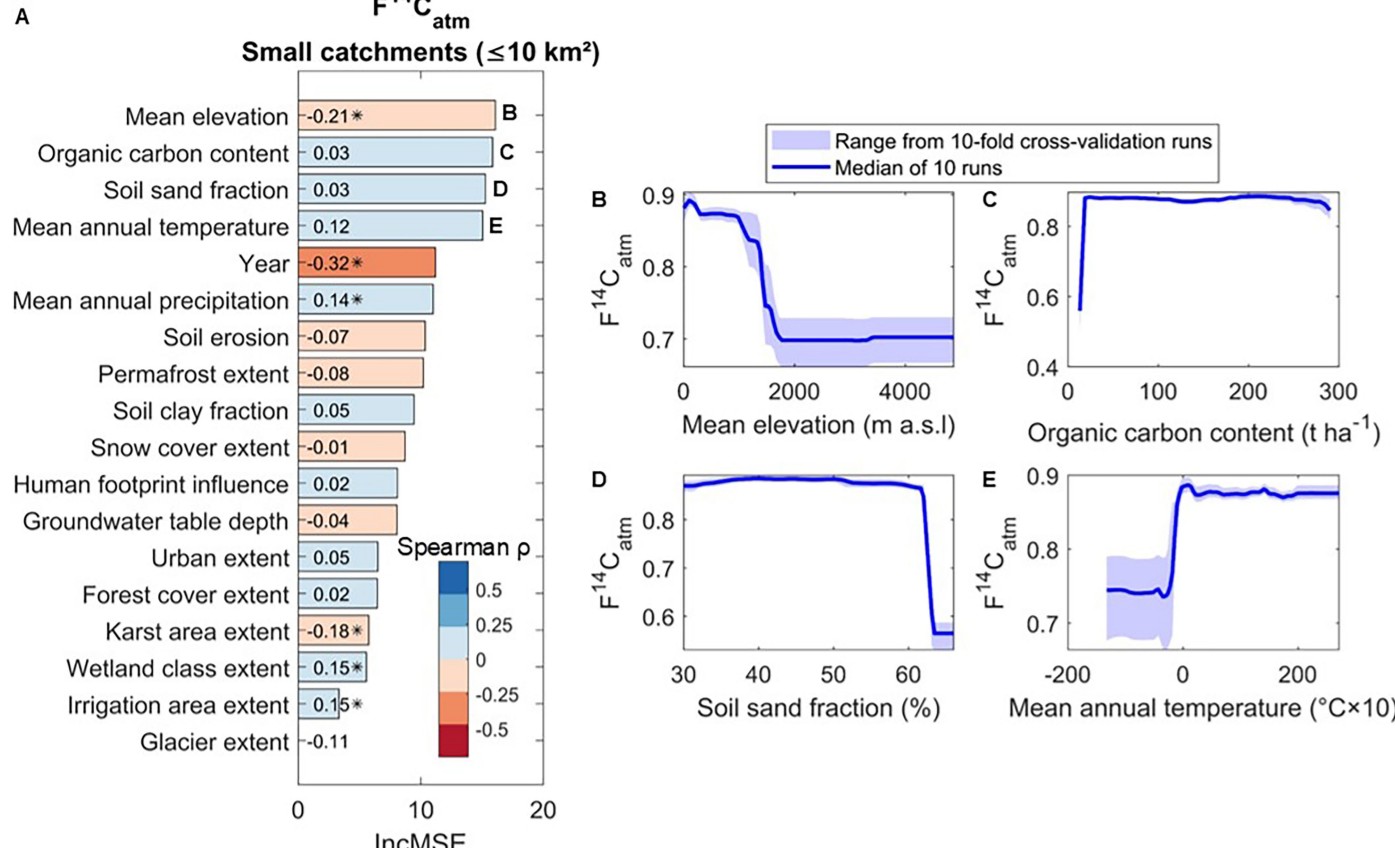

**Extended Data Fig. 4 | Potential controls on DIC $F^{14}C_{atm}$ in small catchments (≤10 km²). a**, Ranking of variables by their potential importance in describing the database $F^{14}C_{atm}$ values using a random forest model; * denotes statistically significant correlations with $F^{14}C_{atm}$ ($P < 0.05$) from a Spearman's rank test calculated independently for each variable. **b**–**e**, Partial dependence plots showing how $F^{14}C_{atm}$ responds to the variations of a specific catchment characteristic while all other characteristics were held constant in the random forest model; plots are only shown for catchment characteristics identified by the random forest model as potentially significant controls on DIC $F^{14}C_{atm}$: mean elevation (**b**), soil organic carbon content (**c**), soil sand fraction (**d**) and mean annual air temperature (in °C multiplied by 10) within the 1-km²-radius reach of the sampling location (**e**). See Supplementary Tables 4 and 5 for a full description of variables.

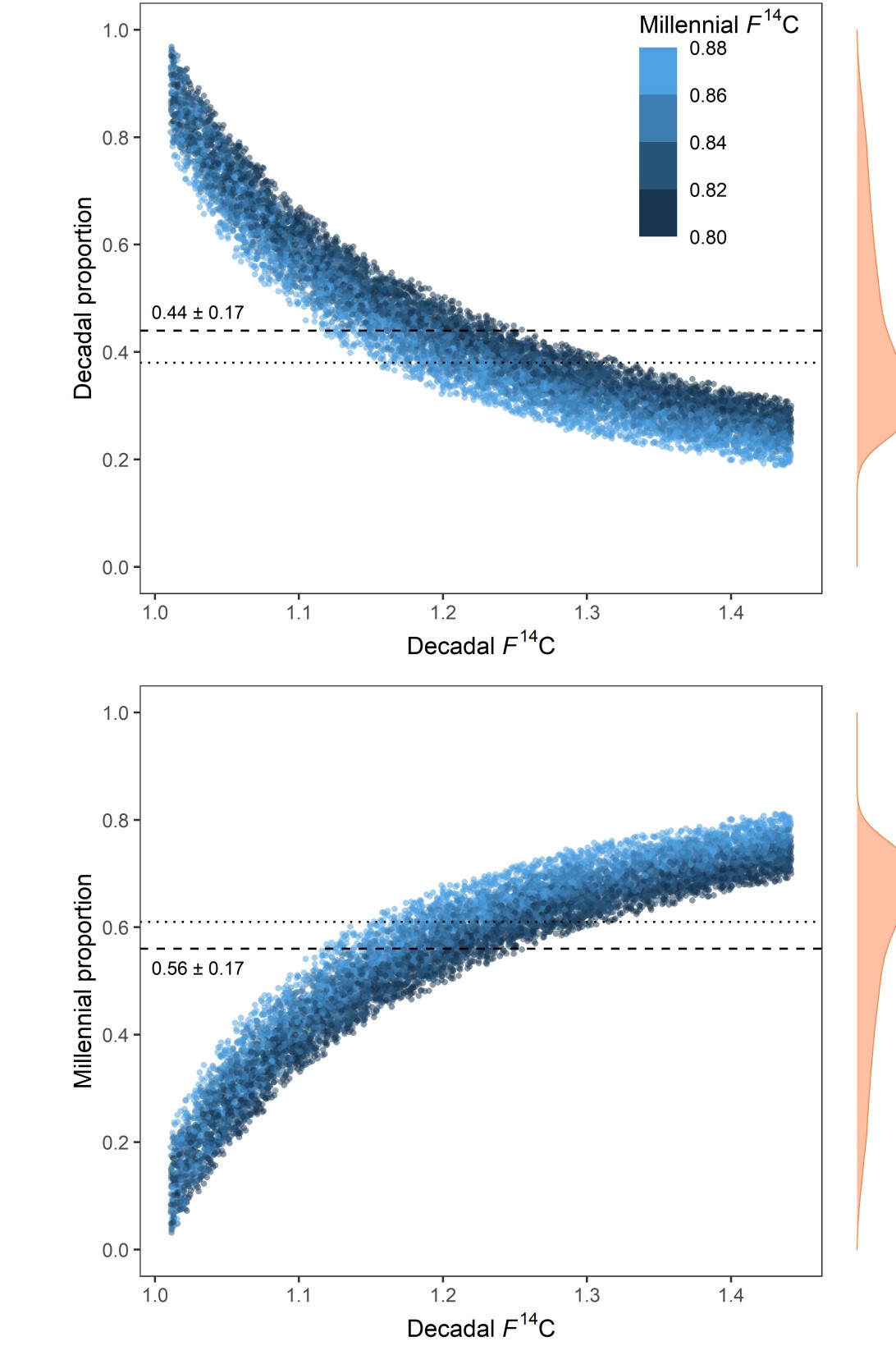

**Extended Data Fig. 5 |** See next page for caption.

**Extended Data Fig. 5 | Variable contributions of decadal and millennial carbon sources to total river DIC flux.** The variable contribution of annual-decadal biomass and soil carbon and centennial-millennial soil carbon sources to total river DIC flux (2.5 Pg C year$^{-1}$ (refs. 6,42)), assuming petrogenic contributions ranging from 0.150 to 0.218 Pg C year$^{-1}$ (refs. 22,39). After accounting for petrogenic inputs, the residual flux (2.28–2.35 Pg C year$^{-1}$, 0.978–1.007 $F^{14}$C) is modelled on the basis of two endmembers: decadal-aged carbon representing atmospheric $CO_2$ fixed in vegetation and soil carbon between 1950 and 2023 (Fig. 1c), millennial-aged carbon representing carbon in the top 30 cm of soils, globally (1,390 ± 310 years (ref. 27)). The variable contributions of these endmembers are shown relative to the decadal ($x$ axis) and millennial (dot colour) $F^{14}$C input value using Monte Carlo simulations; dotted horizontal lines indicate median contributions and dashed lines indicate mean contributions; the values represent the mean ± 1$\sigma$ proportional contribution for each carbon source. Density plots are shown to the right of the panels.

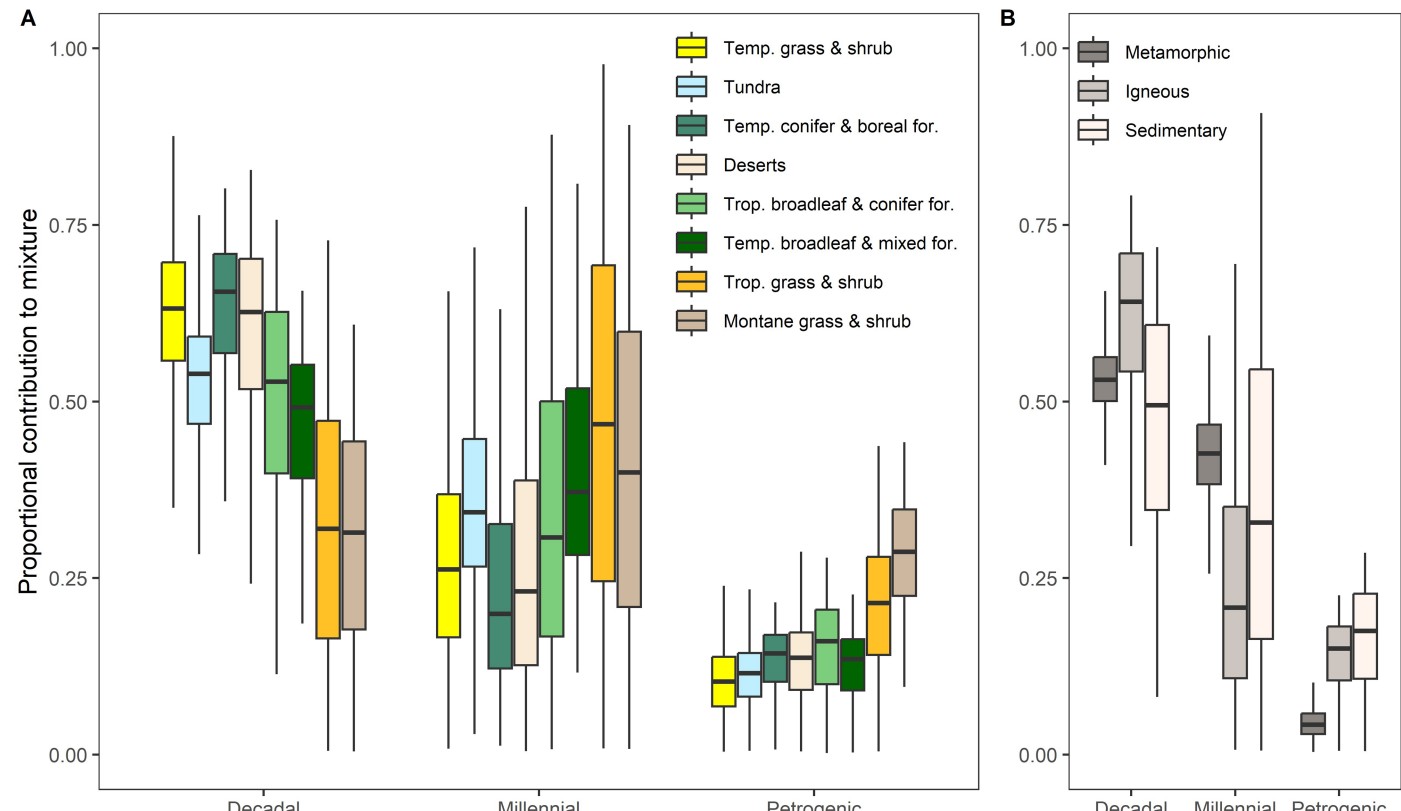

**Extended Data Fig. 6 | Bayesian isotope mixing model of carbon sources to river emissions.** The proportional contribution of different aged carbon sources to river carbon emissions using a Bayesian isotope mixing model[52,72]. We use three endmembers to define the potential carbon sources available in global river catchments: decadal-aged carbon representing atmospheric $CO_2$ fixed in vegetation and soil carbon between 1950 and 2023 (Fig. 1c), millennial-aged carbon representing carbon in the top 30 cm of soils, globally (1,390 ± 310 years (ref. 27)), and petrogenic carbon representing carbonate minerals and rock organic matter. The line in the middle of the boxes represents the median, the box limits represent the upper and lower quartiles and the whiskers extend to 1.5 times the interquartile range for the possible solutions in the mixing model. We modelled the contribution of these sources to river carbon $F^{14}C$ observations (duplicates removed) grouped by biome (**a**) and lithology (**b**), following Fig. 2.

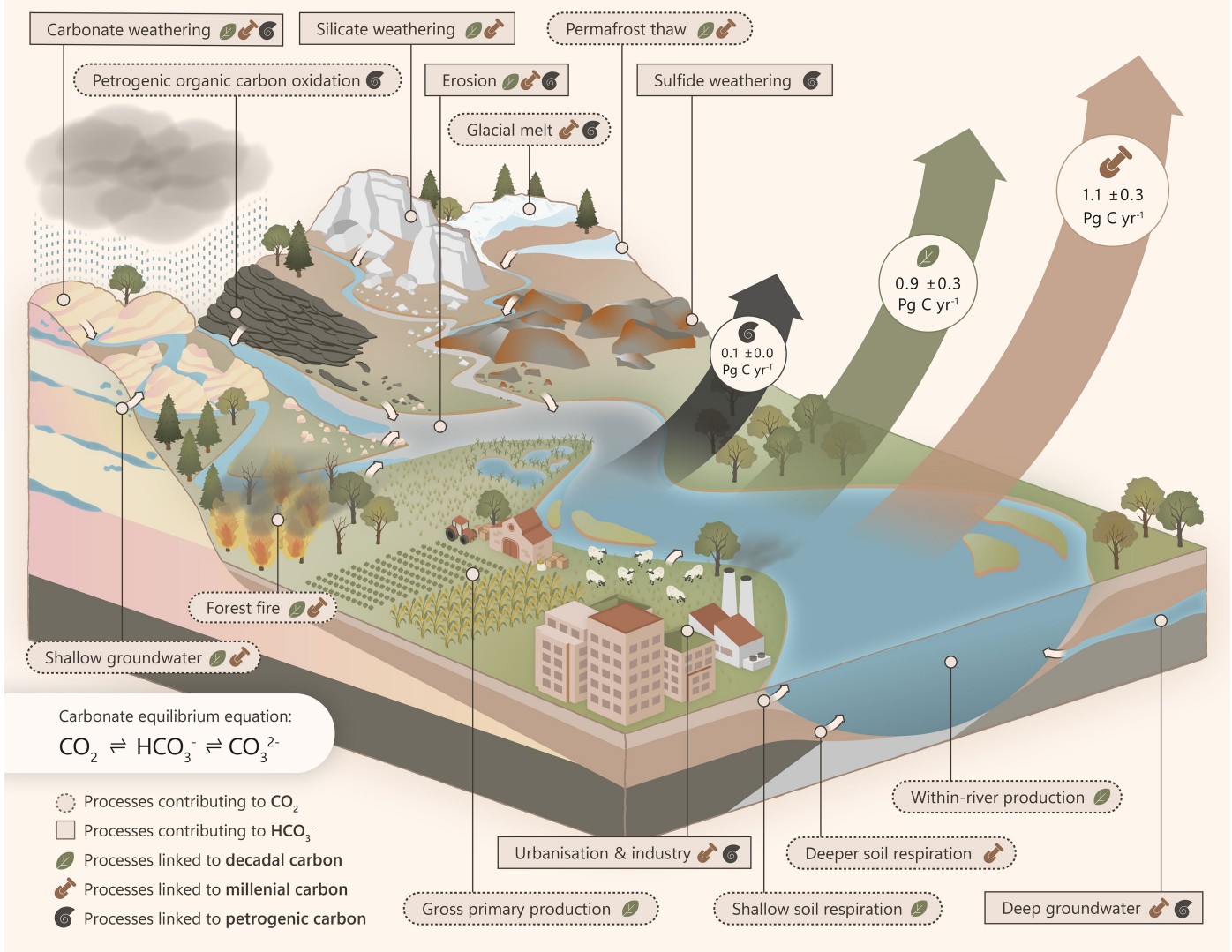

**Extended Data Fig. 7 | Conceptual model of carbon source contributions to global river CO₂ emissions.** The potential contribution of different carbon sources generated by an example range of processes within river catchments to carbonate equilibrium and subsequent CO₂ emissions (Supplementary Information Section 2). Shown here are examples of natural (for example, erosion) and anthropogenic (for example, agriculture) processes that can route carbon of different ages into rivers through direct and indirect impacts such as physical soil disturbance and climate change alterations of chemical and biological reactions. These processes, the different carbon sources (decadal, millennial and petrogenic) and their contribution to DIC or CO₂ within carbonate equilibria are indicative. Note that, in some cases, carbon is routed into rivers by groundwater, which is shown separately in the figure for illustrative purposes; the magnitude of CO₂ emissions from the different potential carbon sources are defined in Table 1; the full carbonate equilibrium is shown in equation (S2).

**Extended Data Table 1 | Summary of ¹⁴C observations in river DIC, CO₂ and CH₄**

| | | *n* obs. | *n* studies | Fraction Modern ($F^{14}C$) | | | Radiocarbon age (¹⁴C years) | | |
|---|---|---|---|---|---|---|---|---|---|
| | | | | mean | median | 1σ | mean | median | 1σ |
| **Original** | All data | 1195 | 67 | 0.921 | 0.952 | 0.168 | 661 | 395 | 1139 |
| **database** | DIC | 973 | 49 | 0.913 | 0.947 | 0.179 | 731 | 437 | 1242 |
| | $CO_2$ | 197 | 19 | 0.969 | 0.982 | 0.079 | 253 | 146 | 468 |
| | $CH_4$ | 25 | 8 | 0.836 | 0.793 | 0.165 | 1439 | 1863 | 1603 |
| **Duplicates** | All data | 1020 | 67 | 0.919 | 0.953 | 0.175 | 679 | 387 | 1188 |
| **removed** | DIC | 884 | 49 | 0.914 | 0.950 | 0.184 | 722 | 412 | 1264 |
| | $CO_2$ | 117 | 19 | 0.961 | 0.965 | 0.074 | 320 | 286 | 482 |
| | $CH_4$ | 19 | 8 | 0.879 | 0.900 | 0.167 | 1036 | 846 | 1364 |

The number of observations and studies compiled to build the database (Supplementary Table 1) and summary statistics of $F^{14}C$ values and radiocarbon age for all data and by compound.