## [Peer Review File · Nature]

Old carbon routed from land to the atmosphere by global river systems

Corresponding Author: Dr Joshua Dean

Version 0:

Reviewer comments:

Referee #1

(Remarks to the Author)

The manuscript by Dean et al. presents a first order estimate of the age of the carbon emitted as CO₂ from the global river network, quantifying contributions from fossil carbon (>55,000 years old), millennial carbon (hundreds to thousands of years old) and decadal carbon (fixed in 1955 or later). The authors estimate that about half of the CO₂ emitted from rivers stems from the fossil or millennial carbon pools, which would have important implications for our conception of the role of river CO₂ emissions in the global carbon budget. In earlier budget analysis, these CO₂ emissions are often regarded as part of the contemporary GPP, which is released through root respiration or heterotrophic respiration of biomass in soils or the rivers themselves. The authors state that their results have two important implications: 1) the amount of (semi-)terrestrial GPP that is looped back to the atmosphere through river CO₂ emissions is only about half of the global river CO₂ emission flux, and the amount of contemporarily fixed CO₂ that is stored in the terrestrial biosphere is higher than expected; and 2) a substantial amount of old carbon is brought back into the atmosphere through river CO₂ outgassing, including fossil carbon that returns from the geological carbon cycle into the biological carbon cycle.

Overall, the manuscript has the potential to become a paper that could be published in Nature, because the findings of this study are very important for our understanding of the global carbon cycle and thus of interest for the broader readership of Nature. However, I would suggest some major revisions before this paper can be considered for publication. I have some concerns regarding the methodology and the basic conclusions the authors draw. But this is something that can be fixed. Moreover, I think that the paper reads a bit too technical and would be hard to follow for the broader readership. It would be good, for instance, wherever the authors reports values of F₁₄C, to also report the radiocarbon ages, which are much easier to interpret.

On the other hand, the authors performed a number of interesting analysis, like on the statistical distribution of F₁₄C values per biome type, catchment size class, and lithological composition, which are only found in the extended data and supplementary information. Here, it would be much more interesting to move these parts into the main text, and try to explain the large-scale processes that drive difference in the age of CO₂ emitted from rivers.

Please find a few major and general comments below:

Major comments:

1) Fossil C in riverine CO₂ emissions

The authors state that HCO₃⁻ (major component of DIC) and CO₂ in river water have more or less the same radiocarbon age, because the isotopic signature of these DIC species homogenize rapidly (relative to water residence time in the river network). Thus, they assume that the measured radiocarbon age of DIC (mainly HCO₃⁻) can be used as surrogate for the radiocarbon age of the CO₂ emitted from the river. Note that the largest part of fossil DIC (highest radiocarbon age) is derived from weathered CO₃ minerals. This part of the DIC, together with younger CO₂ from the (soil-)atmosphere bound in the weathering process, does not directly drive CO₂ emissions from the river. Instead, most of this DIC is transported to the coast. It may be that through the interaction with other DIC species and the equilibration of the isotopic signal, some of that fossil carbon is emitted as CO₂ from the river to the atmosphere. However, the same amount of carbon in the DIC exported to the coast would be replaced by younger (decadal to millennial) carbon.

The authors start their quantification of the proportions of decadal, millennial, and fossil carbon in the river CO₂ emissions by subtracting the previously estimated amount of carbonate mineral dissolution (with or without an estimate of oxidation rates of fossil organic carbon) as prior estimate from the flux of river CO₂ emissions. Even if you believe that the ¹⁴C isotope of all

DIC species is completely equilibrated, it make no sense to believe that all this fossil carbon ends up in the emitted CO₂. Note that there is still a global riverine flux of about 0.3 Pg DIC/yr to the coast. This means, you also have to account for this lateral DIC export, and take the full 2.3 Pg C/yr of DIC (CO₂ emission + lateral export) as basis for the calculation. Further, if your objective is to quantify which carbon sources (as radiocarbon age groups) drive the river CO₂ emissions, then it would make more sense to assume that the fossil (or petrogenic) carbon inputs (which are mainly contributed by carbonate mineral dissolution) rather drive the DIC export to the coast. Then, it would also make more sense to simply split the river CO₂ emissions between the decadal and the millennial carbon pool. I think this way of calculation would make more sense, and it would not change the main message/result of your study. Finally, you give evidence that ¹⁴C isotopy of CO₂ and DIC is close to equilibration, however with an obvious bias towards CO₂ being younger than DIC. You report an average difference in F¹⁴C of CO₂ vs. DIC of 0.02. Couldn't you simply use this number to estimate radiocarbon age of CO₂ rather than assuming that it is equal to that of DIC? How does this difference in F¹⁴C translates to a difference in age? Also, from Figure 1B in the main text, it looks like the average F¹⁴C of CO₂ is clearly higher than that of DIC, and that by more than 0.02. Is that true? And if so, how do you explain that?

2) Reorganization of results

I think that the analyses of F¹⁴C differences between different biomes, groups of lithology and catchment size classes is quite interesting. It would be good if the extended data Figures 1A, 2 and 3 could be merged into one figure and added to the main text. It would be good to use this figure to try to explain difference in DIC ages between different types of rivers, and discuss the large-scale processes that are at play. That could be a good addition to figure 3, which is also not yet well described in the main text.

There is space to cut down other material for compensation. For instance, the authors performed a random forest model. They describe that model shortly over one paragraph in the main text, without revealing any really interesting or important results. There are no real results shown, neither in the main text nor in the extended data. I really do not see any value from that analysis and would suggest removing it from the manuscript.

General comments

L59-61: A part of the CO₂ entering rivers comes from root respiration. This is respiration of organic carbon (in the form of sugars) produced through photosynthesis, but this organic carbon is not part of the biomass. Accordingly, it would be more correct to change "terrestrial respiration of organic carbon recently fixed into terrestrial biomass" into "terrestrial respiration of organic carbon recently fixed by photosynthesis". That would include also biomass, and therefore both autotrophic and heterotrophic respiration. Moreover, you might want to change "terrestrial" into "(semi-)terrestrial" (or something similar), given the importance of wetlands as sources of C to inland waters.

L69-70: That sentence looks awkward. You mean chemical rock weathering and erosion can mobilize this carbon which is then routed to the rivers?

L70-71: Are you talking about the age of the carbon that is respired, or about when this respiration took place.

L71: "The latter, aged-carbon" That looks weird. Maybe better "Carbon at the upper end of this age range" or simply "Older carbon".

L86: For someone not familiar with the metrics you use here, it is difficult to judge whether 0.02 is large or not. Would it be possible to express that in relative terms or in terms of implied age and given in years?

L98-99: Now I am a bit confused. One paragraph before, you stated that you use DIC age as surrogate for CO₂ age. Here, the difference in average F¹⁴C of DIC and CO₂ is much larger than 0.02, and the radiocarbon age of DIC is more than twice of that of CO₂.

L115: What is the exact number (maybe relative to whole data set)?

L122-126: The differences in mean F¹⁴C per lithology group are smaller than the standard deviation of F¹⁴C values within each lithology group. Have you performed any statistical test to prove that these groups are statistically different from each other? The mean F¹⁴C value of sedimentary rocks is about 0.05 lower than that of igneous rock group. However, the mean F¹⁴C value for igneous rocks is also about 0.05 lower than that of metamorphic rocks. How would you explain these differences? It would be good if you could translate the F¹⁴C values into radiocarbon age here as well. That would make the interpretation of your findings easier.

L134-143: This description of results from your random forest model are not very informative. You name many parameters here, but you do not describe how the retained predictors actually influence the predicted F¹⁴C values. Forest cover, for instance, does it increase or decrease the F¹⁴C values?

L149-150: Do you mean "limited data availability"?

L147-156: What is the F¹⁴C (and radiocarbon ages) that you assumed for these petrogenic inputs and the river CO₂ emissions? How sure can you be that they are representative for the global scale, in particular for the global CO₂ emissions? I see from Figure 1 that your data set is geographically skewed, with only few data from extensive regions in particular in Siberia and Africa. Have you performed any weighting of the data across geographic regions, biomes, lithologies, or basin size to obtain representative average F¹⁴C values? Finally, if the F¹⁴C is 0.994-1.031 after subtracting the contributions from petrogenic C inputs (which are about one tenth of the total flux), doesn't that mean that the vast

majority of river CO₂ emissions is fueled by relatively young carbon. Anyway, it would be good to translate all the F¹⁴C values into radiocarbon ages to be able to interpret these findings more easily.

Fig. 3: That is an important conceptual figure. It would be good if it was better described in the text, with all the processes that you show here.

L733 and following, and Fig. 2 from extended material: The analysis per biome looks actually interesting. I don't understand why you do not really discuss these results in the main text.

L793-796: That is not necessary, as random forest deals naturally with multi-collinearity

L848: There is a "which" too much.

L862-865: That means you exclude CO₂ from organic soils?

Fig. S1: It looks like that is strong tendency for higher F¹⁴C of CO₂ vs DIC, implying that CO₂ is mostly younger than DIC. You mentioned an average difference of 0.02. What would be the implied average difference in age? Would it be possible to make this plot also with radiocarbon ages?

Referee #2

(Remarks to the Author)

General comments

Dean et al. present a first global estimate of the age of carbon (C) that is emitted by streams and rivers. They use radiocarbon (¹⁴C) analysis to partition the riverine CO₂ emission flux between fossil (dead C from carbonate weathering and rock organic C), millennial (old soil C), and decadal (C recently fixed by vegetation) sources. Based on a combination of data mining from the literature and new data, they show that about half of the riverine CO₂ emissions originates from C of millennial to fossil origin.

There is growing evidence from local case studies that old C can be emitted from rivers, but both a global synthesis and a conceptual framework are still lacking. While much research has focused on quantifying the export of old C from weathering and rock organic C (i.e. the fossil fraction as described by the authors), very little is known about the magnitude of older soil C export (i.e. the millennial fraction). By showing that this release of old terrestrial C to the atmosphere is significant at a global scale, this study addresses an important gap in our understanding of the linkages between terrestrial and aquatic C stores. While this is a first-order attempt at a global estimate of old C emissions, which will need to be refined as we get more data from different regions and systems, the authors' findings are timely and should be of immediate interest. Their findings have implications for global C budgets, including the possibility that anthropogenic C storage in the biosphere is more important than previously thought.

While I am overall supportive of the study and its findings, I have a few concerns regarding the methodology that I would encourage the authors to address. My first concern is related to a potential observation bias. The authors briefly mention this issue, but it would be useful to quantify it further. Could the authors determine how many of the studies in their dataset specifically target systems where we would expect old C export, e.g. thawing permafrost, glacier retreat, or peatland drainage? Do the authors' findings still hold if these studies are excluded from the database? If most studies in the dataset focus on regions with expected old C emissions, this could skew the global estimate towards higher fossil/millennial C contributions.

My second concern is related to the isotope mixing models. The Bayesian approach, while presumably more thorough than the simpler two-endmember model, seems to have an underdetermination issue. This is because the authors are assuming three different C sources, but only have data for a single tracer (¹⁴C), leading to an underdetermined system. This issue can result in uncertain and potentially biased estimates of the contributions from each C source. Did the authors assess the reliability of their estimates, e.g. did they perform any sensitivity analysis? One potential workaround to the underdetermination issue would be to constrain the model with a strong prior on the fossil C contribution, but this does not seem to have been done.

On another note, I am wondering whether using corrected F¹⁴C_{atm} values instead of raw F¹⁴C values in the mixing models might help reduce some of the uncertainties related to the changes in ¹⁴C atmospheric input over time.

My third comment is about the upscaling approach. While I understand the purpose of developing a first-order, global estimate, I think introducing one additional step in the analysis could be beneficial. For instance, could the authors provide lithology-specific percent contributions for each age range (decadal, millennial, fossil; i.e. like they did through the Bayesian model), and then scale the relative emissions from each lithology? This could potentially be done by overlapping the river network and lithology maps from HydroATLAS. This additional step would better highlight how geology affects riverine CO₂ emissions and improve the accuracy of the authors' upscaled global flux.

Regarding the implications of the authors' upscaling for the global C budget, could the authors try and revisit the budget presented in Regnier et al. (2022)? A revised budget (with for instance an increased terrestrial storage of anthropogenic C) could be a useful addition to Figure 3, which is very conceptual at the moment.

The reporting of the methodology is detailed and transparent. Statistical tests are appropriate and uncertainties are clearly

presented. One exception might be Extended Data Figure 5, where it is unclear to me what the error bars represent. The manuscript also references previous literature appropriately. I have listed more specific comments below.

Specific comments

Abstract is clear, but some improvements could be made. "Centennial" should be replaced with "millennial" for consistency with the rest of the paper. The last sentence is not entirely clear to me, please try to link it better to the findings.

Fluxes are reported in different units throughout the manuscript, e.g. Tg CH₄ yr⁻¹, Pg C-CO₂ yr⁻¹. I suggest using consistent units throughout, e.g. Pg C yr⁻¹.

L88. I found the justification of using 14C-DIC as a proxy for 14C-CO₂ well supported. I wonder if one of the paired DIC/CO₂ measurements was conducted in an area underlain by carbonate geology? It would be interesting to see how DIC and CO₂ ages compare in rivers with potentially old C contributions and high pH.

L128-132. Can this be tested? Have these predictors (e.g. peatland extent, permafrost extent) been included in the random forest model?

L142. The use of "terrestrial GPP" here is a bit confusing. I would suggest replacing with "recently fixed carbon" or similar.

Figures. Please consider combining Figures 1 and 2 (in Figure 1, either B or C should be removed), and have Extended Data Figure 5 in the main body of the paper – that is, if and when the underdetermination issue is sorted.

Some elements of Supplementary Discussion S3 should appear in the Methods.

Referee #3

(Remarks to the Author)

Summary: Dean et al. assesses the age and source of carbon emissions from global rivers, focusing on Dissolved Inorganic Carbon (DIC), carbon dioxide (CO₂) and methane (CH₄). They used Fraction Modern F₁₄C as a surrogate of carbon isotope to quantify carbon age, and compiled a comprehensive global database of F₁₄C for DIC and CO₂ in global rivers (~1,200 data points). The study concludes that a significant portion (47-56%) of global river CO₂ emissions is derived from old carbon (centennial or older). If this is true, this previously unrecognized flux of old carbon from land to the atmosphere is not accounted for in global carbon models. The results could have enormous implications for understanding and quantifying carbon cycling. The topic is of interest to the carbon community and to those who are interested in carbon – climate feedback.

I enjoyed reading the introduction. The terms (different carbon ages) are well defined. The importance of carbon emission are well contextualized in the bigger carbon cycles. I am not an expert in carbon isotopes but I like the idea of using F₁₄C as a surrogate for the isotope composition of river CO₂ emissions, especially when this is supported by additional data that show that paired DIC and CO₂ F₁₄C values are within 0.02 of one another. This work compiled a global database of radiocarbon content F₁₄C in river CO₂ and DIC, some are new data whereas most of the data are from published work. Random forest models were used to identify the influential factors of F₁₄C. An independent Bayesian isotope mixing models are also carried out, which leads to the same conclusion.

Originality and Significance: The study is novel as it quantifies carbon age in riverine emissions at the global scale, an underexplored aspect of the global carbon cycle. By compiling a large database of radiocarbon measurements across the globe, it provides a new perspective on the importance of old carbon flux in rivers and their vulnerability to climate change. To me the major finding is that there are much more old carbon in riverine carbon emission, which expand the current understanding of riverine carbon dynamics and challenge traditional carbon models that largely overlook old carbon emissions.

The idea of aged carbon fluxes into rivers and streams are not new, as shown in some of the relevant papers the authors cited (Barnes et al., 2018, <https://pubs.acs.org/doi/full/10.1021/acs.est.7b04717> ; Butman et al., 2015, <https://www.nature.com/articles/ngeo2322>). Existing work has mostly looked into aged carbon (delta14C-DOC) in dissolved organic carbon (DOC) and estimated about 3%-9% as coming from aged carbon in DOC, which is much smaller than the reported 47-56% reported here. The idea of old carbon also collaborates with recent literature on deep respiration, for example, Tune et al., 2020, <https://agupubs.onlinelibrary.wiley.com/doi/full/10.1029/2020JG005795> .

Methodology. I am not sure if I can fully believe the numbers reported, such as 47-56% from old carbon. And I am talking as a modeler and someone whose expertise is in crunching numbers. My major struggle is that I do not fully understand how these important numbers were calculated. There are some text in line 169-174, and the paper refers to the Methods section and Extended Figure 4. The methods include equation 4, which I am still puzzled after reading multiple times. In order to calculate the total fraction of old carbon in DIC globally, I expect the concentrations of DIC and CO₂ are important. But I don't see that in the equation. Are DIC and CO₂ concentrations not important at all? What is the reasoning behind this? I understand that the approach is using a "global" perspective and use the global riverine carbon emission to back calculate how much is from old carbon. To me this seems like a circular argument. Wouldn't the numbers differ substantially in different places with different old carbon content and DIC concentrations?

Monte Carlo simulation was also used to “generate 10,000 model runs, varying the fossil flux (0.15-0.218 Pg C y⁻¹), and the F¹⁴C values of the decadal (1.011-1.442) and millennial (0.808-0.874) inputs to Eq. 4.” Again, there are no concentrations involved.

Even if I understand the calculation clearly, the reported numbers would have gigantic uncertainties. We are talking about 1,200 data points sparsely distributed across the globe, with no indication of when they were measured. The F¹⁴C values could differ across different times of the year significantly. For example, we generally know that at low river flow and dry time, there are much high proportions of water coming from older groundwater compared to conditions at high river flow where typically there are much more water coming from relatively shallow zone (e.g., <https://agupubs.onlinelibrary.wiley.com/doi/full/10.1029/2010WR009341>). Stream flow during dry time probably has much more old carbon compared to wet times of the year. There are often orders of magnitude difference in river flow in different times of the year, which could lead to substantial differences in export of carbon at different ages (e.g., Duvert et al., 2016, <https://www.nature.com/articles/s41561-018-0245-y>; Duvert et al., 2019, <https://agupubs.onlinelibrary.wiley.com/doi/full/10.1029/2018JG004912>; Johnson et al., 2008, <https://agupubs.onlinelibrary.wiley.com/doi/full/10.1029/2008GL034619>).

So in generally I am not very confident about these numbers. As the paper later discussed, whether the old carbon contributes 47-56% or some other numbers like 20-30% could have enormous implications on how we think about carbon cycling, and could completely change how we think about modern C cycle. This makes it even more important to get the right numbers.

In general, there are many numbers reported throughout the paper and in Methods section. In Methods, the purpose of each section is often not stated clearly. For example, when I first read the section on “data extraction from HydroATLAS”, I had no idea what is this for, until I read to the section on “random forest model”, and realize that HydroATLAS is to extract catchment characteristics. but I think someone who does not do random forest model might still at loss. The Methods section generally lacks streamlining and organization, which makes it challenging to follow the quantification process, and to assess the validity and accuracy of these numbers.

Additional comments:

1. The abstract stated in abstract that “We show that river CO₂ and CH₄ emissions are vulnerable to anthropogenic perturbations of both short-term and long-term carbon cycles, which have the potential to increase this substantial routing of carbon from land to the atmosphere.” This statement is more a speculation than a conclusion. It is not substantiated by data and figures. Some line 272-282 documents this calculation but this is purely depends on the number 47-56% that I am concerned about.
2. The main text only has three figures. Why not put some of the extended / SI figures in main text? For example, some of the random forest figures indicating influential factors. The authors plotted a lot of bar figures for different factors (drainage area, biome, ...). It is however not clear though which factors are the most influential factors. Would it be more effective to plot a figure ranking the most influential factors?
3. Line 121-143, this part talks a lot about extended figures and supporting SI figures. It would make it much easier for the readers if the authors can combine these figures and move them to the main text
4. Line 133-143, what is the reasoning behind dividing the catchments into small and bigger ones? Do the authors expect different drivers? If so, state explicitly. The approach makes the sample size smaller and harder for a random forest model to figure out the predominant influential factors. Do you expect different ranking for the importance of the feature? Can you provide feature importance ranking for small and large rivers? This could help understand what drivers of old carbon dominate in different category. Right now, the paper only provide a list, not a ranking figure.
5. Fig. 3: the figure can be modified to reflect how this conclusion from this work change the conceptual model compared to the existing conceptual model. For example, if the numbers reported here are true, this figure can be something like a “before” and “after” figure, with different arrow size to reflect the larger old carbon flux compared to the existing conceptual model. The current figure highlights various processes contributing to carbon of different age, which is not the major finding of this work and are barely discussed / supported in the paper.
6. Line 810 – 811: 90% training and 10% testing; this seems larger percentage of training compared to the typical 75%-25% division, although it is probably ok.
7. The figures with error bars do not include the definition of error bars. This includes Fig. 1 and many extended data figures.

Version 1:

Reviewer comments:

Referee #1

(Remarks to the Author)

I feel the authors' revisions were responsive to my main concerns and they were able to substantially improve their manuscript. I only have those two very minor comments listed below. Overall, I think this study is a very valuable contribution to the scientific literature on river C cycling. I suggest its publication after these minor revisions.

L41 : Maybe you rather mean « net ecosystem exchange » (NEE, = exchange of CO₂ with atmosphere) rather than “net ecosystem production” (NEP) . NEP is defined as balance between gross primary production and ecosystem respiration. Global values of NEP are much higher than that of NEE, because NEP does not account for the recycling of human

appropriated biomass (harvested wood, crop, and fodder plants for livestock), but those are included in NEE. Global NEE is ~2 Pg C/yr. Global NEP is estimated somewhere between 5 and 10 Pg C/yr.

Extended data figure 3 and 4: For the plots on the right hand side, I wonder if it wasn't better to have partial dependence plots, as those better describe the behavior of the fitted RF model in response to predictor values. At least you could add the corresponding partial dependence plots to the SI. I think these plots are very helpful for interpreting the RF model.

Referee #2

(Remarks to the Author)

The revised paper by Dean et al. maintains the core message of the initial version but presents it with greater clarity and stronger justification. I thank the authors for their well articulated replies to my comments as well as to those from other reviewers. I support the addition of the downstream DIC flux into their mass balance (as per R1's suggestions), which I had overlooked in my initial review.

I appreciate the authors' efforts to examine the potential for sampling bias in their dataset. This additional analysis adds confidence to their results. I also independently checked the data in Table S1 and searched for occurrences of 'peatland' or 'permafrost' or 'glacier' in titles. This returned <200 occurrences out of 1,195 data points, which suggests that sampling bias towards systems where we expect old carbon should be minimal.

I am still not entirely convinced about the use of a 3-endmember Bayesian analysis, as a single tracer may not be enough to resolve 3 sources. The fact that the authors obtained realistic results might be because they used very narrow ranges of F14C values for the 3 sources, which might have helped constrain the potential solutions – but the extent of this influence should be considered. The paper by Stock et al. (<https://peerj.com/articles/5096/>) has an interesting discussion about underdetermined systems and the way Bayesian models handle them.

I understand that the Bayesian approach is not central to the findings, and I do not wish to delay the publication of this important paper – but I would invite the authors to at least mention in the Methods the potential limitations of using this approach. Alternatively, please consider running a sensitivity analysis of the impact of using broader ranges for the 3 sources. If results change very substantially when using slightly broader source values, this would indicate a high sensitivity of the model to assumptions – which should be acknowledged. The authors could also consider adding $\delta^{13}\text{C}$ as a second tracer. I would understand if they were reluctant to do so though, because $\delta^{13}\text{C}$ is not conservative and in-stream processes may complicate the $\delta^{13}\text{C}$ signal, so adding it might introduce noise rather than help resolve source contributions – but might be worth exploring.

Congratulations on this important contribution.

Clément Duvert

Referee #3

(Remarks to the Author)

The manuscript has been thoroughly revised in response to comments from three reviewers. The statistical tests are appropriate, and the error bars are accurately reported. The additional explanation in the methods section has also improved the clarity of the approach. That said, I have a few points I would like the authors to further address:

1. The paper does not clearly define the physical meaning of F14C. Line 85 describes it as a surrogate for the isotopic composition of river CO₂ emissions and references source 34, but it does not explain how it is calculated or what it represents physically. Readers need to consult reference 34 to understand its meaning, which might be difficult for a broader audience. Providing a clear and accessible explanation would improve understanding. Additionally, the term "Fraction" may be misleading, as F14C is not a fraction in the conventional sense, such as a percentage, and thus requires clarification.
2. In particular, Item 1 is relevant to the later global scale calculations. I don't think Equation (5) is correct. The F14C values in Equation (5) represent the fraction of carbon originating from decadal, millennial, or petro sources, which differ from the F14C in Equation (2). If the F14C in Equation (2) is being used, then Equation (5) does not represent a mass balance equation. For instance, if $F_{14C_decadal} = 1.1$, $F_{14C_millennial} = 0.9$, and $F_{14C_petro} = 0.6$, this would result in $F_{14C_river} = 2.6$, which exceeds the maximum values in your Fig 1C and is not meaningful. The global mean F_{14C_river} is around 0.9 - 1.0 (line 104 and 105). If you are referring to the fractions of carbon from different age sources, distinct symbols should be used instead of F14C.
3. The approach is really to figure out within the overall total river DIC flux (equation 4), what is the fraction of C in the three age categories. But then $F_{14_petro} = 0$ (line 985-986). Do you mean the fraction of the petro C is zero, or the F14C in equation (2) relating to age is equal to zero. In either case, what is the assumption behind $F_{14_petro} = 0$?

Detailed comments:

L32: usually we say "surface waters", not "water surface"

L127 – 132: could this mean larger rivers have high proportion of water coming from deep flow paths with more older carbon?

L165-168: that means warmer and more humid places flush more recent C to rivers and streams, which makes sense. Does that also mean more arid places tend to store C and contribute more to

older carbon?

L168: what is high "IncMSE"?

L246 – 247, "we propose ..." I don't think this statement reflect the recent advances on our understanding of how water flows the landscape and carbon dissolved carbon transport to streams and rivers. a lot of carbon can come from depth deeper than soils. At least I would not say "... majority .. is produced from near surface or ... surface litter". Most precipitated waters infiltrates and flows through subsurface of different depths. Water flowing through surface / top soils is typically very small.

See, for example,

Liu et al., 2024: <https://www.nature.com/articles/s41561-024-01483-5>

Stewart et al., 2024: <https://agupubs.onlinelibrary.wiley.com/doi/10.1029/2023WR035940>

McCormick et al., 2021: <https://www.nature.com/articles/s41586-021-03761-3>

Tune et al., 2020: <https://agupubs.onlinelibrary.wiley.com/doi/full/10.1029/2020JG005795>

23/01/2025

Dear Editor,

Thank you for your assessment of our manuscript at *Nature*. We present here our response to the reviewer comments and revised manuscript. The reviewer comments were very helpful in strengthening our manuscript, and we hope you will consider our revised study for publication.

Below we reply point by point to the reviewers in blue text, and the line numbers indicated refer to the clean, revised manuscript submission. We have also numbered key reviewer comments [R1-1] to streamline how we refer to different responses to the common themes in the Reviewer comments.

In summary here, the main changes to the manuscript in response to the reviews are:

- We now clearly demonstrate that the sample set is globally representative (Reviewer 2 and 3) in terms of its spatial coverage of samples across terrestrial biomes and lithology. This is done by expanded discussion and a new supplementary Fig. S4. The analysis shows that, if anything, the database is slightly biased towards “younger” systems that drain igneous-type bedrock, suggesting our estimates of old carbon inputs may indeed be conservative.
- We have added a new schematic figure to better summarise the key outcomes of our study. The new Fig. 3 shows the shift in paradigm from river CO₂ being dominated by very young carbon inputs (largely driven by the seminal paper in 2005 on the Amazon River, ref¹¹) mixing with geological weathering inputs (Fig. 3A), to our new view which stems from the $F^{14}\text{C}$ measurements. The river DIC $F^{14}\text{C}$ values require an additional old (^{14}C -depleted) input (Fig. 3B). We summarise the carbon cycle impacts that are discussed in the main text in this figure, too.
- We have included the riverine DIC flux to the ocean in our assessment of the total DIC inventory of river waters (the sum of CO₂ lost and DIC and CO₂ carried to the oceans), following this omission being pointed out by Reviewer 1. This was an important revision and we have rerun our calculations. It has served to further bolster the need for a ^{14}C -depleted input to river DIC and CO₂.
- We have updated how we report and discuss the quantitative outputs from Monte Carlo simulations and the Bayesian isotope mixing model, which provides a more robust reporting of our findings.
- We have expanded our discussion of the potential controls and drivers of the $F^{14}\text{C}$ variability. This had been done in the previous version (via the Random Forest model analysis), but the reviewer comments encouraged us to bring more of this discussion into the main text and modify Fig. 2 to highlight better the outputs.

Many thanks,
Joshua Dean & Bob Hilton
On behalf of all co-authors.

31st October 2024

Dear Dr Dean

Your manuscript, "Old carbon routed from land to the atmosphere by global river systems", has now been seen by three referees, whose comments are attached below. While they find your work of potential interest, they have raised a number of concerns that first need to be addressed before we can consider the paper further for possible publication in Nature.

In view of the additional work that seems required to address these concerns, we appreciate that the necessary revisions may take some time. But let me assure you that we will nevertheless be happy to look at a revised manuscript (unless, of course, something similar has by then been accepted at Nature or appeared elsewhere).

- Many thanks again for soliciting the reviews – we were encouraged by the positive comments. We have addressed the comments point by point below, and above provide a summary of the key changes, which directly tackle the concerns raised by the reviewers.

As a signatory of the Enabling FAIR Data in Earth, space and environmental sciences, Nature is committed to supporting FAIR principles in data sharing and citation. Where community repositories are available, we require data sharing through such repositories for papers in the Earth, space and environmental sciences for papers published in Nature. Where such repositories are not available, large datasets may be hosted in generalised data repositories such as Figshare, Dryad or Zenodo. See our FAIR data in Earth science editorial for more details.

- We will use Zenodo to archive the Github repository of the manuscript analyses for the final version of the manuscript. We have included all the data and code used in this analysis as supplementary materials alongside this revised submission.

Nature is committed to improving transparency in authorship. As part of our efforts in this direction, we are now requesting that all authors identified as 'corresponding author' create and link their Open Researcher and Contributor Identifier (ORCID) with their account on the Manuscript Tracking System prior to acceptance. ORCID helps the scientific community achieve unambiguous attribution of all scholarly contributions. You can create and link your ORCID from the home page of the Manuscript Tracking System by clicking on 'Modify my Springer Nature account' and following the instructions in the link below. If you experience problems in linking your ORCID, please contact the Platform Support Helpdesk.

Referee #1 (Remarks to the Author):

The manuscript by Dean et al. presents a first order estimate of the age of the carbon emitted as CO₂ from the global river network, quantifying contributions from fossil carbon (>55,000 years old), millennial carbon (hundreds to thousands of years old) and decadal carbon (fixed in 1955 or later). The authors estimate that about half of the CO₂ emitted from rivers stems from the fossil or millennial carbon pools, which would have important implications for our conception of the role of river CO₂ emissions in the global carbon budget. In earlier budget analysis, these CO₂ emissions are often regarded as part of the contemporary GPP, which is released through root respiration or heterotrophic respiration of biomass in soils or the rivers themselves. The authors state that their results have two important implications: 1) the amount of (semi-) terrestrial GPP that is looped back to the atmosphere through river CO₂ emissions is only about half of the global river CO₂ emission flux, and the amount of contemporarily fixed CO₂ that is stored in the terrestrial biosphere is higher than expected; and 2) a substantial amount of old carbon is brought back into the atmosphere through river CO₂ outgassing, including fossil carbon that returns from the geological carbon cycle into the biological carbon cycle.

Overall, the manuscript has the potential to become a paper that could be published in Nature, because the findings of this study are very important for our understanding of the global carbon cycle and thus of interest for the broader readership of Nature. However, I would suggest some major revisions before this paper can be considered for publication. I have some concerns regarding the methodology and the basic conclusions the authors draw. But this is something that can be fixed.

Thank you for your positive comments on our work and the detailed and thoughtful review which has allowed us to improve the paper. With regards to the method aspects raised, we reply in detail below. In summary, we thank the reviewer for spotting that we had neglected the residual DIC flux to the oceans in our radiocarbon mass balance assessment. By including river DIC alongside the river CO₂ evasion fluxes we find a relatively minor change to the outputs of our analysis, but one that strengthens the case that a source of ¹⁴C-depletion is needed beyond petrogenic inputs. Note that for consistency, we now use the term “petrogenic” instead of “fossil” throughout to avoid ambiguity.

Moreover, I think that the paper reads a bit too technical and would be hard to follow for the broader readership. It would be good, for instance, wherever the authors reports values of F₁₄C, to also report the radiocarbon ages, which are much easier to interpret.

We have edited the paper with this comment in mind. We note that all three reviewers were able to clearly articulate our main findings, suggesting that the previous version could be followed well, but hopefully our revisions have improved on this.

With regards to the reporting of radiocarbon data, we make the case below that we prefer to keep the Fraction Modern ($F^{14}\text{C}$) terminology. We agree that the radiocarbon age metric is perhaps quicker to see if it is “old” or “young” – as such, we have added it to a number of the plot axis and figure captions for the readers, please see the revised Fig. 1B and Fig. 2C. However, we do not use it extensively in the text because the ¹⁴C-years notation has two main drawbacks in our case: i) by reporting an ¹⁴C age, many readers could confuse this with an absolute date, which it is not because this cannot be easily quantified due to the carbon

coming from a mixture of sources – we want to avoid that confusion; ii) when $F^{14}\text{C}$ values are above 1 (i.e., $F^{14}\text{C}$ is higher than the 1950 atmospheric $^{14}\text{CO}_2$ value the age notation is normalised to), which is the case for many samples in the database, we cannot report a ^{14}C age – it is simply “modern”. For these reasons, we have retained the $F^{14}\text{C}$ notation for consistency throughout.

On the other hand, the authors performed a number of interesting analysis, like on the statistical distribution of $F^{14}\text{C}$ values per biome type, catchment size class, and lithological composition, which are only found in the extended data and supplementary information. Here, it would be much more interesting to move these parts into the main text, and try to explain the large-scale processes that drive difference in the age of CO_2 emitted from rivers.

We are glad these analyses were appreciated. We do already discuss the key parts of this analysis in the main text. However, we agree that this could be better represented in the figures, and have now added a combination of these figures as the revised Fig. 2, in line with comments from Reviewers 2 and 3.

Please find a few major and general comments below:

Major comments:

[R1-1]

1) Fossil C in riverine CO_2 emissions

The authors state that HCO_3^- (major component of DIC) and CO_2 in river water have more or less the same radiocarbon age, because the isotopic signature of these DIC species homogenize rapidly (relative to water residence time in the river network). Thus, they assume that the measured radiocarbon age of DIC (mainly HCO_3^-) can be used as surrogate for the radiocarbon age of the CO_2 emitted from the river. Note that the largest part of fossil DIC (highest radiocarbon age) is derived from weathered CO_3 minerals. This part of the DIC, together with younger CO_2 from the (soil-)atmosphere bound in the weathering process, does not directly drive CO_2 emissions from the river. Instead, most of this DIC is transported to the coast. It may be that through the interaction with other DIC species and the equilibration of the isotopic signal, some of that fossil carbon is emitted as CO_2 from the river to the atmosphere. However, the same amount of carbon in the DIC exported to the coast would be replaced by younger (decadal to millennial) carbon.

The authors start their quantification of the proportions of decadal, millennial, and fossil carbon in the river CO_2 emissions by subtracting the previously estimated amount of carbonate mineral dissolution (with or without an estimate of oxidation rates of fossil organic carbon) as prior estimate from the flux of river CO_2 emissions. Even if you believe that the ^{14}C isotopy of all DIC species is completely equilibrated, it make no sense to believe that all this fossil carbon ends up in the emitted CO_2 . Note that there is still a global riverine flux of about 0.3 Pg DIC/yr to the coast. This means, you also have to account for this lateral DIC export, and take the full 2.3 Pg C/yr of DIC (CO_2 emission + lateral export) as basis for the calculation.

Thank you for pointing this out. We had omitted the river DIC flux to the coastal ocean in our previous version of the river DIC radiocarbon mass balance that focused only on the total CO₂ evasion flux. The reviewer is very much correct that the sum of weathering inputs, atmospheric carbon inputs (some from weathering), and any respiration DIC inputs (as CO₂) must contribute to the sum of both i) the river vertical CO₂ emissions and ii) the lateral DIC export. Our data show that DIC species in rivers mix and exchange isotopically (Fig. S1), meaning that respiration inputs of CO₂ into the river DIC can pass their $F^{14}\text{C}$ value (corrected for stable isotope fractionation) into the HCO₃⁻ pool that dominates river DIC. In that way, even though carbonate weathering alone does not drive any net CO₂ release from rivers as the reviewer points out, the $F^{14}\text{C}$ value of all the inputs are mixed in the DIC pool. In that way, the released CO₂ can acquire this “geological” carbon composition and source.

In response to our omission of the river DIC flux in our calculations, we now explicitly state that we include the lateral DIC flux alongside the vertical CO₂ release (and carbonate precipitation) in the isotope mass balance approach. We have explained this in the manuscript (L. 177-186; Table 1), and have modified our equations to reflect that fact, including expanding this part of the methods in response to comment by Reviewer 3 (L. 935-1017).

These amendments required us to rerun our mixing analysis. The results have changed modestly, and resulted in a higher estimated contribution from millennial-aged carbon (see the comparison Table below, and the revised Table 1 in the revised manuscript):

	Revised values (Pg C y⁻¹)	Original values (Pg C y⁻¹)
Decadal carbon	0.9 ± 0.3	0.79 – 0.91
Millennial carbon	1.1 ± 0.3	0.90 – 1.03
Petrogenic carbon	0.1 ± 0.0*	0.15 – 0.218

* Adjusted value from the range 0.15 – 0.218 Pg C y⁻¹ reported in the same format as the other fluxes for consistency, and converted into the proportional contribution to the vertical CO₂ emission component of the total river DIC flux – see updated methods (L. 1003-1012).

[R1-2]

Further, if your objective is to quantify which carbon sources (as radiocarbon age groups) drive the river CO₂ emissions, then it would make more sense to assume that the fossil (or petrogenic) carbon inputs (which are mainly contributed by carbonate mineral dissolution) rather drive the DIC export to the coast. Then, it would also make more sense to simply split the river CO₂ emissions between the decadal and the millennial carbon pool. I think this way of calculation would make more sense, and it would not change the main message/result of your study.

We respectfully disagree with this comment, and the underlying assumptions of the previous comment. The comments suggest that carbonate weathering and production of DIC produces a pool that does not interact with any other DIC inputs at all. This goes against conceptual understanding and our empirical evidence. First, the DIC system is conceptualised as an exchange at pH~7-8 between CO₂ + H₂O ⇌ H₂CO₃ ⇌ H⁺ + HCO₃⁻. This is what we see in the systems where we have paired total $F^{14}\text{C}$ -DIC (i.e., CO₂ + HCO₃⁻) and $F^{14}\text{C}$ -CO₂ – they have the same $F^{14}\text{C}$ within 0.02 – which is a very minor difference. For instance, when comparing

values of $F^{14}\text{C} = 1.00$ and $F^{14}\text{C} = 0.98$, this equates to 162 ^{14}C years (please see our response to the next comment). We also note that the total DIC pool contributing to river CO_2 outgassing is the main underlying premise of the foundational work quantifying global river CO_2 emissions, because most rivers are supersaturated in CO_2 with respect to the atmosphere (Raymond et al. 2013, ref¹). For the primary analysis, we have therefore chosen to retain the potential for weathering to drive both lateral DIC export and vertical CO_2 emissions (Table 1; Methods).

Finally, you give evidence that ^{14}C isotopy of CO_2 and DIC is close to equilibration, however with an obvious bias towards CO_2 being younger than DIC. You report an average difference in $F^{14}\text{C}$ of CO_2 vs. DIC of 0.02. Couldn't you simply use this number to estimate radiocarbon age of CO_2 rather than assuming that it is equal to that of DIC? How does this difference in $F^{14}\text{C}$ translates to a difference in age? Also, from Figure 1B in the main text, it looks like the average $F^{14}\text{C}$ of CO_2 is clearly higher than that of DIC, and that by more than 0.02. Is that true? And if so, how do you explain that?

The difference of 0.02 $F^{14}\text{C}$ is only for the paired total-DIC and CO_2 samples. This difference is very small compared to the variability we have in the dataset ($F^{14}\text{C}$ range: 0.005-1.262, mean $\pm 1\sigma$: 0.919 ± 0.175 for DIC and CO_2). For reference, for an $F^{14}\text{C}$ value of 1.00, the difference to 0.98 would be 162 ^{14}C years; we report this in the main text (L. 90).

While the CO_2 $F^{14}\text{C}$ may be partially offset from the paired DIC by 0.02 in the paired analysis, we are wary of using a single empirical curve to “correct” for it because we do not have paired data across the full range of $F^{14}\text{C}$ values – this would represent a valuable and important future research direction, which we now mention in Supplementary Information S2 (L. 164-167).

When considering the entire database, $F^{14}\text{C}$ was less variable and more similar among CO_2 and DIC pools at pH values below 7, representing where CO_2 dominated or was equal to DIC in the carbonate equilibrium (Fig. S9). At higher pH values, $F^{14}\text{C}$ was more variable, particularly for DIC, indicating more variable inputs including contributions from older sources. Where lower (older) $F^{14}\text{C}_{\text{atm}}$ -DIC values are seen at higher pH, these are matched by generally lower $F^{14}\text{C}_{\text{atm}}$ - CO_2 values. This relationship is seen in the database in general: where DIC- $F^{14}\text{C}_{\text{atm}}$ values tended to be lower (older), e.g., in the Montane Grassland and Shrubland biome (Fig. 2C) and in catchments underlain by sedimentary lithologies (Fig. 2B), CO_2 - $F^{14}\text{C}_{\text{atm}}$ values were also lower.

Therefore, the main reason we propose for the difference between $F^{14}\text{C}$ in DIC and CO_2 in the full database is that there is likely a sampling bias. Past studies have generally collected CO_2 where the pH of the river is low and so most DIC is present as CO_2 . These settings tended to be peatland streams, or similar, and have relatively low (or minor) inputs from carbonate weathering. For example, several method development studies for $^{14}\text{CO}_2$ were undertaken in acidic peatland streams (e.g., Billett & Garnett, 2010 <https://doi.org/10.4319/lom.2010.8.45>; Garnett et al., 2016 ref⁵⁶). Therefore, we could consider CO_2 measurements may be biased towards a specific type of system. In contrast, river DIC can be sampled and measured easily across a range of pHs, including where CO_2 is

likely lower meaning CO_2 cannot be directly sampled as easily for $F^{14}\text{C}$ analysis. This can be seen in the larger number of DIC measurements in the database compared to CO_2 . We can also explore this by separately considering pH relationships with DIC and CO_2 . Where pH is lower, petrogenic inputs from carbonate inputs are likely limited because carbonate weathering raises pH. We see a trend of lower (older) $F^{14}\text{C}$ in both CO_2 and DIC as pH increases (Fig. S9B-C), indicating that petrogenic inputs from carbonate weathering can influence the river vertical CO_2 emissions. The trend is more pronounced in DIC, likely because of the sampling bias we highlight. This is one reason why we decided to treat the database all together, and not separate DIC from CO_2 .

We now add the core points of this response, and additional figures exploring the relationships between pH and $F^{14}\text{C}$, to Supplementary Information S2 (L. 152-162).

2) Reorganization of results

I think that the analyses of $F^{14}\text{C}$ differences between different biomes, groups of lithology and catchment size classes is quite interesting. It would be good if the extended data Figures 1A, 2 and 3 could be merged into one figure and added to the main text. It would be good to use this figure to try to explain difference in DIC ages between different types of rivers, and discuss the large-scale processes that are at play. That could be a good addition to figure 3, which is also not yet well described in the main text.

Thank you for prompting us to reconsider this. We have combined Extended Data Fig. 1A, 2 and 3 as suggested into the revised Fig. 2. We now also expand the single paragraph discussing these figures into three separate short paragraphs to discuss the underlying processes governing $F^{14}\text{C}_{\text{atm}}$ patterns by catchment size, lithology and biome in more detail (L. 127-153).

We have moved the original Fig. 3 to the Extended Data and have now included a new figure (Fig. 3) in the main text to demonstrate the impact our findings have on our current understanding of the global carbon budget following advice from Reviewers 2 and 3.

There is space to cut down other material for compensation. For instance, the authors performed a random forest model. They describe that model shortly over one paragraph in the main text, without revealing any really interesting or important results. There are no real results shown, neither in the main text nor in the extended data. I really do not see any value from that analysis and would suggest removing it from the manuscript.

We experimented with different approaches to analyse the potential importance of these variables in describing the $F^{14}\text{C}$ dynamics in our database. The random forest model was one such attempt. Although the results were not especially conclusive, we believe it is important to document this analysis for reference. And this somewhat non-conclusive result is still a result: there are no clear and consistent drivers of $F^{14}\text{C}$ at the global scale except catchment size, lithology and biome. This is reflected in the variables highlighted as important by the random forest model ($\text{IncMSE} > 15$): Mean annual precipitation, mean elevation, mean annual temperature, karst area extent, forested area extent, soil organic carbon content, and the sand fraction in the soil. We now present this more clearly in the main text and have

updated the figures presenting the random forest results because both Reviewers 2 and 3 were in favour of retaining this analysis [R1-3].

General comments

L59-61: A part of the CO₂ entering rivers comes from root respiration. This is respiration of organic carbon (in the form of sugars) produced through photosynthesis, but this organic carbon is not part of the biomass. Accordingly, it would be more correct to change “terrestrial respiration of organic carbon recently fixed into terrestrial biomass” into “terrestrial respiration of organic carbon recently fixed by photosynthesis”. That would include also biomass, and therefore both autotrophic and heterotrophic respiration. Moreover, you might want to change “terrestrial” into “(semi-)terrestrial” (or something similar), given the importance of wetlands as sources of C to inland waters.

We agree that not all of the carbon described here is necessarily fixed into biomass, so we have corrected this sentence as suggested [L. 62]. We retain “terrestrial” in an attempt to avoid overly technical language as requested by this reviewer earlier.

L69-70: That sentence looks awkward. You mean chemical rock weathering and erosion can mobilize this carbon which is then routed to the rivers?

Corrected (L. 70).

L70-71: Are you talking about the age of the carbon that is respired, or about when this respiration took place.

We are referring to the age of the carbon respired – we have corrected this sentence accordingly (L. 71-72).

L71: “The latter, aged-carbon” That looks weird. Maybe better “Carbon at the upper end of this age range” or simply “Older carbon”.

Agreed – corrected (L. 74).

L86: For someone not familiar with the metrics you use here, it is difficult to judge whether 0.02 is large or not. Would it be possible to express that in relative terms or in terms of implied age and given in years?

We now express this difference in ¹⁴C years for additional context for example when comparing values of $F^{14}\text{C} = 1.00$ and $F^{14}\text{C} = 0.98$ of 162 ¹⁴C years (L. 90), but we believe the more important context is the comparative range of the $F^{14}\text{C}$ found within the database which is mentioned in the same sentence (L. 91).

L98-99: Now I am a bit confused. One paragraph before, you stated that you use DIC age as surrogate for CO₂ age. Here, the difference in average F¹⁴C of DIC and CO₂ is much larger

than 0.02, and the radiocarbon age of DIC is more than twice of that of CO₂.

Here we refer to the difference between all CO₂ and all DIC $F^{14}\text{C}$ measurements in the database (DIC $n = 973$, CO₂ $n = 197$; supplementary discussion S1); we have adjusted the wording accordingly (L. 104). The 0.02 difference in $F^{14}\text{C}$ referred to in the introduction (L. 90) relates to the paired $F^{14}\text{C}$ -CO₂ and $F^{14}\text{C}$ -DIC measurements only (i.e., samples collected concurrently from the same waters). Please see our response to this reviewer's earlier comment regarding CO₂-DIC isotopic equilibrium.

L115: What is the exact number (maybe relative to whole data set)?

Thank you for this suggestion, we have now added this information (L. 121).

L122-126: The differences in mean F14C per lithology group are smaller than the standard deviation of F14C values within each lithology group. Have you performed any statistical test to prove that these groups are statistically different from each other? The mean F14C value of sedimentary rocks is about 0.05 lower than that of igneous rock group. However, the mean F14C value for igneous rocks is also about 0.05 lower than that of metamorphic rocks. How would you explain these differences? It would be good if you could translate the F14C values into radiocarbon age here as well. That would make the interpretation of your findings easier.

The non-parametric Kruskal-Wallis test result was shown in Extended Data Fig. 3 – now Fig. 2B in the main text. The result is $p \ll 0.05$, indicating significant differences between the three lithologies. This is further demonstrated by pair-wise Wilcoxon T-tests, also shown in the figure panel: the horizontal square brackets indicate the pairing, and the numbers are the p-values from the T-tests, all of which are $\ll 0.05$ indicating statistically significant differences. Age in ^{14}C -years are now indicated on Fig. 2C to aid interpretation. We now discuss these results in more detail in the main text (L. 134-143).

[R1-3]

L134-143: This description of results from your random forest model are not very informative. You name many parameters here, but you do not describe how the retained predictors actually influence the predicted F14C values. Forest cover, for instance, does it increase or decrease the F14C values?

We appreciate the concerns raised by the Reviewer, and have revised and improved the reporting of the random forest results, rather than removing it, because of the positive comments on this analysis by Reviewers 2 and 3.

The random forest approach assesses the importance of variables by shuffling each variable in turn, running the model, and calculating the resulting decline in model accuracy. This method identifies which variables significantly impact the prediction of the target variable but does not reveal the direction or trend of each variable's influence. We thus also calculated the Spearman rank correlation coefficient between each variable and the target variable, which provides insights into the trends in their relationships. We present these in the revised Extended Data Fig. 3-4 and discussion in the main text (L. 155-173).

In our revision, we redesigned and modified the random forest result visualisation and linked it to our previously plotted scatter diagrams and reporting of the Spearman rank correlation. This approach offers a more intuitive way to understand the relationships between variables.

In the revised Extended Data Fig. 3-4, the colour of the bars represents the Spearman correlation coefficient: blue indicates a positive correlation, meaning the target variable increases as the variable increases, while orange represents a negative correlation, where the target variable decreases as the variable increases. Asterisks denote statistical significance of the correlation.

For example, forest cover extent is identified by the random forest model the fifth most influential variable on DIC $F^{14}C_{\text{atm}}$ for large catchments. Its Spearman correlation coefficient is negative, indicating that $F^{14}C_{\text{atm}}$ tends to decrease as forest cover increases. This trend is also reflected in the scatter plot of the revised figure (Extended Data Fig. 3F), which visually presents this relationship.

L149-150: Do you mean “limited data availability”?

Yes thank you, corrected (L. 179).

[R1-4]

L147-156: What is the $F^{14}C$ (and radiocarbon ages) that you assumed for these petrogenic inputs and the river CO_2 emissions? How sure can you be that they are representative for the global scale, in particular for the global CO_2 emissions? I see from Figure 1 that your data set is geographically skewed, with only few data from extensive regions in particular in Siberia and Africa. Have you performed any weighting of the data across geographic regions, biomes, lithologies, or basin size to obtain representative average $F^{14}C$ values? Finally, if the $F^{14}C$ is 0.994-1.031 after subtracting the contributions from petrogenic C inputs (which are about one tenth of the total flux), doesn't that mean that the vast majority of river CO_2 emissions is fueled by relatively young carbon. Anyway, it would be good to translate all the $F^{14}C$ values into radiocarbon ages to be able to interpret these findings more easily.

The $F^{14}C$ content of the petrogenic inputs was set at 0.001 ± 0.001 for the Monte Carlo simulations and the Bayesian isotope mixing model – petrogenic carbon will have no measurable ^{14}C content (representing a radiocarbon age $\gg 50,000$ ^{14}C years), but using an $F^{14}C$ value of zero will not allow for solutions to the mass balances to be found, so we instead use a very low $F^{14}C$ value in line with previous mass balances (e.g., Elder et al., 2018, <https://doi.org/10.1038/s41558-017-0066-9>). The $F^{14}C$ content of the other end members (decadal and millennial) are defined in the Methods (L. 989-991): decadal inputs = 1.011-1.442 (“modern”), millennial inputs = 0.808-0.874 (1390 +/- 310 from ref²⁸).

For the second part of this comment, please see our detailed responses to Reviewer 2 [R2-1], where we discuss the representativeness of the database, and in a separate response where we discuss upscaling approaches [R2-2]. In short, the database is representative in terms of biome and lithology. We have, for example, missed large parts of the Eurasian Arctic, which

would arguably release older carbon from permafrost stores compared to Africa. However, this is only supposition and is an important potential focus for future work as mentioned in Supplementary Information S4.

For the third part of the comment regarding the residual $F^{14}\text{C}$ value that remains after correcting for petrogenic inputs in the two end-member Monte Carlo analysis, we first undertake a sense check analysis. From the mean $F^{14}\text{C}$ content of our database (0.919, or an average age of 679 ^{14}C years), we can check whether this could just represent a simple mixing between petrogenic and decadal carbon using a simple two end-member mass balance, where the proportional contributions from the two carbon sources sums to 1:

$$F^{14}\text{C}_{\text{mean}} = F^{14}\text{C}_{\text{decadal}} + F^{14}\text{C}_{\text{petro}}$$

where we use the same end-member values for $F^{14}\text{C}_{\text{decadal}}$ as in the Monte Carlo and Bayesian analyses (1.226 ± 0.216) and an $F^{14}\text{C}_{\text{petro}}$ value of (0.001 ± 0.001 , because using 0 would yield no solution). We then solve for the proportional contributions of these two end-members that would give an $F^{14}\text{C}_{\text{mean}}$ value of 0.919: 0.75 decadal carbon and 0.25 petrogenic carbon. Based on a total river DIC flux of 2.5 Pg C y^{-1} (see response to this reviewer's first main comment), this would require petrogenic contributions of $0.625 \text{ Pg C y}^{-1}$, more than triple the current highest estimate of river DIC contributions from weathering ($0.218 \text{ Pg C y}^{-1}$). This is not feasible based on the current state of knowledge, and provides support for an additional input of ^{14}C -depleted DIC which we identify as millennial carbon.

We have now updated the two end-member Monte Carlo simulation to account for lateral DIC export to the oceans, as importantly highlighted by this reviewer. The residual $F^{14}\text{C}$, after correcting for the background petrogenic inputs, in the revised simulation is 0.978-1.007 (179 ^{14}C years - "modern"), which is more ^{14}C -depleted (older) than the previous residual (0.994-1.031; 48 ^{14}C years - "modern"). Given that a simple mixing of petrogenic and decadal carbon would require an outsized contribution from petrogenic carbon to explain the mean $F^{14}\text{C}$ value of our database, we require an additional ^{14}C -depleted end-member to explain our observations, which is likely to be aged soil organic carbon – the millennial end-member we include in our analysis.

Fig. 3: That is an important conceptual figure. It would be could if it was better described in the text, with all the processes that you show here.

We have moved the original Fig. 3 to the Extended Data Fig. 7 and have now included a new figure (Fig. 3) in the main text to demonstrate the impact our findings have on our current understanding of the global carbon budget following advice from Reviewers 2 and 3. Both figures have now been better incorporated into the main text discussion.

L733 and following, and Fig. 2 from extended material: The analysis per biome looks actually interesting. I don't understand why you do not really discuss these results in the main text.

We now discuss these results in more detail in the main text (L. 145-153).

L793-796: That is not necessary, as random forest deals naturally with multi-collinearity

The Reviewer is right that the random forest model itself can deal with multi-collinearity. But the metric we used for measuring the importance of variables, IncMSE (Increase in Mean Squared Error), is unsuitable for datasets with correlated variables. When a random forest model uses IncMSE to measure variable importance, highly correlated variables can have a significant impact on the results. Their importance may be distributed, leading to underestimated scores, or redundant variables may appear overly influential due to correlation. This reduces interpretability, making it difficult to identify true drivers of the target variable, risking misinterpretation in complex models (Dorman et al., 2013 <https://doi.org/10.1111/j.1600-0587.2012.07348.x> ; Stein et al., 2021 <https://doi.org/10.1029/2020WR028300>). This is why we excluded the highly correlated variables first.

L848: There is a “which” too much.

Corrected (L. 995).

L862-865: That means you exclude CO₂ from organic soils?

This analysis is just to separate the petrogenic component of this flux from the other two end members (decadal and millennial). CO₂ from organic soils is included in the millennial end member in the Monte Carlo and Bayesian analyses (Extended Data Fig. 5-6).

Fig. S1: It looks like that is strong tendency for higher F¹⁴C of CO₂ vs DIC, implying that CO₂ is mostly younger than DIC. You mentioned an average difference of 0.02. What would be the implied average difference in age? Would it be possible to make this plot also with radiocarbon ages?

The relationship does indeed have a slight tendency towards younger CO₂ as the reviewer suggests. The slope of the relationship is 0.66, as opposed to an ideal relationship of 1. However, the majority of the data lies between F¹⁴C 0.9-1, where there is very good agreement between the paired values; unfortunately, the lack of values below 0.9 means it is challenging to make predictions or explore this relationship in detail below these values with the existing paired CO₂-DIC dataset.

It is not straight forward, nor especially informative, to add ¹⁴C years to these specific plots as several of the values are F¹⁴C > 1, meaning a radiocarbon age cannot be calculated and are considered “modern”. The main comparison presented here is whether or not the CO₂-DIC observations contain the same amount of radiocarbon, which is reported throughout this manuscript at F¹⁴C (fraction modern) for consistency. We do, however, report the difference as estimated in ¹⁴C years as requested – please also see our response to this reviewer’s earlier comment regarding CO₂-DIC isotopic equilibrium.

Referee #2 (Remarks to the Author):

General comments

Dean et al. present a first global estimate of the age of carbon (C) that is emitted by streams and rivers. They use radiocarbon (^{14}C) analysis to partition the riverine CO_2 emission flux between fossil (dead C from carbonate weathering and rock organic C), millennial (old soil C), and decadal (C recently fixed by vegetation) sources. Based on a combination of data mining from the literature and new data, they show that about half of the riverine CO_2 emissions originates from C of millennial to fossil origin.

There is growing evidence from local case studies that old C can be emitted from rivers, but both a global synthesis and a conceptual framework are still lacking. While much research has focused on quantifying the export of old C from weathering and rock organic C (i.e. the fossil fraction as described by the authors), very little is known about the magnitude of older soil C export (i.e. the millennial fraction). By showing that this release of old terrestrial C to the atmosphere is significant at a global scale, this study addresses an important gap in our understanding of the linkages between terrestrial and aquatic C stores. While this is a first-order attempt at a global estimate of old C emissions, which will need to be refined as we get more data from different regions and systems, the authors' findings are timely and should be of immediate interest. Their findings have implications for global C budgets, including the possibility that anthropogenic C storage in the biosphere is more important than previously thought.

Thank you for your positive comments, we are pleased that the main messages and importance of the study come across. The comments below have been extremely helpful to refine our work.

[R2-1]

While I am overall supportive of the study and its findings, I have a few concerns regarding the methodology that I would encourage the authors to address. My first concern is related to a potential observation bias. The authors briefly mention this issue, but it would be useful to quantify it further. Could the authors determine how many of the studies in their dataset specifically target systems where we would expect old C export, e.g. thawing permafrost, glacier retreat, or peatland drainage? Do the authors' findings still hold if these studies are excluded from the database? If most studies in the dataset focus on regions with expected old C emissions, this could skew the global estimate towards higher fossil/millennial C contributions.

We thank the reviewer for raising this important point. We originally attempted to address potential bias within the database in the manuscript, mainly in the more detailed discussion in the supplement in Supplementary Information S1 (representativeness) and S4 (potential bias of studies looking for old carbon). However, at the request of the reviewer, we have tried to quantify this in more detail.

It is not straightforward to identify and exclude old values from the database based on the purpose of the original study. We already removed any potential values from groundwater seeps or other old end-member observations during the initial data collection: we included only values from open river and stream channels. Any further removal would be subjective anyway, potentially adding a different kind of bias. We have further explored representativeness below, based on the existing database, and include an additional figure in the supplement (Fig. S4) and a brief discussion on the core points below in the Supplementary Information S1 (L. 38-50).

First, we consider catchment size. If there is a skew towards “disturbed” catchments with old carbon, then smaller catchments should be older than larger catchments as larger catchments are less likely to be impacted by specific processes impacting the age of carbon exported, such as localised weathering or erosion. In contrast, we found that $F^{14}\text{C}$ values were lower (i.e., older) as catchments got larger (revised Fig. 2A and Extended data Fig. 2). This contradicts the above hypothesis, suggesting that smaller catchments have younger carbon and that contributions of old carbon to river CO_2 are occurring across large scales (now included in the main text L. 127-132).

Next, we compared the proportional global landmass surface areas of the lithological classes defined in the methods section with the proportion of values in our database from each lithological class. For Igneous lithologies, these represent 14% of the global landmass while 29% of our database values come from these lithologies. For metamorphic lithologies, the global landmass is 13% compared with 7% of our database values. And for sedimentary lithologies, the global landmass is 63% compared with 55% of our database. Overall, there is a fairly good match in terms of representative proportions, with a slight bias towards igneous lithologies which tended to have the less depleted (younger) ^{14}C values (revised Fig. 2B).

Finally, we compared the global landmass surface areas of the biome classes defined in the methods section with the proportion of values in our database from each biome:

Biome	Global landmass coverage	Database proportion	Representativeness
Temp. grass & shrub	8%	2%	Under by 6%
Trop. grass & shrub	17%	4%	Under by 13%
Temp. conifer & boreal for.	14%	25%	Over by 11%
Trop. broadleaf & conifer for.	18%	21%	Within 3%
Temp. broadleaf & mixed for.	12%	31%	Over by 19%
Tundra	9%	7%	Within 2%
Montane grass & shrub	4%	5%	Within 1%
Deserts	19%	4%	Under by 15%

In general, there is fairly good agreement, with most percentages within $\pm 10\%$. The most under-represented biomes were tropical grasslands and shrublands (-13%) and deserts (-15%). Tropical grasslands and shrublands were the biome with second most depleted (older) ^{14}C values (revised Fig. 2C), indicating a potential bias in the database towards younger carbon; while deserts tended to be younger and are not very productive systems anyway, so likely contribute a relatively small amount of carbon to the global river CO_2 emissions. The

most over-represented were temperate conifer & boreal forests (+11%) and temperate and mediterranean broadleaf mixed forests (+19%). The former biome tended to contain less depleted (younger) ^{14}C and the latter tended to contain more depleted (older) ^{14}C (revised Fig. 2C), so there is no clear bias towards older or younger river CO_2 emissions in the database based on biome representativeness. These biome biases do indicate a geographical bias which we highlight already as important directions for future research (Supplementary Text S4; L. 255-262).

My second concern is related to the isotope mixing models. The Bayesian approach, while presumably more thorough than the simpler two-endmember model, seems to have an underdetermination issue. This is because the authors are assuming three different C sources, but only have data for a single tracer (^{14}C), leading to an underdetermined system. This issue can result in uncertain and potentially biased estimates of the contributions from each C source. Did the authors assess the reliability of their estimates, e.g. did they perform any sensitivity analysis? One potential workaround to the underdetermination issue would be to constrain the model with a strong prior on the fossil C contribution, but this does not seem to have been done.

The reviewer suggests we could run a mixing model with a prior on the petrogenic carbon contribution – this is exactly what we had done and is the main output that we discuss in the original manuscript: that is, Extended data Fig. 5 using a Monte Carlo simulation. We have now made edits to the manuscript to make sure it is clear how these two approaches contribute to the paper (L. 177-186).

We agree that the Bayesian approach has some drawbacks (underdetermination). For this reason we used it as a secondary approach, or a check on the two end-member Monte Carlo simulations, which uses a petrogenic carbon contribution prior based on independent petrogenic carbon inputs estimates (Gaillardet et al., 2019 and Zondervan et al., 2023, refs^{22,40}). Interestingly, the Bayesian three-end member mixing model does produce similar results and requires two ^{14}C -depleted reservoir contributions for the vast majority of solutions.

On another note, I am wondering whether using corrected $F^{14}\text{C}_{\text{atm}}$ values instead of raw $F^{14}\text{C}$ values in the mixing models might help reduce some of the uncertainties related to the changes in ^{14}C atmospheric input over time.

We thank the reviewer for this interesting suggestion. We originally ran the petrogenic-constrained (see response to previous comment) two-end member Monte Carlo simulation and the Bayesian simulation using $F^{14}\text{C}$ values and a wider potential range of the decadal end member (mean atmospheric $F^{14}\text{C}-\text{CO}_2$ 1950-2019 = 1.226 ± 0.216) to account for the temporal variability in atmospheric $F^{14}\text{C}-\text{CO}_2$ which we have shown to be an important contributor to the database values through time (revised Fig. 1C). By using $F^{14}\text{C}_{\text{atm}}$ values and this wide range of $F^{14}\text{C}$ for the decadal end-member, we would be double accounting for the variability of atmospheric $^{14}\text{CO}_2$.

To address this comment, we tried rerunning the model using $F^{14}\text{C}_{\text{atm}}$ values and a narrower range of the decadal end-member ($F^{14}\text{C} = 1.0\text{-}1.1$) because there will still be some variability in atmospheric ^{14}C inputs and the $F^{14}\text{C}_{\text{atm}}$ normalisation does not fully correct for atmospheric $^{14}\text{CO}_2$ variability as we discussed in the original submission (e.g., $F^{14}\text{C}_{\text{atm}}$ values > 1 in revised Extended Data Fig. 1). In the table below we compare the outputs of the two model runs for both the proportional contributions and their estimated vertical CO_2 emission to the atmosphere (Pg C y^{-1}), alongside their uncertainties:

	$F^{14}\text{C}$ data (mean \pm σ)	$F^{14}\text{C}_{\text{atm}}$ data (mean \pm σ)
Figure	Fig. R1 - LEFT	Fig. R1 - RIGHT
Decadal proportion	0.44 ± 0.17	0.49 ± 0.09
Decadal flux	$1.0 \pm 0.4 \text{ Pg C y}^{-1}$	$1.1 \pm 0.2 \text{ Pg C y}^{-1}$
Millennial proportion	0.56 ± 0.17	0.51 ± 0.09
Millennial flux	$1.3 \pm 0.4 \text{ Pg C y}^{-1}$	$1.2 \pm 0.2 \text{ Pg C y}^{-1}$

While we recognise that, as the reviewer suggests, this does indeed reduce the uncertainties presented, we are concerned this creates a false narrative. By incorporating the actual variability in atmospheric $^{14}\text{CO}_2$ since 1950 in the original simulations, we provided a more realistic representation of the uncertainty within the system and our analysis. By using the $F^{14}\text{C}_{\text{atm}}$ data in the mixing model simulations, we would be underrepresenting the real uncertainty introduced by the atmospheric $^{14}\text{CO}_2$ variability caused by atmospheric nuclear weapons testing and the subsequent decay and dilution of this signal in the atmosphere. Because the $F^{14}\text{C}_{\text{atm}}$ normalisation cannot fully account for atmospheric $^{14}\text{CO}_2$ variability through time, we believe it is better to run the analyses with the $F^{14}\text{C}$ data (adjusted for the change in total carbon flux in the system, see response to Reviewer 1 [R1-1]), even though the reported uncertainties from this analysis are higher than when using $F^{14}\text{C}_{\text{atm}}$.

Fig. R1. Comparison of petrogenic-constrained two-end member Monte Carlo simulations using $F^{14}C$ versus $F^{14}C_{\text{atm}}$. LEFT – the outputs of the Monte Carlo simulation using $F^{14}C$ data from the database. RIGHT – the outputs of the Monte Carlo simulation using $F^{14}C_{\text{atm}}$ data from the database. *The figures otherwise follow Extended Data Fig. 5 in the manuscript.*

[R2-2]

My third comment is about the upscaling approach. While I understand the purpose of developing a first-order, global estimate, I think introducing one additional step in the analysis could be beneficial. For instance, could the authors provide lithology-specific percent contributions for each age range (decadal, millennial, fossil; i.e. like they did through the Bayesian model), and then scale the relative emissions from each lithology? This could potentially be done by overlapping the river network and lithology maps from HydroATLAS. This additional step would better highlight how geology affects riverine CO₂ emissions and improve the accuracy of the authors' upscaled global flux.

We like this idea, and had discussed doing something similar ourselves earlier in the analysis. Unfortunately, we think that this step would be pushing the database too far at this point in time. That is because to do this properly would require: i) basin specific river $p\text{CO}_2$

measurements and CO₂ evasion fluxes; ii) basin specific carbonate weathering rates and total DIC flux (because the Bayesian analysis remains unconstrained by petrogenic inputs, please see our previous response on this); iii) combined with coverage of $F^{14}\text{C}$ measurements spatially and temporally.

Unfortunately, at present most global basins only have estimated river $p\text{CO}_2$ and CO₂ evasion fluxes (e.g., Raymond et al. 2013, ref¹). Global carbonate weathering rates are well known in large basins (Gaillardet et al., 2019, ref²²) but are more uncertain at the smaller basin scale of many of the database $F^{14}\text{C}$ measurements. We believe this is a really interesting avenue for future research or research project design and we have explained these themes in the revised version (Supplementary Text S4, L. 229-237). In lieu of upscaling estimates based on basin lithology, we have expanded on the discussion surrounding the geological, biome and other controls on the $F^{14}\text{C}$ values in the database, and have moved the figures exploring this data into the main text (revised Fig. 2 and associated discussion in the main text, and the revised presentation of the random forest model [R1-3]).

Regarding the implications of the authors' upscaling for the global C budget, could the authors try and revisit the budget presented in Regnier et al. (2022)? A revised budget (with for instance an increased terrestrial storage of anthropogenic C) could be a useful addition to Figure 3, which is very conceptual at the moment.

We have moved the original Fig. 3 to the Extended Data and have now included a new figure (Fig. 3) in the main text to demonstrate the impact our findings have on our current understanding of the global terrestrial carbon balance. In the new figure, we show the previous paradigm and updated impact on the global carbon budget of our findings following suggestions from Reviewer 3 – this budget is based on Regnier et al. (2022; ref³).

The reporting of the methodology is detailed and transparent. Statistical tests are appropriate and uncertainties are clearly presented. One exception might be Extended Data Figure 5, where it is unclear to me what the error bars represent. The manuscript also references previous literature appropriately. I have listed more specific comments below.

We have now added a description of the boxplots to the figure caption to clarify what these bars mean (Extended Data Fig. 6).

Specific comments

Abstract is clear, but some improvements could be made. “Centennial” should be replaced with “millennial” for consistency with the rest of the paper. The last sentence is not entirely clear to me, please try to link it better to the findings.

We correct “centennial” to “millennial” as requested.

We have now updated the final sentences of the abstract to reflect this comment and those from Reviewer 3 (L. 39-44):

“This previously unrecognised release of pre-industrial aged carbon to the atmosphere from long-term soil, sediment and geologic carbon stores via lateral hydrological routing equates to $1.2 \pm 0.3 \text{ Pg C yr}^{-1}$, similar in magnitude to terrestrial net ecosystem production. A consequence of this flux is a greater than expected net loss of carbon from aged organic matter stores on land. This requires a reassessment of the fate of anthropogenic carbon in terrestrial systems, and in global carbon cycle budgets and models.”

Fluxes are reported in different units throughout the manuscript, e.g. Tg CH₄ yr⁻¹, Pg C-CO₂ yr⁻¹. I suggest using consistent units throughout, e.g. Pg C yr⁻¹.

At the global scale, methane fluxes are generally reported as Tg CH₄ y⁻¹ (e.g., Saunio et al., 2024; <https://doi.org/10.5194/essd-2024-115>), and for consistency with the literature we have kept this unit where used in the introduction (L. 52). In the same sentence, we also described the CO₂ flux in Pg C y⁻¹, but in a slightly more lengthy manner because we are defining carbon dioxide as CO₂ at the same time. Everywhere else in the manuscript we have reported fluxes as Pg C y⁻¹ and further specify CO₂ or DIC when required in the context of what is being reported/discussed. We have gone through the manuscript to ensure all fluxes are reported predominantly in Pg C y⁻¹ to avoid confusion (e.g., revised Table 1).

L88. I found the justification of using ¹⁴C-DIC as a proxy for ¹⁴C-CO₂ well supported. I wonder if one of the paired DIC/CO₂ measurements was conducted in an area underlain by carbonate geology? It would be interesting to see how DIC and CO₂ ages compare in rivers with potentially old C contributions and high pH.

We are glad our efforts to justify the use of ¹⁴C-DIC as a proxy are appreciated, echoing comments from other reviewers. The suggestion here to explore the full range of pH, lithology and other potential influences on DIC and CO₂ isotope equilibrium dynamics is an excellent suggestion for future research. Unfortunately, the existing paired DIC-CO₂ isotope data is limited, so we have added this suggestion for future research to the end of Supplementary Text S2 (L. 164-167).

L128-132. Can this be tested? Have these predictors (e.g. peatland extent, permafrost extent) been included in the random forest model?

This is evident in specific case studies (e.g., Dean et al., 2020; ref²¹). In our analysis, wetland extent, soil organic carbon content, and permafrost extent were included in the random forest model (for a full list of variables included in the model, please see Supplementary Table S4). In catchments smaller $\leq 10 \text{ km}^2$, soil organic carbon content was found to be a potentially important variable contributing to $F^{14}\text{C-DIC}$, but with a low R^2 value and a relatively unclear directionality of influence (Extended Data Fig. 4; please see the revised reporting and discussion on the random forest results following comments by Reviewers 1 and 3 [R1-3]). Alongside the more detailed consideration of the random forest model, we also now further explore the influence of catchment size, lithology and biome on river $F^{14}\text{C}$ in the main text (L. 127-173).

We have also now demonstrated, in response to Reviewer 1 [R1-4], that petrogenic carbon contributions to the global river DIC-CO₂ flux are not large enough to explain the $F^{14}\text{C}$ values in our database, demonstrating that another ^{14}C -depleted carbon source is required, which is likely to be aged soil carbon.

L142. The use of “terrestrial GPP” here is a bit confusing. I would suggest replacing with “recently fixed carbon” or similar.

Corrected (L. 172).

Figures. Please consider combining Figures 1 and 2 (in Figure 1, either B or C should be removed), and have Extended Data Figure 5 in the main body of the paper – that is, if and when the underdetermination issue is sorted.

We have combined Figures 1 and 2 as suggested, and moved the old Fig. 1C to Fig. S1, and the old Fig 2B to Extended Data Fig. 2.

Some elements of Supplementary Discussion S3 should appear in the Methods.

We have chosen to keep Extended Data Fig. 5 – now Extended Data Fig. 6 – in the extended data as it is a secondary analysis to independently explore the outputs of our petrogenic-constrained two-end member mixing model (Extended Data Fig. 5). We also retain the supplementary Discussion S3 around this mixing model in the supplement for the same reason. We have now clarified the wording and justification surrounding this in the main text (L. 210-215).

Referee #3 (Remarks to the Author):

Summary: Dean et al. assesses the age and source of carbon emissions from global rivers, focusing on Dissolved Inorganic Carbon (DIC), carbon dioxide (CO₂) and methane (CH₄). They used Fraction Modern $F^{14}\text{C}$ as a surrogate of carbon isotope to quantify carbon age, and compiled a comprehensive global database of $F^{14}\text{C}$ for DIC and CO₂ in global rivers (~1,200 data points). The study concludes that a significant portion (47-56%) of global river CO₂ emissions is derived from old carbon (centennial or older). If this is true, this previously unrecognized flux of old carbon from land to the atmosphere is not accounted for in global carbon models. The results could have enormous implications for understanding and quantifying carbon cycling. The topic is of interest to the carbon community and to those who are interested in carbon – climate feedback.

This is a nice summary of the implications of our study – thank you. The thoughtful and detailed comments below are important and we have replied and used them to improve our paper and strengthen the description of assumptions, caveats in the approach, and resultant certainties of the main outputs.

I enjoyed reading the introduction. The terms (different carbon ages) are well defined. The importance of carbon emission are well contextualized in the bigger carbon cycles. I am not an expert in carbon isotopes but I like the idea of using F14C as a surrogate for the isotope composition of river CO₂ emissions, especially when this is supported by additional data that show that paired DIC and CO₂ F14C values are within 0.02 of one another. This work compiled a global database of radiocarbon content F14C in river CO₂ and DIC, some are new data whereas most of the data are from published work. Random forest models were used to identify the influential factors of F14C. An independent Bayesian isotope mixing models are also carried out, which leads to the same conclusion.

Originality and Significance: The study is novel as it quantifies carbon age in riverine emissions at the global scale, an underexplored aspect of the global carbon cycle. By compiling a large database of radiocarbon measurements across the globe, it provides a new perspective on the importance of old carbon flux in rivers and their vulnerability to climate change. To me the major finding is that there are much more old carbon in riverine carbon emission, which expand the current understanding of riverine carbon dynamics and challenge traditional carbon models that largely overlook old carbon emissions.

The idea of aged carbon fluxes into rivers and streams are not new, as shown in some of the relevant papers the authors cited (Barnes et al., 2018, <https://pubs.acs.org/doi/full/10.1021/acs.est.7b04717> ; Butman et al., 2015, <https://www.nature.com/articles/ngeo2322>). Existing work has mostly looked into aged carbon ($\delta^{14}\text{C}$ -DOC) in dissolved organic carbon (DOC) and estimated about 3%-9% as coming from aged carbon in DOC, which is much smaller than the reported 47-56% reported here. The idea of old carbon also collaborates with recent literature on deep respiration, for example, Tune et al., 2020, <https://agupubs.onlinelibrary.wiley.com/doi/full/10.1029/2020JG005795>.

We agree that the idea of older C inputs is not new. However, as stated here, previous work has focused on other carbon pools (e.g., DOC being quite young overall, POC being quite old overall), and not the DIC and CO₂ components of river CO₂ emissions. We agree that the presence of old DIC makes sense, given that deep respiration processes can produce very high $p\text{CO}_2$ in the near sub-surface of the critical zone, and we now include this great paper in the introduction (ref²⁷).

Methodology. I am not sure if I can fully believe the numbers reported, such as 47-56% from old carbon. And I am talking as a modeler and someone whose expertise is in crunching numbers. My major struggle is that I do not fully understand how these important numbers were calculated. There are some text in line 169-174, and the paper refers to the Methods section and Extended Figure 4. The methods include equation 4, which I am still puzzled after reading multiple times. In order to calculate the total fraction of old carbon in DIC globally, I expect the concentrations of DIC and CO₂ are important. But I don't see that in the equation. Are DIC and CO₂ concentrations not important at all? What is the reasoning behind this? I understand that the approach is using a "global" perspective and use the global riverine carbon emission to back calculate how much is from old carbon. To me this seems like a

circular argument. Wouldn't the numbers differ substantially in different places with different old carbon content and DIC concentrations?

The reviewer calls for more clarity on our radiocarbon mass balance approach. We have completely rewritten the section in the methods on this, adding an expanded set of underlying equations which we hope allow the reviewer to fully break down our approach (L. 932-1012). Our previous version had simply given the single governing equation. It is important to note that the fluxes of carbon (lateral DIC export and vertical CO₂ emissions) are central to the approach; these fluxes are driven by concentrations and hydraulics, so both key components of this flux are implicitly included in the calculations. At the global scale, and with the inconsistent availability of paired concentration and hydraulic characteristics with the $F^{14}\text{C}$ data extracted from the literature (hence our use of HydroATLAS), it is not currently possible to dive deeper into these aspects. We hope our approach is clearer now.

We have also thought about other ways to help support our conclusion that we need another ^{14}C -depleted (old) source of DIC, other than from petrogenic sources, to explain the global $F^{14}\text{C}$ data we present. A basin specific approach could be one option – but as we reply to Reviewer 2 [R2-2], this is not possible at present with the database and new measurements we present: we don't have good enough river CO₂ evasion flux data for many global rivers.

As an alternative, we have considered the simple case where only rock weathering and decadal (very young) carbon could explain the river DIC and CO₂ $F^{14}\text{C}$ content [R1-4]. In this case, we find that the amount of rock weathering we would need 0.625 Pg C y⁻¹ to explain the ^{14}C -depletion. This is three times larger than any current estimate and demonstrates clearly that the $F^{14}\text{C}$ DIC data require an alternative ^{14}C -depleted input. Thus, while future basin specific work will be able to refine our mixing contributions (and shed additional light on the drivers of patterns), this analysis shows we need an old carbon input from organic matter degradation in soil or rivers.

In addition, by including the riverine lateral DIC flux to the ocean as spotted by Reviewer 1 [R1-1], the data call for even more ^{14}C depletion not derived from rock weathering. That said, we are aware that these global calculations have caveats. We have thus updated how we present (mean $\pm 1\sigma$) and discuss our estimates (please see our detailed response to this Reviewer's later comment where we explain these scenarios in more detail [R3-1]).

Monte Carlo simulation was also used to “generate 10,000 model runs, varying the fossil flux (0.15-0.218 Pg C y⁻¹), and the $F^{14}\text{C}$ values of the decadal (1.011-1.442) and millennial (0.808-0.874) inputs to Eq. 4.” Again, there are no concentrations involved.

Concentration and hydraulic controls are implicitly included in the flux component of the mass balance calculations as described in our response above.

Even if I understand the calculation clearly, the reported numbers would have gigantic uncertainties. We are talking about 1,200 data points sparsely distributed across the globe, with no indication of when they were measured. The $F^{14}\text{C}$ values could differ across different times of the year significantly. For example, we generally know that at low river

flow and dry time, there are much high proportions of water coming from older groundwater compared to conditions at high river flow where typically there are much more water coming from relatively shallow zone

(e.g., <https://agupubs.onlinelibrary.wiley.com/doi/full/10.1029/2010WR009341>). Stream flow during dry time probably has much more old carbon compared to wet times of the year. There are often orders of magnitude difference in river flow in different times of the year, which could lead to substantial differences in export of carbon at different ages (e.g., Duvert et al., 2016, <https://www.nature.com/articles/s41561-018-0245-y>; Duvert et al., 2019, <https://agupubs.onlinelibrary.wiley.com/doi/full/10.1029/2018JG004912>; Johnson et al., 2008, <https://agupubs.onlinelibrary.wiley.com/doi/full/10.1029/2008GL034619>).

Sample timing for every value used in this study is available in Table S1. Time is an important component of our analysis as was shown in Fig. 2A (now revised Fig. 1C; Extended Data Fig. 1). However, sub-annual analyses are challenging, if not impossible, to include in our analysis with the available data. With 1200 values included in our analysis spread across a wide range of locations around the globe, including good representation of lithological and biome variability (please see our response to Reviewer 2 [R2-1] on the representativeness of the database), we can assume we cover at least some of the variability highlighted by the reviewer here. Many of these detailed variations in river carbon supply will require detailed future work to untangle, which we highlight in the final paragraph of the Supplementary Text S4 (L. 255-262).

[R3-1]

So in generally I am not very confident about these numbers. As the paper later discussed, whether the old carbon contributes 47-56% or some other numbers like 20-30% could have enormous implications on how we think about carbon cycling, and could completely change how we think about modern C cycle. This makes it even more important to get the right numbers.

We agree it is important to get the most accurate and precise estimates that we can with the available information at hand. However, we would make the case that any old carbon contribution to this vertical CO₂ loss from river surfaces shifts our current paradigm and is an important contribution. So, even while we have relatively large uncertainty, and future catchment-specific studies will surely provide novel insight to help refine these estimates, a key focus of our paper is to make the point that the data require some ¹⁴C-depleted (likely millennial-aged) inputs alongside the petrogenic ones.

With this comment in mind, and in line with others from the reviews, we have made a new figure to illustrate this shift in how we think about river CO₂ emissions in the modern carbon cycle (Fig. 3). The existing view is that only very young CO₂ contributes to river degassing (Fig. 3A) and that old carbon in DIC is from rock weathering (carbonate minerals). This is largely a reflection of the pioneering paper by Mayorga et al., 2005 (ref¹¹), on the “age of the Amazon’s breath”. However, by recognising the need for old biospheric carbon input on top of petrogenic inputs to explain the measured *F*¹⁴C values in our database, we have to shift this view. Any old carbon input from soil or river organic matter degradation requires a net loss from that old soil carbon pool in the modern day (Fig. 3B). This means the uptake by the

youngest terrestrial carbon pool must be larger than previous estimates. This was all discussed in the previous version, but we hope the new figure makes the importance of our finding clearer.

In terms of providing confidence in our assertion that an old biospheric input is needed, and that our approach provides a robust first quantification of this, we take three approaches to reassure the reviewer.

First, we consider the simplest scenario that the DIC and CO₂ pool in rivers only corresponds to a mixture of very young and petrogenic inputs [R1-4]. This is the current paradigm where there are no ¹⁴C-depleted inputs other than petrogenic sources (new Fig. 3A). We find that the amount of petrogenic carbon inputs would need to be three times larger than any current estimate of weathering inputs. Given that our database may be slightly biased against settings with sedimentary rocks where petrogenic inputs are present [R2-1], this clearly demonstrates that the *F*¹⁴C DIC data require another ¹⁴C-depleted input that has a higher *F*¹⁴C value than petrogenic carbon sources.

Second, there is a growing recognition of the importance of the whole soil profile (i.e. deep soils) in soil respiration fluxes (Hicks-Pries et al., 2017, ref³⁰). This literature is very much in step with our findings. Despite never being fully demonstrated until the analysis we present here, it should not be a surprise perhaps that water flowing through soils and the shallow subsurface will mobilise CO₂ and DIC derived from older organic matter respiration given previous evidence of mobilisation of POC from deep soil layers into rivers (Eglinton et al., 2021 <https://doi.org/10.1073/pnas.2011585118>). We have clarified discussion of these themes in the manuscript (L. 68-82).

Finally, when we try and quantify the potential contributions of old carbon to the total river DIC pool, we take two approaches to quantify the relative inputs of Decadal, Millennial and petrogenic carbon sources. We run a binary mixing model to constrain the relative potential contributions of Decadal and Millennial carbon after accounting for known weathering carbon inputs to global rivers. And we run an unconstrained three end-member Bayesian approach which has the drawback with no priors. The two approaches produce similar outputs (Table 1), both requiring a sizable component of the river CO₂ flux to be derived from some ¹⁴C-depleted input other than petrogenic – i.e., old organic matter in soil or river sediments.

We are aware that future work may refine our findings based on new datasets. This is likely to be in the form of detailed work in space and time across river catchments which collect *F*¹⁴C samples in a way we have not been able to do so far (even with 1200 measurements). In other words, based on the data available now, our interpretation of old carbon inputs being required to explain the data is conceptually sensible, and we are able to provide a quantitative handle on it for the first time. As such, our study makes a new and important contribution to this research theme.

To address this important comment, we have changed the way we report the outputs of the main Monte Carlo simulations throughout. We now report the mean proportional

contributions of the decadal and millennial contributions $\pm 1\sigma$ of the 10,000 model runs (Table 1; Extended Data Fig. 5) (L. 1002-1003). We also report the equivalent mean $\pm 1\sigma$ of the Bayesian outputs from the two different model run scenarios (Table 1; Table S5). We hope this provides more confidence in the uncertainty of the values we report, in contrast to the ranges previously reported.

In general, there are many numbers reported throughout the paper and in Methods section. In Methods, the purpose of each section is often not stated clearly. For example, when I first read the section on “data extraction from HydroATLAS”, I had no idea what is this for, until I read to the section on “random forest model”, and realize that HydroATLAS is to extract catchment characteristics. but I think someone who does not do random forest model might still at loss. The Methods section generally lacks streamlining and organization, which makes it challenging to follow the quantification process, and to assess the validity and accuracy of these numbers.

We appreciate that our Methods descriptions may not have been as transparent as we had intended. We have now expanded the *Study Approach* section at the beginning of the Methods to give a full, sequential outline of each step we took in our analysis (L. 579-626). This is then followed by a detailed, step-by-step breakdown of each aspect in the full methods section. We have also edited the method throughout for readability and greatly expanded the section describing how we derived the flux numbers to increase clarity into how we generated the numbers presented in our analysis (L. 932-1012).

Additional comments:

1. The abstract stated in abstract that “We show that river CO₂ and CH₄ emissions are vulnerable to anthropogenic perturbations of both short-term and long-term carbon cycles, which have the potential to increase this substantial routing of carbon from land to the atmosphere.” This statement is more a speculation than a conclusion. It is not substantiated by data and figures. Some line 272-282 documents this calculation but this is purely depends on the number 47-56% that I am concerned about.

We have now updated the final sentences of the abstract to reflect this comment and those from Reviewer 2 (L. 39-44):

“This previously unrecognised release of pre-industrial aged carbon to the atmosphere from long-term soil, sediment and geologic carbon stores via lateral hydrological routing equates to $1.2 \pm 0.3 \text{ Pg C yr}^{-1}$, similar in magnitude to terrestrial net ecosystem production. A consequence of this flux is a greater than expected net loss of carbon from aged organic matter stores on land. This requires a reassessment of the fate of anthropogenic carbon in terrestrial systems, and in global carbon cycle budgets and models.”

2. The main text only has three figures. Why not put some of the extended / SI figures in main text? For example, some of the random forest figures indicating influential factors. The authors plotted a lot of bar figures for different factors (drainage area, biome, ...). It is however not clear though which factors are the most influential factors. Would it be more effective to plot a figure ranking the most influential factors?

We have now updated the reporting and discussion of the random forest model analysis to make clear the ranking and relative importance of the different variables – please see our response to Reviewer 1 on this [R1-3]. In the revised figure, we replace the previous visualisation, which used point sizes to represent the importance of variables as determined by the Random Forest model, with a ranking bar chart and the linear regressions for the variables identified as most important by the model (IncMSE > 15). The bars are arranged in descending order of variable importance quantified by IncMSE from random forest model, and the original variable names rather than abbreviations are labelled on the chart for clarity (Tables S4-5).

3. Line 121-143, this part talks a lot about extended figures and supporting SI figures. It would make it much easier for the readers if the authors can combine these figures and move them to the main text

We have now incorporated the core of these figures into a revised Fig. 2, after combining the previous Fig. 1 and 2 – please see responses to Reviewers 1 and 2, and the figures in the revised manuscript.

4. Line 133-143, what is the reasoning behind dividing the catchments into small and bigger ones? Do the authors expect different drivers? If so, state explicitly. The approach makes the sample size smaller and harder for a random forest model to figure out the predominant influential factors. Do you expect different ranking for the importance of the feature? Can you provide feature importance ranking for small and large rivers? This could help understand what drivers of old carbon dominate in different category. Right now, the paper only provide a list, not a ranking figure.

This is partly operational (the resolution of HydroATLAS is not great below 10 km²) and partly mechanistic (smaller catchments could theoretically have older carbon because they are more vulnerable to localised disturbances of old carbon compared to larger catchments where old signals can be masked by wider-scale processes). We now explore this mechanism in the main text (L. 127-132) and explain the HydroATLAS limitation in our updated study approach at the start of the methods (L. 601-607).

Operationally, because of the low resolution of the HydroATLAS data, there was a high chance of mismatching catchment information to the $F^{14}C$ measurements collected from small catchments (< 10 km²). This meant it was difficult to ensure the correct catchment characteristics were extracted for each value in the database. For small catchments we therefore used data only at the reach-scale (catchment characteristics for a 1 km radius from the sampling point), while for large catchments we were able to use the full catchment data. We checked that the catchment characteristics were well-matched by comparing catchment size from the original publications and from HydroATLAS as described in the methods (Fig. S6). We have also clarified this in the *Study Approach* at the beginning of the methods.

We have also now updated the reporting and discussion of the random forest model analysis to make clear the ranking and relative importance of the different variables – please see our response to Reviewer 1 on this [R1-3].

5. Fig. 3: the figure can be modified to reflect how this conclusion from this work change the conceptual model compared to the existing conceptual model. For example, if the numbers reported here are true, this figure can be something like a “before” and “after” figure, with different arrow size to reflect the larger old carbon flux compared to the existing conceptual model. The current figure highlights various processes contributing to carbon of different age, which is not the major finding of this work and are barely discussed / supported in the paper.

We have moved the original Fig. 3 to the Extended Data and have now included a new figure (Fig. 3) in the main text to demonstrate the impact our findings have on our current understanding of the global terrestrial carbon balance. In the new figure, we show the previous paradigm and updated impact on the global carbon budget of our findings.

6. Line 810 – 811: 90% training and 10% testing; this seems larger percentage of training compared to the typical 75%-25% division, although it is probably ok.

Here, the 90% and 10% refer to the 10-fold cross-validation ratio, not the training and testing split. This 10-fold cross-validation provides a more stable and reliable model performance evaluation by repeatedly splitting the data for training and validation, reducing overfitting and maximizing data usage. This cross-validation method divides the data into 10 subsets (folds) and using 9 subsets for training while the remaining subset is used for validation. This ensures that the model is trained and validated on different data subsets, reducing the risk of overfitting to any particular portion of the data. In general, if you’re already using a 10-fold cross-validation, it is not necessary to set aside a separate dataset for testing model performance, particularly when working with limited data. Therefore, since the goal of using the Random Forest model in this study is to analyse the importance of variables rather than to make predictions, and given that the dataset in our study is not sufficiently large, we did not set aside a separate portion of the data for model testing. Instead, we use all the data for 10-fold cross-validation.

7. The figures with error bars do not include the definition of error bars. This includes Fig. 1 and many extended data figures.

These are now described in the revised figures (Fig. 1B, Fig. 2, Extended Data Fig. 2, Extended Data Fig. 6).

Referees' comments:

Referee #1 (Remarks to the Author):

I feel the authors' revisions were responsive to my main concerns and they were able to substantially improve their manuscript. I only have those two very minor comments listed below. Overall, I think this study is a very valuable contribution to the scientific literature on river C cycling. I suggest its publication after these minor revisions.

L41 : Maybe you rather mean « net ecosystem exchange » (NEE, = exchange of CO₂ with atmosphere) rather than “net ecosystem production” (NEP) . NEP is defined as balance between gross primary production and ecosystem respiration. Global values of NEP are much higher than that of NEE, because NEP does not account for the recycling of human appropriated biomass (harvested wood, crop, and fodder plants for livestock), but those are included in NEE. Global NEE is ~2 Pg C/yr. Global NEP is estimated somewhere between 5 and 10 Pg C/yr.

Corrected to Net Ecosystem Exchange here and in the opening paragraph of the main text – we thank the reviewer for spotting this.

Extended data figure 3 and 4: For the plots on the right hand side, I wonder if it wasn't better to have partial dependence plots, as those better describe the behavior of the fitted RF model in response to predictor values. At least you could add the corresponding partial dependence plots to the SI. I think these plots are very helpful for interpreting the RF model.

We thank the reviewer for this suggestion. We had originally used simple linear plots to demonstrate the relationships between the underlying data and the model parameters, but we agree that partial dependence plots are a more accurate representation of the relationships highlighted by the random forest model. We have therefore replaced the panels suggested in Extended Data Fig. 3-4, and added a short explanation in the methods:

We assessed the association between predictor variables and $F^{14}C_{atm}$ with partial dependence plots using the *pdp R* package⁶⁷. The partial dependence plots show how $F^{14}C_{atm}$ changes when a given input variable (Table S5) varies but all other variables are held constant within the random forest model. We performed the partial dependence analysis 10 times (mirroring the 10 iterations of random forest models from using 10-fold cross validation), and plotted the mean values from these 10 runs, with the variability across the runs indicated by the shaded area (Extended Data Fig. 3-4).

Extended Data Fig. 3 | Potential controls on DIC $F^{14}C_{atm}$ in large catchments (> 10 km²).

(A) Ranking of variables by their potential importance in describing the database $F^{14}C_{atm}$ values using a Random Forest model; * denotes statistically significant correlations with $F^{14}C_{atm}$ ($p < 0.05$) from an independent Spearman's rank test calculated independently for each variable. (B-F) Partial dependence plots showing how $F^{14}C_{atm}$ responds to the variations of a specific catchment characteristic while all other characteristics were held constant in the random forest model – plots are only shown for catchment characteristics identified by the random forest model as potentially significant controls on DIC $F^{14}C_{atm}$: (B) mean elevation, (C) mean annual precipitation, (D) mean annual air temperature (in °C multiplied by 10), (E) the extent of karst area and (F) the extent of forested area within the catchment upstream of the sampling location; R^2 and p -values are from linear regression; the increase in the mean square error (IncMSE) is from the random forest modeling, the Spearman's ρ values were calculated independently for each variable; note the logarithmic x-axes in panels (B-C); see Table S4-5 for full description of variables.

Extended Data Fig. 4 | Potential controls on DIC $F^{14}C_{atm}$ in small catchments ($\leq 10 \text{ km}^2$).

(A) Ranking of variables by their potential importance in describing the database $F^{14}C_{atm}$ values using a Random Forest model; * denotes statistically significant correlations with $F^{14}C_{atm}$ ($p < 0.05$) from an independent Spearman's rank test calculated independently for each variable. (B-E) Partial dependence plots showing how $F^{14}C_{atm}$ responds to the variations of a specific catchment characteristic while all other characteristics were held constant in the random forest model – plots are only shown for catchment characteristics identified by the random forest model as potentially significant controls on DIC $F^{14}C_{atm}$: (B) mean elevation, (C) soil organic carbon content, (D) soil sand fraction and (E) mean annual air temperature (in $^{\circ}\text{C}$ multiplied by 10) within the 1 km^2 radius reach of the sampling location; R^2 and p-values are from linear regression; the increase in the mean square error (IncMSE) is from the random forest modeling, the Spearman's p-values were calculated independently for each variable. Note the logarithmic x-axes in panels (B–E); see Table S4-5 for full description of variables.

This analysis altered the interpretation of the outputs very slightly, please see our response to Reviewer 3 re. L165-168.

Referee #2 (Remarks to the Author):

The revised paper by Dean et al. maintains the core message of the initial version but presents it with greater clarity and stronger justification. I thank the authors for their well articulated replies to my comments as well as to those from other reviewers. I support the addition of the downstream DIC flux into their mass balance (as per R1's suggestions), which I had overlooked in my initial review.

I appreciate the authors' efforts to examine the potential for sampling bias in their dataset. This additional analysis adds confidence to their results. I also independently checked the data in Table S1 and searched for occurrences of 'peatland' or 'permafrost' or 'glacier' in titles. This returned <200 occurrences out of 1,195 data points, which suggests that sampling bias towards systems where we expect old carbon should be minimal.

We thank the reviewer for undertaking this independent check. We explored study titles similarly and have now included this analysis in the supplementary text where we discuss bias within the database – see the second paragraph of Supplementary Text S4:

Searching the article titles in our database for terms including “peat”, “permafrost”, “glacier/glacial” and “weathering” returned 282 instances, or ~24% of the database, suggesting there is limited bias towards studies specifically targeting old carbon.

I am still not entirely convinced about the use of a 3-endmember Bayesian analysis, as a single tracer may not be enough to resolve 3 sources. The fact that the authors obtained realistic results might be because they used very narrow ranges of $F^{14}C$ values for the 3 sources, which might have helped constrain the potential solutions – but the extent of this influence should be considered. The paper by Stock et al. (<https://peerj.com/articles/5096/>) has an interesting discussion about underdetermined systems and the way Bayesian models handle them. I understand that the Bayesian approach is not central to the findings, and I do not wish to delay the publication of this important paper – but I would invite the authors to at least mention in the Methods the potential limitations of using this approach. Alternatively, please consider running a sensitivity analysis of the impact of using broader ranges for the 3 sources. If results change very substantially when using slightly broader source values, this would indicate a high sensitivity of the model to assumptions – which should be acknowledged. The authors could also consider adding $d^{13}C$ as a second tracer. I would understand if they were reluctant to do so though, because $d^{13}C$ is not conservative and in-stream processes may complicate the $d^{13}C$ signal, so adding it might introduce noise rather than help resolve source contributions – but might be worth exploring.

We have added a clearer statement on the limitations of the Bayesian approach as a final paragraph in Supplementary Text S3:

The Bayesian isotope mixing model outputs suggests that decadal carbon sources may have accounted for 31-64% (mean = 50%, median = 51%) of emissions, millennial carbon inputs 23-47%, and petrogenic carbon 5-28% (Table S2; where total river vertical CO_2 emissions = $2.0 \pm 0.2 \text{ Pg C y}^{-1}$). There is a considerable amount of variability in these estimated proportional contributions (Extended Fig. 6). This variability is to be expected given the diversity of catchments from which samples were collected, the number of potential sources used, and the range of possible contributions which could give rise to the $F^{14}C$ observations. It would be unconstructive to further constrain the input components of this analysis to reduce the wide range of potential solutions given the diversity of samples in the database; increasing the number of potential carbon sources would only spread the uncertainty across more sources (noting we chose these three sources following conceptual models elsewhere^{17,18}), and constraining the existing source $F^{14}C$ ranges would make them potentially less applicable across the type and timing of samples in the database. Despite these limitations, the values this model produces agree with the dual mixing model with Monte Carlo simulations where petrogenic contributions were used as a prior (as discussed in the main text; Table 1), overall supporting the predicted contributions from different aged carbon sources to river carbon emissions.

We would like to reiterate that the Bayesian analysis was only intended as a sense check for the petrogenic input constrained Monte Carlo simulations reported in the main text and as such a sensitivity analysis is unwarranted. And despite the limitations noted by the reviewer, the Bayesian analysis supported the Monte Carlo analysis findings (Table 1). As also noted by the reviewer, we have not attempted to add in the additional $\delta^{13}C$ isotope to these analyses because it is hugely challenging to account for fractionation due to the non-conservative nature of this isotope in DIC in rivers.

Congratulations on this important contribution.
Clément Duvert

Referee #3 (Remarks to the Author):

The manuscript has been thoroughly revised in response to comments from three reviewers. The statistical tests are appropriate, and the error bars are accurately reported. The additional explanation in the methods section has also improved the clarity of the approach. That said, I have a few points I would like the authors to further address:

1. The paper does not clearly define the physical meaning of F14C. Line 85 describes it as a surrogate for the isotopic composition of river CO₂ emissions and references source 34, but it does not explain how it is calculated or what it represents physically. Readers need to consult reference 34 to understand its meaning, which might be difficult for a broader audience. Providing a clear and accessible explanation would improve understanding. Additionally, the term "Fraction" may be misleading, as F14C is not a fraction in the conventional sense, such as a percentage, and thus requires clarification.

Fraction modern ($F^{14}C$) is a well-established terminology. It is an expression of the ¹⁴C content of the samples relative to the established "modern" atmospheric ¹⁴CO₂ baseline of 1950 CE. We have already added in indications of radiocarbon age calculated from ¹⁴C content in Figs 1 and 2 in the revised manuscript, and when reporting $F^{14}C$ values we have added in context where practical. Further we have already defined $F^{14}C$ in the second paragraph of the Methods summary. We have also defined it in the first sentence of the final paragraph of the introduction in the Main Text, prior to the line reference the reviewer highlights. We believe this is satisfactory to define the concept of "age" or "source" of carbon within the study systems. The confusion likely came from our mistake in equation (5), please see our response below.

2. In particular, Item 1 is relevant to the later global scale calculations. I don't think Equation (5) is correct. The F14C values in Equation (5) represent the fraction of carbon originating from decadal, millennial, or petro sources, which differ from the F14C in Equation (2). If the F14C in Equation (2) is being used, then Equation (5) does not represent a mass balance equation. For instance, if F14C_{decadal} = 1.1, F14C_{millennial} = 0.9, and F14C_{petro} = 0.6, this would result in F14C_{river} = 2.6, which exceeds the maximum values in your Fig 1C and is not meaningful. The global mean F14C_{river} is around 0.9 - 1.0 (line 104 and 105). If you are referring to the fractions of carbon from different age sources, distinct symbols should be used instead of F14C.

We apologise, this was a mistake when altering the symbology of the equations when expanding this part of the methods in response to this reviewer's original comments on the equations behind Table 1. The equation should read as a mass balance, as now described in the updated text in the revised Methods:

We can also express global river $F^{14}C$ of DIC and CO₂ ($F^{14}C_{river}$) as the mass balance of the three main carbon sources defined in this study, where the proportional contributions from all three carbon sources ($a + b + c$) sum to 1:

$$F^{14}C_{river} = a \times F^{14}C_{decadal} + b \times F^{14}C_{millennial} + c \times F^{14}C_{petro} \quad (5)$$

3. The approach is really to figure out within the overall total river DIC flux (equation 4), what is

the fraction of C in the three age categories. But then $F_{14_petro} = 0$ (line 985-986). Do you mean the fraction of the petro C is zero, or the F_{14C} in equation (2) relating to age is equal to zero. In either case, what is the assumption behind $F_{14_petro} = 0$?

Here we mean that the ^{14}C content of petrogenic carbon is zero, i.e. it contains no measurable ^{14}C above analytical background, which happens when carbon is approximately $> 60,000$ years old. This is equivalent to 0 in F_{14C} units, compared to the year 1950 which is 1 in F_{14C} units. Petrogenic carbon is by definition older than 60,000 years and therefore contains no measurable ^{14}C and thus we assume an F_{14C} value of 0. This is stated in the Supplementary Text S3, and we now add this information following equation (8) where this assumption is introduced:

We can then calculate the non-petrogenic F_{14C} value ($F_{14C_{decadal+millennial}}$), where the $F_{14C_{petro}}$ value = 0. The petrogenic source is assumed to contain no radiocarbon (i.e., $F_{14C} = 0.0$ Fraction Modern).

Detailed comments:

L32: usually we say “surface waters”, not “water surface”

Here we refer to emissions from river water surfaces to the atmosphere. We have corrected to “... from their water surfaces...” because we don’t want to infer other surface waters (e.g., lakes, ponds etc.) in this statement.

L127 – 132: could this mean larger rivers have high proportion of water coming from deep flow paths with more older carbon?

Yes, and/or the exposure/mobilisation of old carbon stores at landscape surfaces through processes such as erosion. We have added this interpretation here:

In contrast, we found that $F_{14C_{atm}}$ values were lower (older) as catchments got larger (Extended Data Fig. 2), suggesting that contributions of old carbon from deeper hydrologic flow paths or exposed old carbon stores to river CO_2 are occurring across large scales.

L165-168: that means warmer and more humid places flush more recent C to rivers and streams, which makes sense. Does that also mean more arid places tend to store C and contribute more to older carbon?

We agree with the reviewer and have altered the interpretation accordingly. The final interpretation has changed slightly due to the incorporation of partial dependence plots following comments from Reviewer 1:

Mean elevation (large and small catchments) and karst area (large catchments, only) had a negative relationship with $F_{14C_{atm}}$, indicating catchments with higher elevations and carbonate lithologies released more ^{14}C -depleted (older) DIC. Mean annual precipitation (large catchments) and temperature (large and small catchments) were generally positively related to $F_{14C_{atm}}$, although there was an upper limit to this influence in large catchments: above 2,000 mm rainfall and above 20 °C $F_{14C_{atm}}$ tended to decrease (Extended Data Fig. 3B, D). This suggests catchments receiving higher precipitation and with warmer temperatures tended to release less ^{14}C -depleted (younger) DIC, although the limit to this mechanism indicates that more arid or especially warm and wet regions may store more carbon and/or release older carbon.

L168: what is high “IncMSE”?

We refer to IncMSE > 15 here, we have added this to the sentence indicated:

In small catchments, the high (> 15) IncMSE values from the random forest model for soil organic carbon and sand content demonstrate the potential importance of small-scale controls on the age of DIC released by rivers, influencing organic carbon mobilisation and hydrologic flow paths.

L246 – 247, “we propose ...” I don’t think this statement reflect the recent advances on our understanding of how water flows the landscape and carbon dissolved carbon transport to streams and rivers. a lot of carbon can come from depth deeper than soils. At least I would not say “... majority .. is produced from near surface or ... surface litter”. Most precipitated waters infiltrates and flows through subsurface of different depths. Water flowing through surface / top soils is typically very small. See, for example,

Liu et al., 2024: <https://www.nature.com/articles/s41561-024-01483-5>

Stewart et al., 2024: <https://agupubs.onlinelibrary.wiley.com/doi/10.1029/2023WR035940>

McCormick et al., 2021: <https://www.nature.com/articles/s41586-021-03761-3>

Tune et al., 2020: <https://agupubs.onlinelibrary.wiley.com/doi/full/10.1029/2020JG005795>

There appears to be some confusion here due to our editing of this section in our revised manuscript. The three paragraphs from this point are intended to refer to i) decadal carbon source, ii) millennial carbon sources, and iii) petrogenic carbon sources. We have now edited this section to clarify this progression, and included these important references highlighted by the reviewer:

The second largest proportion ($41 \pm 16\%$) of the CO₂ emitted by rivers is attributed to rapid, decadal carbon cycling through ecosystems (Fig. 3B). The majority of this decadal-aged proportion of river CO₂ is likely produced at the near surface via root respiration and/or surface litter decomposition. Some of this CO₂ may be used during chemical weathering of carbonate and silicate minerals to generate DIC, and some carried as dissolved CO₂ laterally to rivers and streams. Within-river aquatic metabolism is likely to supplement these rapid-cycling river CO₂ emissions¹⁰ alongside degradation of young and reactive river dissolved and particulate organic carbon^{44,47}. This fraction of river CO₂ emissions is a loss pathway from ecosystem respiration, whose transit time for carbon is typically on the order of years to decades³⁶.

The largest proportion ($52 \pm 16\%$) of river CO₂ emissions is sourced from millennial-aged carbon based on the global scale assessment here (Fig. 3B). Hydrological flow paths can mobilise dissolved CO₂ and DIC produced by soil respiration from deeper in the soil profile. This depth may coincide with the production of CO₂ via root respiration, linking decadal through to millennial aged CO₂ sources. However, inputs of older CO₂ from deeper in soil profiles, recently eroded or degraded soil surfaces, hyporheic zones and degradation of older river dissolved and particulate organic carbon could all contribute^{13,478}.

The remaining $7 \pm 1\%$ of river CO₂ emissions is derived from petrogenic carbon (Fig. 3B). Hydrological flow paths can also readily reach deeper into the bedrock underlying soils, supplying rivers⁴⁹, soils²⁶ and plants⁵⁰, and connectivity can also occur where bedrock is exposed, or soil coverage is minimal. The petrogenic carbon contained within carbonate rocks and rock organic matter can thus be mobilised by chemical weathering and erosion and delivered to river systems (Fig. 3).

The latter two old (millennial and petrogenic) components of river CO₂ may not necessarily have contributed to local ecosystem respiration. Instead, they represent a leak of older terrestrial carbon that escapes to the atmosphere via river surfaces (Extended Data Fig. 7).

We have also then accordingly edited the second to last paragraph in this final section to improve the logical progression:

Whether or not anthropogenic perturbation has increased the leak of old carbon to the atmosphere via rivers that we observe here, remains a significant knowledge gap. However, the dataset shows a trend of increasing age of $F^{14}C_{\text{atm}}$ in river DIC, CO_2 and CH_4 during the observation period (Extended Data Fig. 1). This could indicate increasing emissions of old carbon through time due to destabilisation of global soil carbon stocks^{14,15,31,48,52}, and changes to weathering, erosion and rock oxidation rates^{21,38,53} as a result of climate and anthropogenic perturbations. Anthropogenic climate change may increase CO_2 supply to rivers as soils warm and/or get wetter, and microbial respiration increases⁵⁴, while the delivery of DIC and CO_2 from rock weathering may also increase as landscapes warm^{22,53} [text moved from the section above]. However, we don't know if the trend in Extended Data Fig. 1 is due to increased perturbation (Extended Data Fig. 7), the declining atmospheric $^{14}\text{CO}_2$ signal moving through the biosphere (Fig. 1C), sampling bias (Supplementary Information S4), or a combination of these. Regardless, our analysis indicates that river CO_2 emissions are responsive to inputs from old carbon sources that may occur under past, present and future anthropogenic perturbations, which could increase under direct anthropogenic disturbance regimes such as landscape drainage, clearance, burning, and agricultural soil cultivation, as well as due to anthropogenic climate change [text moved from the above section].